# A Bio-Indicator Pilot Study Screening Selected Heavy Metals in Female Hair, Nails, and Serum from Lifestyle Cosmetic, Canned Food, and Manufactured Drink Choices

**DOI:** 10.3390/molecules28145582

**Published:** 2023-07-22

**Authors:** Asmaa Fathi Hamouda, Shifa Felemban

**Affiliations:** 1Department of Biochemistry, Faculty of Science, University of Alexandria, Alexandria 21111, Egypt; 2Department of Chemistry, Faculty of Applied Science, University College-Al Leith, University of Umm Al-Qura, Makkah 21955, Saudi Arabia; sgflemban@uqu.edu.sa

**Keywords:** heavy metals, hair, nail, serum, cosmetic, CBC, cholinesterase, p53, AutoDock

## Abstract

Lifestyles, genetic predispositions, environmental factors, and geographical regions are considered key factors of heavy metals initiatives related to health issues. Heavy metals enter the body via the environment, daily lifestyle, foods, beverages, cosmetics, and other products. The accumulation of heavy metals in the human body leads to neurological issues, carcinogenesis, failure of multiple organs in the body, and a reduction in sensitivity to treatment. We screened for Cr, Al, Pb, and Cd in selected foods, beverages, and cosmetics products depending on questionnaire outcomes from female volunteers. We also screened for Cr, Al, Pb, and Cd on hair, nails, and serum samples using inductively coupled plasma mass spectrometry (ICP-MS) from the same volunteers, and we analyzed the serum cholinesterase and complete blood picture (CBC). We performed an AutoDock study on Cr, Al, Pb, and Cd as potential ligands. Our results indicate that the most elevated heavy metal in the cosmetic sample was Al. In addition, in the food and beverages samples, it was Pb and Al, respectively. The results of the questionnaire showed that 71 percent of the female volunteers used the studied cosmetics, food, and beverages, which were contaminated with Cr, Al, Pb, and Cd, reflecting the high concentration of Cr, Al, Cd, and Pb in the three different types of biological samples of sera, nails, and hair of the same females, with 29 percent of the female volunteers not using the products in the studied samples. Our results also show an elevated level of cholinesterase in the serum of group 1 that was greater than group 2, and this result was confirmed by AutoDock. Moreover, the negative variation in the CBC result was compared with the reference ranges. Future studies should concentrate on the actions of these heavy metal contaminations and their potential health consequences for various human organs individually.

## 1. Introduction

Heavy metals are metallic hazards, which may be considered pseudo-elements that conflict with the metabolic role of the human body. Some metals such as aluminum (Al) can be eliminated from the body, but other heavy metals are stored in the body leading to chronic disease [1]. The toxic effects of heavy metals depend on the root, dose, and duration of exposure. Heavy metals lead to an indication of lipid peroxidation products, genetic mutation, and interaction with many different body proteins, such as albumin, and cell receptors [2,3,4,5]. A large amount of information on the relationship between heavy metals and health problems has been provided in a previous publication [6].

Heavy metals are considered a major cause of human toxicity, infertility, chronic and immune diseases, DNA alterations, and other issues. Hence, continuous screening and monitoring of environmental toxicity in the human body such as heavy metals are vital [1,7,8,9]. In addition, the results should always be compared to reference ranges that are determined by various factors, such as the kind of population, race, age, lifestyle, sex, geographical region, and the analytical method of heavy metal analysis; as a consequence, the reference values vary (see Table 1) [1,7,8,9].

Mineral trace elements and heavy metals in hair, nails, and serum significantly predict nutritional value, risk of malnutrition, and predisposition to certain illnesses. However, an analysis of metals in one cm of hair reveals the genuine concentration over the growth period and reflects the level of metals in the serum at the time the hair was grown [10,11]. An analysis of metals in hair is an indicator of forensic and environmental pollution and the concentration of mineral, trace element, and heavy metal toxicity, as their accumulation and longevity are ten times the concentration of that in urine or blood serum samples over a three month period (see Table 1) [10,11,12].

Furthermore, hair and nails are considered end-fate metabolic products, and they are long-term loaded with heavy metals and minerals. Human nails also reflect concentrations of the end productivity of metals and minerals in the metabolism, which lead to illness and toxicity risk, for at least three months to one year of exposure more than that of urine or blood serum samples [13].

Heavy metals, such as chromium (Cr), aluminum (AL), cadmium (Cd), and lead (Pb), enter the human body in several ways, such as soil, water, fresh and manufactured food, fish, beverages, paint, detergents, cosmetics products, dairy products, and others daily life products and supplies [14,15,16].

Heavy metals react biologically in vivo with metal cations after losing electrons and cause acute or chronic diseases in different organs of the body [1,6,17]. The highest exposure to heavy metals leads to severe complications in many organs, including gastrointestinal illness, nephrotoxicity, and carcinogenicity. Otherwise, the lowest exposure leads to neuropsychiatric issues, such as anxiety and chronic fatigue [1,6,17]. Aluminum contamination is present in water and food, as well as food additives, aluminum cooking tools, and other manufacturing products. Toxicity includes the initiation of inflammatory aluminous of the lung, neurotoxicity, nephrotoxicity, bone issues, encephalopathy, myelotoxicity, and anemia [1,6,17].

Cadmium (Cd) exposure may be involved through smoking and food and drink contamination, which is capable of increasing the concentrations of Cd blood and urine, leading to bone issues and kidney failure, as well as initiation of carcinogenesis in the human body [1,6,17].

Lead (Pb) is rare in the Earth’s soil. The largest amount of environmental pollution, such as gasoline and, in essence, lead, which is used in different industrial activities, including the manufacturing of food and beverages, has led to various contaminants being absorbed, starting with the skin and nose, inducing inflammation, digestion toxicity, respiratory difficulties, neurotoxicity, and chronic diseases, such as cardiovascular and kidney disease [1,6,17].

Food, drinks, water, and cosmetics are routinely used in daily human life, and they are contaminated with heavy metals, such as Cr, AL, Cd, and Pb, which are used as preservatives or coloring agents in many products [14,15]. Previous publications detected AL, Cd, Pb, and other heavy metals in many cosmetics products used in Saudi Arabia [14,15]. Furthermore, there have been many investigations on the presence of many heavy metals in fresh fruits and vegetables, fish, meats, cheeses, and dairy products, as well as canned foods and drinks [14,15].

Heavy metal tests and continuous screening and monitoring are important to check for abnormal levels of toxic or potentially harmful metals. Individuals are exposed to toxic metals in the environment, including food, beverages, fish, cosmetics, and water, all of the time [1,6,12,17]. Hence, tiny amounts of these metals can initiate acute or chronic heavy metal poisoning. Symptoms of acute heavy metal poisonings include nausea, diarrhea, vomiting, acute renal failure, gastrointestinal hemorrhage, hemolysis, dermatitis, and sustained pain. Furthermore, chronic diseases caused by heavy metal toxicity include lung cancer, pulmonary fibrosis, neurotoxicity, nephrotoxicity, bone disease, encephalopathy, myelotoxicity, anemia, pneumonitis, proteinuria, diabetes, osteomalacia, and other health issues (Table 1) [1,6,12,17].

From this point of view, we screened for the presence of the studied heavy metals in the serum, hair, and nails of female university students. We also analyzed the concentrations of the studied heavy metals in selected food, drink, water, and cosmetic products used by the female students as reported in a questionnaire study. We found a relationship between the heavy metals and the biological profile of CBC and the cholinesterase enzyme. We also conducted a docking analysis of the interactions of the studied heavy metals and selected protein receptors. Figure 1 shows a diagram of the experiment.

**Table 1 molecules-28-05582-t001:** Studied heavy metals’ source, effects, and reference range [1,6,7,8,9,12].

Heavy Metal	Source [1,6,7,8,9,12]	Effect [1,6,7,8,9,12]	Serum Reference Range (µg/L) [1,6,7,8,9,12]
Chromium (Cr)	Manufactured food, drink, water, and cosmetic products; steel industries; tanneries; fly ash; soil	Acute renal failure, gastrointestinal hemorrhage, hemolysis, lung cancer, and pulmonary fibrosis	(0.1–2.1)
Aluminum (AL)	Manufactured food, drink, water, and cosmetic products; cooking tools; soil	Neurotoxicity, nephrotoxicity, bone disease, encephalopathy, myelotoxicity, and anemia	(0–9)
Cadmium (Cd)	Manufactured food, drinks, water, and cosmetic products; plastic stabilizers; electroplating; phosphate fertilizers; paints and pigments; soil; smoking	Pneumonitis, proteinuria, osteomalacia, and lung cancer	(0–4.9)
Lead (Pb)	Manufactured food, drinks, water, and cosmetic products; herbicides; batteries waste; leaded fuel; insecticides; soil	Vomiting, diarrhea, abdominal pain, osteoporosis, neurologic degeneration, copper deficiency, and anemia	(0–190)

## 2. Results

### 2.1. Screening Heavy Metals in Nonbiological Samples

The questionnaire outcome in the current work reported that 71 percent of the female volunteers used the studied cosmetics, food, beverages, and water, and 29 percent reported that they do not use those samples. Figure 2 shows concentrations of Cr, Al, Cd, and Pb in μg/g in the studied and used products in the current investigation. Furthermore, Figure 2A shows the concentration of Cr, Al, Cd, and Pb in each cosmetic product analyzed separately in the current study in μg/g, from sample one to sample nine (sample 1 = hair cream; sample 2 = skin cream; sample 3 = shampoo; sample 4 = shower gel; sample 5 = body lotion 1; sample 6 = body lotion 2; sample 7 = handwash; sample 8 = skin cream; sample 9 = toothpaste). The report outcomes indicate that the most elevated heavy metal in all of the samples, from sample 1 to sample 9, was Al. Figure 2B reports that the most contaminated cosmetic sample, according to the cumulative frequency of the rule (80/20) of the Pareto curve, in descending order, was as follows: sample 9, sample 3, sample 4, sample 6, sample 8, sample 5, sample 7, sample 1, and sample 2.

In addition, Figure 2C shows a heat map of the concentrations of Cr, Al, Cd, and Pb in each cosmetic product, and it can be observed that the most elevated heavy metal was Al in all of the studied cosmetic products after which followed Pb, Cr, and Cd.

Moreover, Figure 3A shows the concentrations in μg/g of Cr, Al, Cd, and Pb in each food product analyzed in the current study, from sample one to sample eight (sample 1 = canned meat 1; sample 2 = canned meat 2; sample 3 = cheese sample 1; sample 4 = cheese sample 2; sample 5 = cheese sample 3; sample 6 = canned fish 1; sample 7 = canned fish 2; sample 8 = canned fish 3). The report outcomes indicate that the most elevated heavy metal in all of the samples, from sample 1 to sample 9, was Pb and Al. Figure 3B reports that the most contaminated food sample, according to the cumulative frequency of the rule (80/20) of the Pareto curve, in descending order, was as follows: sample 6, sample 7, sample 4, sample 8, sample 5, sample 1, sample 2, and sample 3. In addition, Figure 3C shows a heat map of the concentrations of Cr, Al, Cd, and Pb in each food product, separately, and it can be observed that the most elevated heavy metal was Al in all of the studied foods products, followed by Pb, Cd, and then Cr.

On the other hand, Figure 4A shows the concentration of Cr, Al, Cd, and Pb in each beverage and water product, separately, analyzed in the current study in μg/g, from sample one to sample seven (sample 1 = manufacturer juice 1 (mango); sample 2 = manufacturer juice 2 (orange); sample 3 = manufacturer juice 3 (guava); sample 4 = manufacturer soft drink 4; sample 5 = water sample 1 brand; sample 6 = water sample 2 brand; sample 7 = water sample 3 brand). The report outcomes indicate that the most elevated heavy metal in soft drink sample 4 was Cr, followed by Al and then Pb.

Figure 4B reports that there were no contaminated beverage or water samples according to the cumulative frequency of the rule (80/20) of the Pareto curve, and only sample 4, according to the cumulative frequency of the rule (80/20) of the Pareto curve, was contaminated with heavy metals. In addition, Figure 4C shows a heat map of the concentrations of Cr, Al, Cd, and Pb in each beverage and water product separately, and it presents the most elevated heavy metals which were Cr, followed by Al and then Pb in sample 4 only.

### 2.2. Screening Heavy Metals in the Biological Samples (Serum, Hair, and Nails) and Cholinesterase, CBC, and the Relationship among Parameters

Table 2 and Figure 5 show the data represented as the mean ± SD of the four heavy metal elements that were analyzed in the three different biological samples (i.e., serums, nails, and hair) in group 1 vs. group 2. Table 2 and Figure 5A show that the concentration of Cr (μg/L) in the serum for group 1 was 455.6 ± 168.9 vs. 168.8 ± 46 in group 2, *p* ≤ 0.05. On the other hand, the concentration of Al (μg/L) in the serum was 1.68 ± 2 in group 1 vs. 0.64 ± 1.14 in group 2, as shown in Table 2 and Figure 5A, *p* ≤ 0.05. Also, the concentration of Cd (μg/L) in serum group 1 was 9.43 ± 3.43 vs. 3.08 ± 1.62 in group 2, *p* ≤ 0.05. The concentration of Pb (μg/L) in serum group 1 was 0.33 ± 0.23 vs. 0.25 ± 0.17 in group 2, *p* ≤ 0.05, according to Table 2 and Figure 5A, while the concentration of cholinesterase (U/L) in the serum was 7409.7 ± 1953.6 in group 1 vs. 5883.4 ± 1457.4 in group 2, as can be seen in Table 2 and Figure 5B, *p* ≤ 0.05.

As Table 2 and Figure 5C show, the concentration of Cr (μg/g) in the nails in group 1 was 1725.6 ± 256.9 vs. 1282.7 ± 267.3 in group 2, *p* ≤ 0.05. The concentration of Al (μg/g) in the nails in group 1 was 34.8 ± 2.84 vs. 27.7 ± 5.54 in group 2, *p* ≤ 0.05, while the concentration of Cd (μg/g) in the nails in group 1 was 89 ± 4.58 vs. 61.6 ± 4.75 in group 2, *p* ≤ 0.05. On the other hand, the concentration of Pb (μg/g) in the nails in group 1 was 26.5 ± 4.54 vs. 18.5 ± 2.98 in group 2, *p* ≤ 0.05, as can be seen in Table 2 and Figure 5C.

Table 2 and Figure 5D show that the concentration of Cr (μg/g) in the hair in group 1 was 1511 ± 291 vs. 1107.5 ± 63.6 in group 2, *p* ≤ 0.05. The concentration of Al (μg/g) in the hair in group 1 was 26.14 ± 3.17 vs. 15 ± 2.19 in group 2, *p* ≤ 0.05. The concentration of Cd (μg/g) in the hair in group 1 was 56.8 ± 3.60 vs. 36.8 ± 3.63 in group 2, *p* ≤ 0.05. The concentration of Pb (μg/g) in the hair in group 1 was 16.5 ± 2.78 vs. 12 ± 1.44 in group 2, *p* ≤ 0.05.

Table 2 and Figure 5E show a heat map of the concentration of Cr, Al, Cd, and Pb in all of the different biological samples (i.e., serums, nails, and hair) in group 1 vs. group 2, where the greatest concentration of heavy metals, in descending order, was situated in the nails, hair, and then serum. In addition, the most elevated heavy metal was Cr in all of the studied biological samples.

Figure 5F reports that there was a cumulative frequency of the rule (80/20) of the Pareto curve in the biological samples’ concentration of cumulative heavy metals in descending order of the nails, hair, and then serum in group 1, as well as in the same order for group 2. This is under the level of the group, but if we arranged them as individuals, in descending order, they would be arranged as nails 1, hair 1, nails 2, hair 2, serum 1, and serum 2, where 1 and 2 refer to group 1 and group 2, respectively.

Furthermore, Table 2 shows, according to Fisher’s exact test, the percentage of normal cases of the different biological tissue samples (serums, nails, and hair) compared to the reference ranges.

Table 2 shows that the percentage of normal cases for Cr (μg/L) in the serum compared with the reference group 1 was 0% vs. 0% in group 2, *p* ≤ 0.05. The percentage of normal cases for Al (μg/L) in the serum compared with the reference group 1 was 98.6% vs. 100% in group 2, *p* ≤ 0.05. In addition, the percentage of normal cases of Cd (μg/L) in the serum compared with the reference in group 1 was 9.9% vs. 89.7% in group 2, *p* ≤ 0.05. Moreover, the percentage of normal cases of Pb (μg/L) in the serum compared with the reference in group 1 was 100% vs. 100% in group 2, *p* ≤ 0.05, while the percentage of normal cases of cholinesterase (U/L) in the serum compared with the reference in group 1 was 93% vs. 79.3% in group 2, *p* ≤ 0.05.

Table 2 shows that the percentage of normal cases of Cr (μg/g) in the nails compared with the reference in group 1 was 0% vs. 0% in group 2, *p* ≤ 0.05. On the one hand, the percentage of normal cases of Al (μg/g) in the nails compared with the reference in group 1 was 100% vs. 100% in group 2, *p* ≤ 0.05. On the other hand, the percentage of normal cases of Cd (μg/g) in the nails compared with the reference in group 1 was 0% vs. 0% in group 2, *p* ≤ 0.05. In addition, the percentage of normal cases of Pb (μg/g) in the nails compared with the reference in group 1 was 0% vs. 0% in group 2, *p* ≤ 0.05.

Table 2 shows that the percentage of normal cases of Cr (μg/g) in the hair compared with the reference in group 1 was 0% vs. 0% in group 2, *p* ≤ 0.05. On the one hand, the percentage of normal cases of Al (μg/g) in the hair compared with the reference in group 1 was 0% vs. 37.9% in group 2, *p* ≤ 0.05. On the other hand, the percentage of normal cases of Cd (μg/g) in the hair compared with the reference in group 1 was 0% vs. 0% in group 2, *p* ≤ 0.05. Moreover, the percent of normal cases of Pb (μg/g) in the hair compared with the reference in group 1 was 0% vs. 0% in group 2, *p* ≤ 0.05.

Table 3 shows data, represented as the mean ± SD of the complete blood count (CBC), that were analyzed for group 1 vs. group 2. Table 3 shows that the concentration of hemoglobin (g/d) in blood for group 1 was 10.2 ± 1.15 vs. 12.8 ± 0.51 for group 2, *p* ≤ 0.05. The concentration of hematocrit (%) was 38.3 ± 1.28 vs. 42.35 ± 1.29 in group 2, *p* ≤ 0.05. In addition, the concentration of RBCs (×1012/L) in group 1 was 5.04 ± 0.46 vs. 4.11 ± 0.20 in group 2, *p* ≤ 0.05. The concentration of the MCV (fl) in group 1 was 78.9 ± 3.93 vs. 83.2 ± 1.23 in group 2, *p* ≤ 0.05, while the concentration of MCH (pg) was 26 ± 0.60 in group 1 vs. 27.98 ± 0.64 in group 2 (Table 3), *p* ≤ 0.05. Otherwise, the concentration of MCH (pg) in group 1 was 26 ± 0.60 vs. 27.98 ± 0.64 in group 2, *p* ≤ 0.05. The concentration of MCHC (g/dL) in group 1 was 29.5 ± 0.65 vs. 33.8 ± 0.63 in group 2, *p* ≤ 0.05, while the concentration of RDW (%) in group 1 was 15.7 ± 0.63 vs. 13.77 ± 0.36 in group 2, *p* ≤ 0.05, in Table 3. On the other hand, the concentration of platelets (×10^9^/L) in group 1 was 233.5 ± 23 vs. 215 ± 13.3 in group 2, *p* ≤ 0.05. The concentration of MPV (fl) in group 1 was 7.35 ± 0.28 vs. 8.50 ± 0.38 in group 2, *p* ≤ 0.05. The concentration of WBCs (×10^9^/L) in group 1 was 6.40 ± 0.36 vs. 5.42 ± 0.27 in group 2, *p* ≤ 0.05. The concentration of the basophils (×10^9^/L) in group 1 was 0.05 ± 0.02 vs. 0.04 ± 0.01 in group 2, *p* ≤ 0.05. The concentration of eosinophils (×10^9^/L) in group 1 was 0.32 ± 0.06 vs. 0.29 ± 0.05 in group 2, *p* ≤ 0.05. The concentration of neutrophils (×10^9^/L) in group 1 was 4.09 ± 0.69 vs. 4.12 ± 0.19 in group 2, *p* ≤ 0.05. The concentration of lymphocytes (×10^9^/L) in group 1 was 2.22 ± 0.49 vs. 2.03 ± 0.17 in group 2, *p* ≤ 0.05. The concentration of Monocytes (×10^9^/L) in group 1 is 0.66 ± 0.10 vs. 0.51 ± 0.03 in group 2, *p* ≤ 0.05, in Table 3.

Furthermore, Table 3 shows, according to Fisher’s exact test, the percentage of normal cases of complete blood counts (CBCs) compared to the reference ranges. The percentage of normal cases of hemoglobin (g/dL) CBCs compared with the reference in group 1 was 11.3% vs. 100% in group 2. In addition, the percentage of normal cases of hematocrit (%) CBCs compared with the reference in group 1 was 84.5% vs. 100% in group 2, *p* ≤ 0.05. Moreover, the percentage of normal cases of RBC (×1012/L) CBCs compared with the reference in group 1 was 31.0% vs. 100% in group 2, and the percentage of normal cases of MCV (fl) CBCs compared with the reference in group 1 was 42.3% vs. 100% in group 2. The percentage of normal cases of MCH (pg) CBCs compared with the reference in group 1 was 9.9% vs. 100% in group 2. Furthermore, the percentage of normal cases of MCHC (g/dL) CBCs compared with the reference in group 1 was 0.0% vs. 100% in group 2. The percentage of normal cases of RDW (%) CBCs compared with the reference group 1 was 5.6% vs. 100% in group 2. The percentage of normal cases of platelet (×10^9^/L) CBCs compared with the reference in group 1 was 100% vs. 100% in group 2. The percentage of normal cases of MPV (fl) CBCs compared with the reference in group 1 was 33.8% vs. 100% in group 2. The percentage of normal cases of WBC (×10^9^/L) CBCs compared with the reference in group 1 was 100% vs. 100% in group 2. The percentage of normal cases of basophils CBCs compared with the reference in group 1 was 100% vs. 100% in group 2. The percentage of normal cases of eosinophil CBCs compared with the reference group 1 was 100% vs. 100% in group 2. The percent of normal cases of neutrophils CBCs compared with the reference in group 1 was 100% vs. 100% in group 2, the percentage of normal cases of lymphocyte CBCs compared with the reference in group 1 was 100% vs. 100% in group 2, and the percentage of normal cases of monocyte CBCs compared with the reference in group 1 was 98.6 vs. 100% in group 2, *p* ≤ 0.05 (Table 3).

Table 4, Table 5 and Table 6 show the correlations among the studied parameters in the biological samples. Evans (1996) suggested the following absolute values for r: 0.00–0.19, “very weak”; 0.20–0.39, “weak”; 0.40–0.59, “moderate”; 0.60–0.79, “strong”; 0.80–1.0, “very strong”, at *p* ≤ 0.05. Table 4 shows that there is only a weak significant positive correlation between Cd in serum (μg/L) and nails (μg/g) equal to 0.279 at *p* ≤ 0.05 in group 1. In addition, there is a significant negative correlation between Pb in serum (μg/L) and nails (μg/g) equal to −0.245 at *p* ≤ 0.05 in group 1.

Table 5 shows that there was only a weak significant positive correlation between serum cholinesterase (U/L) and Cr (μg/L) equal to 0.299 at *p* ≤ 0.05 in group 1. On the other hand, there was only a weak significant negative correlation between serum cholinesterase (U/L) and Al (μg/L) equal to −0.275 at *p* ≤ 0.05 in group 1 and −0.370 in group 2, respectively.

Table 6 shows that there was only a weak significant negative correlation between serum Cr (μg/L) and hemoglobin equal to −0.243 at *p* ≤ 0.05 in group 1. Also, Table 6 shows that there was only a moderate significant positive correlation between serum Cr (μg/L) and hematocrit equal to 0.522 at *p* ≤ 0.05 in group 1. Otherwise, there was only a weak significant positive correlation between serum Cr (μg/L) and MCV equal to 0.338 at *p* ≤ 0.05 in group 1. On the other hand, there was only a weak significant negative correlation between serum Cr (μg/L) wan platelets equal to −0.380 at *p* ≤ 0.05 in group 2. There was only a weak significant positive correlation between serum Al (μg/L) and MCHC equal to 0.248 at *p* ≤ 0.05 in group 1, and there was only a weak significant negative correlation between serum Al (μg/L) and platelets equal to −0.333 at *p* ≤ 0.05 in group 1. Otherwise, there was only a weak significant negative correlation between serum Cd (μg/L) and monocytes equal to −0.247 at *p* ≤ 0.05 in group 1 and a weak significant negative correlation between serum Cd (μg/L) and eosinophils equal to −0.371 at *p* ≤ 0.05 in group 2. There was only a weak significant negative correlation between serum Pb (μg/L) and hematocrit equal to −0.246 at *p* ≤ 0.05 in group 1 and a weak significant negative correlation between serum Pb (μg/L) and eosinophils to equal −0.268 at *p* ≤ 0.05 in group 1. Also, there was only a weak significant negative correlation between serum Pb (μg/L) and platelets equal to −0.384 at *p* ≤ 0.05 in group 2.

Also, there was only a weak significant positive correlation between serum cholinesterase (U/L) and hematocrit and WBCs equal to 0.409 and 0.240 at *p* ≤ 0.05 in group 1, respectively; additionally, there was a weak negative correlation between serum cholinesterase (U/L) and MCH equal to −0.313 at *p* ≤ 0.05 in group 1. On the other hand, there was a weak negative correlation between serum cholinesterase (U/L) and MCHC and monocytes equal to −0.483 and0.486 at *p* ≤ 0.05 in group 2, respectively (Table 6).

### 2.3. Potential AutoDock Interaction Scenarios for Heavy Metals with Selected Proteins, Receptors, and Hormones in the Human Body Examples of Heavy Metal Toxic Accumulation Effects

The cholinesterase target protein itself (PDB: 1P0P) and its target receptor (PDB: 6EP4) (Table 7, Figure 6) showed binding energy with the four heavy metals (Cr, Compound CID: 23976; Al, Compound CID: 5359268; Cd, Compound CID: 23973; Pb, Compound CID: 5352425). In the current work, as listed in Table 7 and Figure 6, the cholinesterase target protein (1P0P, reflects the binding affinity of the four heavy metals, Cr, Al, Cd, and Pb, ligands as follows: −8.49 (ligand position: 3/100), −8.69 (ligand position: 3/100), −8.46 (ligand position: 3/100), and −8.72 (ligand position: 3/60), respectively. Moreover, 6EP4 showed a binding affinity with the four heavy metals, Cr, Al, Cd, and Pb, ligands as follows: −8.66 (ligand position: 3/100), −8.72 (ligand position: 3/70), −6.9 (ligand position: 2/20), and −7.62 (ligand position: 9/40), respectively, (Table 7, Figure 6A–D).

The P53 gene target’s (PDB: 7VOU, 2LY4, and 2JTX) protein parts and its target receptor (PDB: 6VTH) (Table 7 and Figure 6) showed binding energy with the four heavy metals (Cr, Compound CID: 23976; Al, Compound CID: 5359268; Cd, Compound CID: 23973; Pb, Compound CID: 5352425). In the current work, as listed in Table 7 and Figure 6, 7VOU reflected binding affinity with the four heavy metals, Cr, Al, Cd, and Pb, ligands as follows: −8.72 (ligand position: 3/30), −8.6 (ligand position: 3/30), −8.69 (ligand position: 3/100), and −8.67 (ligand position: 3/40), respectively. On the other hand, 2LY4 showed a binding affinity with the ligands of the four heavy metals, Cr, Al, Cd, and Pb, as follows: −6.9 (ligand position: 2/20), −8.71(ligand position: 3/100), −8.65 (ligand position: 3/50), and −8.7 (ligand position: 3/30), respectively. Moreover, 2JTX showed a binding affinity with the ligands of the four heavy metals, Cr, Al, Cd, and Pb, as follows: −8.63 (ligand position: 3/100), −7.62 (ligand position: 9/70), −7.63 (ligand position: 9/70), and −8.69 (ligand position: 3/100), respectively. Furthermore, 6VTH reflected the binding affinity of the ligands of the four heavy metals, Cr, Al, Cd, and Pb, as follows: −8.74 (ligand position: 3/30), −8.65 (ligand position: 3/90), −8.7 (ligand position: 3/100), and −8.65 (ligand position: 3/100), respectively, (Table 7 and Figure 6A–D).

Table 7 and Figure 6 reflect the binding affinity of the four heavy metals’ ligands to both the investigated hormones and receptors as well. The dopamine target hormone itself (PDB: 5PAH), and the target receptor (PDB: 3PBL) (Table 7, Figure 6) showed binding energy with the four heavy metals (Cr, Compound CID: 23976; Al, Compound CID: 5359268; Cd, Compound CID: 23973; Pb, Compound CID: 5352425). 5PAH reflected the binding affinity with the ligands of the four heavy metals, Cr, Al, Cd, and Pb, as follows: −8.71 (ligand position: 3/100), −7.6 (ligand position: 9/40), −7.63 (ligand position: 9/30), and −8.73 (ligand position: 3/100), respectively (Table 7 and Figure 6A–D). While 3PBL showed binding affinity with the ligands of the four heavy metals, Cr, Al, Cd, and Pb, as follows: −7.65 (ligand position: 9/30), −8.71 (ligand position: 3/100), −8.74 (ligand position: 3/100), and −8.65 (ligand position: 3/100), respectively (Table 7 and Figure 6A–D). Otherwise, the estrogen hormone, itself (PDB: 1FDW), and the target receptor (PDB: 1L2J,4J26, and 4J24) (Table 7 and Figure 6) showed binding energy with the ligands of the four heavy metals, as listed in Table 7, where 1FDW showed a binding affinity with the ligands of the four heavy metals, Cr, Al, Cd, and Pb, as follows: −8.73 (ligand position: 3/100), −8.68 (ligand position: 3/100), −8.65 (ligand position: 3/90), and −8.57 (ligand position: 3/100), respectively (Table 7 and Figure 6A–D). Moreover, 1L2J showed a binding affinity with the ligands of the four heavy metals, Cr, Al, Cd, and Pb, as follows: −8.74 (ligand position: 3/100), −8.72 (ligand position: 3/100), −8.62 (ligand position: 3/100), and −8.67 (ligand position: 3/100), respectively (Table 7 and Figure 6A–D), while 4J26 showed a binding affinity with the ligands of the four heavy metals, Cr, Al, Cd, and Pb, as follows: −8.45 (ligand position: 3/100), −8.69 (ligand position: 3/100), −8.71 (ligand position: 3/100), and −8.7 (ligand position: 3/100), respectively (Table 7 and Figure 6A–D). In addition, 4J24 showed a binding affinity with the ligands of the four heavy metals, Cr, Al, Cd, and Pb, as follows: −8.71 (ligand position: 3/100), −8.71 (ligand position: 3/100), −8.48 (ligand position: 3/100), and −8.48 (ligand position: 3/100), respectively (Table 7 and Figure 6A–D).

The metallothionein target protein, itself (PDB: 1MHU), and its target receptor (PDB: 2MHU and 2F5H) (Table 7 and Figure 6) showed binding energy with the four heavy metals (Cr, Compound CID: 23976; Al, Compound CID: 5359268; Cd: Compound CID: 23973; Pb, Compound CID: 5352425). In the current work, as listed in Table 7 and Figure 6, 1MHU showed a binding affinity with the ligands of the four heavy metals, Cr, Al, Cd, and Pb, as follows: −8.7 (ligand position: 3/100), −8.69 (ligand position: 3/100), −8.65 (ligand position: 3/100), and −8.72 (ligand position: 3/60), respectively (Table 7 and Figure 6A–D), while 2MHU showed a binding affinity with the ligands of the four heavy metals, Cr, Al, Cd, and Pb, as follows: −8.69 (ligand position: 3/100),−8.6 (ligand position: 3/100), −8.72 (ligand position: 3/100), and −8.63 (ligand position: 3/100), respectively (Table 7 and Figure 6). In addition, 2F5H showed binding affinity with the ligands of the four heavy metals, Cr, Al, Cd, and Pb, as follows: −8.48 (ligand position: 3/100), −8.63 (ligand position: 3/100), −8.63 (ligand position: 3/100), and −8.73 (ligand position: 3/100), respectively (Table 7 and Figure 6A–D).

The keratin target protein itself (PDB: 6EC0) and its target receptor (PDB: 4ZRY) (Table 7 and Figure 6) showed binding energy with the four heavy metals (Cr, Compound CID: 23976; Al, Compound CID: 5359268; Cd, Compound CID: 23973; Pb, Compound CID: 5352425). In the current work, as listed in Table 7 and Figure 6, 6EC0 showed binding affinity with the ligands of the four heavy metals, Cr, Al, Cd, and Pb, as follows: −8.7 (ligand position: 3/100), −8.73 (ligand position), −8.49 (ligand position: 3/100), ad −8.49 (ligand position: 3/100), respectively, while 4ZRY showed a binding affinity with the ligands of the four heavy metals, Cr, Al, Cd, and Pb, as follows: −8.7 (ligand position: 3/100), −8.47 (ligand position: 3/100), −8.73 (ligand position: 3/100), and −8.68 (ligand position: 3/100), respectively (Table 7 and Figure 6A–D).

The protein kinase enzyme, itself (PDB: 1P4F and 5IKP), and its target receptor (PDB: 1LHR and 6E0R) (Table 7 and Figure 6) showed binding energy with the four heavy metals (Cr, Compound CID: 23976; Al, Compound CID: 5359268; Cd, Compound CID: 23973; Pb, Compound CID: 5352425). In the current work, as listed in Table 7 and Figure 6, 1P4F showed binding affinity with the ligands of the four heavy metals, Cr, Al, Cd, and Pb, as follows: −8.7 (ligand position: 3/100), −8.66 (ligand position: 3/100), −8.69 (ligand position: 3/100), and −8.26 (ligand position: 4/100); otherwise, 5IKP showed binding affinity with the ligands of the four heavy metals, Cr, Al, Cd, and Pb, as follows: −8.67 (ligand position: 3/100), −8.66 (ligand position: 3/100), −8.67 (ligand position: 3/100), and −8.67 (ligand position: 3/100), respectively (Table 7 and Figure 6). Furthermore, 1LHR reflected the binding affinity of the ligands of the four heavy metals, Cr, Al, Cd, and Pb, as follows: −8.45 (ligand position: 4/100), −8.7 (ligand position: 3/100), −8.71 (ligand position: 3/100), and −8.68 (ligand position: 3/100), respectively (Table 7 and Figure 6), while 6E0R reflected the binding affinity of the four heavy metals, Cr, Al, Cd, and Pb) as follows: −8.46 (ligand position: 3/100), −8.67 (ligand position: 3/100), −8.71 (ligand position: 3/90), and −8.48 (ligand position: 3/80), respectively (Table 7 and Figure 6A–D).

The beta-amyloid protein itself (PDB: 5TXJ, and 3T4G) and its target receptor (PDB: 3Q7G) (Table 7 and Figure 6) showed the binding energy with the four heavy metals (Cr, Compound CID: 23976; Al, Compound CID: 5359268; Cd, Compound CID: 23973; Pb, Compound CID: 5352425). 5TXJ showed the binding affinity with the ligands of the four heavy metals Cr, Al, Cd, and Pb, as follows: −8.68 (ligand position: 3/100), −8.71 (ligand position: 3/100), −8.64 (ligand position: 3/100), and −8.71 (ligand position: 3/100), respectively (Table 7 and Figure 6A–D).

3T4G showed binding affinity with the ligands of the four heavy metals, Cr, Al, Cd, and Pb, as follows: −8.69 (ligand position: 3/90), −8.72 (ligand position: 3/70), −8.7 (ligand position: 3/90), and −8.72 (ligand position: 3/100), respectively (Table 7), while 3Q7G reflected the binding affinity of the ligands of the four heavy metals, Cr, Al, Cd, and Pb, as follows: −8.46 (ligand position: 3/100), −8.64 (ligand position: 3/100), −8.71 (ligand position: 3/100), and −8.48 (ligand position: 3/100), respectively (Table 7 and Figure 6A–D).

The ATPase enzyme protein itself (PDB: 6WLW) and its target receptor (PDB: 6WM3) (Table 7 and Figure 6) showed binding energy with the four heavy metals (Cr, Compound CID: 23976; Al, Compound CID: 5359268; Cd, Compound CID: 23973; Pb, Compound CID: 5352425). 6WLW showed a binding affinity with the ligand of the four heavy metals, Cr, Al, Cd, and Pb, as follows: −8.46 (ligand position: 3/100), −8.71 (ligand position: 3/100), −7.15 (ligand position10/30), and −8.68 (ligand position: 3/90), respectively (Table 7 and Figure 6). Otherwise, 6WM3 showed binding affinity with the ligands of the four heavy metals, Cr, Al, Cd, and Pb, as follows: −7.21 (ligand position: 10/20), −8.64 (ligand position: 3/100), −8.69 (ligand position: 3/90), and −8.47 (ligand position: 3/100), respectively (Table 7 and Figure 6A–D).

Albumin protein, itself (PDB: 5UJB and 6M5D), and its target receptor (PDB: 6HSC, 2ESG, and 6YG9) (Table 7 and Figure 6) showed binding energy with the four heavy metals (Cr, Compound CID: 23976; Al, Compound CID: 5359268; Cd, Compound CID: 23973; Pb, Compound CID: 5352425). 5UJB showed binding affinity with the ligands of the four heavy metals, Cr, Al, Cd, and Pb, as follows: −7.62 (ligand position: 9/30), −8.68 (ligand position: 3/100), −8.69 (ligand position: 3/100), and −8.68 (ligand position: 3/80), respectively (Table 7 and Figure 6), while 6M5D showed binding affinity with the ligands of the four heavy metals, Cr, Al, Cd, and Pb, as follows: −8.72 (ligand position: 3/100), −8.68 (ligand position: 3/100), −8.68 (ligand position: 3/100), and −8.62 (ligand position: 3/90), respectively (Table 7 and Figure 6A–D).

Moreover, 6HSC showed binding affinity with the four heavy metals, Cr, Al, Cd, and Pb, as follows: −8.67 (ligand position: 3/100), −8.71 (ligand position: 3/100), −8.68 (ligand position: 3/100), and −8.71 (ligand position: 3/80), respectively (Table 7 and Figure 6), while 2ESG showed binding affinity with the ligands of the four heavy metals, Cr, Al, Cd, and Pb, as follows: −8.67 (ligand position: 3/100), −8.46 (ligand position: 3/90), −8.71 (ligand position: 3/100), and −8.47 (ligand position: 3/100), respectively (Table 7 and Figure 6). Furthermore, 6YG9 showed binding affinity with the ligands of the four heavy metals, Cr, Al, Cd, and Pb, as follows: −8.72 (ligand position: 3/100), −8.7 (ligand position: 3/80), −8.64 (ligand position: 3/100), and −8.71 (ligand position: 3/80), respectively (Table 7 and Figure 6A–D).

The mono amino oxidase (MAO) enzyme protein itself (PDB: 2BK3, 2BXS, and 7DJU) and its target receptor (PDB: 7EL7 and 2VRM) (Table 7, Figure 6) showed the binding energy with the four heavy metals (Cr, Compound CID: 23976; Al, Compound CID: 5359268; Cd, Compound CID: 23973; Pb, Compound CID: 5352425), while 2BK3 showed binding affinity with the ligands of the four heavy metals, Cr, Al, Cd, and Pb, as follows: −8.42 (ligand position: 3/100) −8.7 (ligand position: 3/100), −8.72 (ligand position: 3/100), and −8.71(ligand position: 3/100), respectively (Table 7 and Figure 6). Otherwise, 2BXS showed binding affinity with the ligands of the four heavy metals, Cr, Al, Cd, and Pb, as follows: −8.72 (ligand position: 3/100), −8.63 (ligand position: 3/100), −8.7 (ligand position: 3/100), and −8.41 (ligand position: 3/100), respectively (Table 7 and Figure 6), and 7DJU reflected the binding affinity of the ligands of the four heavy metals, Cr, Al, Cd, and Pb, as follow: −8.71 (ligand position: 3/100), −8.47 (ligand position: 3/100), −8.41 (ligand position: 3/80), and −8.45 (ligand position: 3/100), respectively (Table 7 and Figure 6). Furthermore, 7EL7 reflected the binding affinity of the ligands of the four heavy metals, Cr, Al, Cd, and Pb, as follows: −8.65 (ligand position: 3/70), −8.68 (ligand position: 3/100), −8.68 (ligand position: 3/90), and −7.23 (ligand position: 10/20), respectively (Table 7 and Figure 6). Moreover, 2VRM also reflected the binding affinity of the ligands of the four heavy metals, Cr, Al, Cd, and Pb, as follows: −7.6 (ligand position: 9/50), −8.71 (ligand position: 3/30), −8.7 (ligand position: 3/30), and −8.63 (ligand position: 3/70), respectively (Table 7 and Figure 6A–D).

The adrenaline hormone itself (PDB: 2HKK and 7BTS) and its target receptor (PDB: 2RH1) (Table 7 and Figure 6) showed binding energy with the four heavy metals (Cr, Compound CID: 23976; Al, Compound CID: 5359268; Cd, Compound CID: 23973; Pb, Compound CID: 5352425). 2HKK reflected the binding affinity of the ligands of the four heavy metals, Cr, Al, Cd, and Pb, as follows: −8.68 (ligand position: 3/100), −8.42 (ligand position: 3/100), −8.64 (ligand position: 3/50), and −8.67 (ligand position: 3/100), respectively (Table 7 and Figure 6), while 7BTS reflected the binding affinity of the ligands of the four heavy metals, Cr, Al, Cd, and Pb, as follows: −8.68 (ligand position: 3/100), −8.56 (ligand position: 3/100), −8.61 (ligand position: 3/100), and −8.66 (ligand position: 3/100), respectively (Table 7 and Figure 6). In addition, 2RH1 reflected the binding affinity of the ligands of the four heavy metals, Cr, Al, Cd, and Pb, as follows: −8.68 (ligand position: 3/100), −8.67 (ligand position: 3/100), −8.69 (ligand position: 3/100), and −8.7 (ligand position: 3/100), respectively (Table 7 and Figure 6A–D).

The cortisol hormone itself (PDB: 2VDX) and its target receptor (PDB: 2VDY and 4P6X) (Table 7 and Figure 6) showed binding energy with the four heavy metals (Cr, Compound CID: 23976; Al, Compound CID: 5359268; Cd, Compound CID: 23973; Pb: Compound CID: 5352425). 2VDX reflected the binding affinity of the ligands of the four heavy metals, Cr, Al, Cd, and Pb, as follows: −8.66 (ligand position: 3/100), −8.61 (ligand position: 3/70), −8.64 (ligand position: 3/100), and −8.46 (ligand position: 3/100), respectively (Table 7 and Figure 6). Moreover, 2VDY showed binding affinity with the ligands of the four heavy metals, Cr, Al, Cd, and Pb, as follows: −8.7 (ligand position: 3/100), −8.73 (ligand position: 3/100), −8.74 (ligand position: 3/100), and −8.65 (ligand position: 3/100), respectively (Table 7 and Figure 6). 4P6X also reflected the binding affinity of the ligands of the four heavy metals, Cr, Al, Cd, and Pb, as follows: −8.71 (ligand position: 3/90), −8.72 (ligand position: 3/100), −8.67 (ligand position: 3/100), and −8.72 (ligand position: 3/100), respectively.

TNF-α cytokinin itself (PDB: 6RMJ) and its target receptor (PDB: 5TLJ and 6PE7) (Table 7 and Figure 6) showed binding energy with the four heavy metals (Cr, Compound CID: 23976; Al, Compound CID: 5359268; Cd, Compound CID, 23973; Pb, Compound CID: 5352425). 6RMJ reflected the binding affinity of the ligands of the four heavy metals, Cr, Al, Cd, and Pb, as follows: −8.47 (ligand position: 3/100), −8.64 (ligand position: 3/100), −8.65 (ligand position: 3/100), and −8.7 (ligand position: 3/100), respectively (Table 7 and Figure 6). In addition, 5TLJ reflected the binding affinity of the ligands of the four heavy metals, Cr, Al, Cd, and Pb, as follows: −8.73 (ligand position: 3/100), −8.67 (ligand position: 3/100), −8.72 (ligand position: 3/100), and −8.7 (ligand position: 3/80), respectively (Table 7 and Figure 6). Moreover, 6PE7 also reflected the binding affinity of the ligands of the four heavy metals, Cr, Al, Cd, and Pb, as follows: −8.71 (ligand position: 3/100), −8.66 (ligand position: 3/100), −8.71 (ligand position: 3/100), and −8.71 (ligand position: 3/60), respectively (Table 7 and Figure 6A–D).

The IL-1𝛽 target receptor (PDB: 4GAF and 4GAI) (Table 7 and Figure 6) showed the binding energy with the four heavy metals (Cr, Compound CID: 23976; Al, Compound CID: 5359268; Cd, Compound CID, compound; Pb, Compound CID: 5352425). 4GAF reflected binding affinity with the ligands of the four heavy metals, Cr, Al, Cd, and Pb, as follows: −8.69 (ligand position: 3/80), −8.7 (ligand position: 3/100), −8.72 (ligand position: 3/90), and −8.67 (ligand position: 3/90), respectively (Table 7 and Figure 6). Moreover, 4GAI reflected the binding affinity of the ligands of the four heavy metals, Cr, Al, Cd, and Pb, as follows: −8.67 (ligand position: 3/100), −8.51 (ligand position: 3/100), −8.63 (ligand position: 3/100), and −8.74 (ligand position: 3/70), respectively (Table 7 and Figure 6A–D).

The COX-2 enzyme protein itself (PDB: 5JW1) and its target receptor (PDB: 4E1G and 3TZI) (Table 7 and Figure 6) showed binding energy with the four heavy metals (Cr, Compound CID: 23976; Al, Compound CID: 5359268; Cd, Compound CID, compound; Pb, Compound CID: 5352425). 5JW1 showed a binding affinity with the ligands of the four heavy metals, Cr, Al, Cd, and Pb, as follows: −8.66 (ligand position: 3/100), −8.66 (ligand position: 3/80), −8.69 (ligand position: 3/100), and −8.6 (ligand position: 3/100), respectively (Table 7 and Figure 6). Moreover, 4E1G showed binding affinity with the ligands of the four heavy metals, Cr, Al, Cd, and Pb, as follows: −8.68 (ligand position: 3/90), −8.71 (ligand position: 3/100), −8.62 (ligand position: 3/100), and −8.72 (ligand position: 3/100), respectively, where 3TZI reflected the binding affinity of the ligands of the four heavy metals, Cr, Al, Cd, and Pb, as follows: −8.68 (ligand position: 3/100), −8.68 (ligand position: 3/100), −8.66 (ligand position: 3/100), and −8.62 (ligand position: 3/100), respectively.

The LOX-1enzyme target receptor (PDB: 1YPO and 1YPU) (Table 7 and Figure 6) showed binding energy with the four heavy metals (Cr, Compound CID: 23976; Al, Compound CID: 5359268; Cd, Compound CID, compound; Pb, Compound CID: 5352425). 1YPO reflected the binding affinity of the ligands of the four heavy metals, Cr, Al, Cd, and Pb, as follows: −8.73 (ligand position: 3/100), −8.71 (ligand position: 3/100), −8.72 (ligand position: 3/100), and −8.71 (ligand position: 3/100), respectively (Table 7 and Figure 6). Moreover, 1YPU also showed binding affinity with the ligands of the four heavy metals, Cr, Al, Cd, and Pb, as follows: −8.74 (ligand position: 3/100), −8.7 (ligand position: 3/100), −8.64 (ligand position: 3/60), and −8.73 (ligand position: 3/100), respectively (Table 7 and Figure 6A–D).

Table 7 reflects the binding affinity of the four ligands for all of the investigated genes, hormones, enzymes, and protein receptors. These docking studies showed that all ligands had the highest binding affinity starting from −6.9 to approximately −8.74 kcal/mole. Moreover, Cr reflected the highest binding affinity equal to −8.74 kcal/mole with 6VTH, 1L2J, and 1YPU, separately. In addition, Cr reflected the lowest binding affinity equal to −6.9 kcal/mole with 2LY4. Al reflects the highest binding affinity equal to 8.73 kcal/mole with 6EC0 and 2VDY, separately. Al reflected the lowest binding affinity equal to −6.9 kcal/mole with 5PAH. Moreover, Cd reflected the highest binding affinity equal to −8.74 kcal/mole with 3PBL and 2VDY, separately. Cd reflected the lowest binding affinity equal to 6.9 kcal/mole with 2EP4. Furthermore, Pb reflected the highest binding affinity equal to −8.74 kcal/mole with 4GAI. Also, it reflected the lowest binding affinity equal to 7.23 kcal/mole with 7EL7, as shown in Table 7. The ligands with the best possible affinity to the genes, hormones, enzymes, and protein receptors are outlined in Table 7, while the comparison of the highest binding energy is provided in Figure 6.

The prediction of the best binding energy (kcal\mol) of the ligand position with the target protein, hormone, gene, and their receptors was conducted in the docking investigation.

## 3. Discussion

Lifestyle is always an essential indicator of the incidence of health issues depending on age, sex, genetic disposition, and other environmental factors [1,6,18,19,20]. Lifestyles, including the daily use of food, beverage, and cosmetic products, are considered an essential concern in modern functional medicine and the initiation of health problems. So, daily exposure to preservatives, pesticides, and other toxic ingredients, such as heavy metals, leads to their accumulation in many organs and may affect the chronological and vascular age of the human body [1,6,18,19,20,21].

The current work reports the screening of the lifestyle choices of female volunteers according to questionnaire outcomes that involved the daily use of food, beverage, and cosmetic products. We chose the samples in order to screen for the most toxic heavy metals and to assess their contamination levels and increase the awareness among female university students. Moreover, we screened for the same selected heavy metals in female university students’ hair, nails, and serum to determine the probability of the predisposition of the accumulation and toxicity of heavy metals in those females depending on their lifestyle choices. We chose the currently studied heavy metals based on their accumulation and toxicity and their potential results in the studied sample based on the recent literature [1,14,22] that agreed with the current results. The results of the questionnaire in the current study showed that 71 percent of the female volunteers in group 1 used the studied cosmetics, food, and beverages that contained a high level of contamination of the selected heavy metals, which was reflected in the elevated concentrations of Cr, Al, Cd, and Pb in the three different biological samples (serums, nails, and hair) that were greater than that of the 29 percent of female volunteers in group 2 who did not use the studied cosmetics, food, and beverages sample (Table 2 and Figure 1, Figure 2, Figure 3, Figure 4 and Figure 5). The current results report the presence of heavy metals, Cr, Al, Cd, and Pb, in the selected cosmetics and food products, and the most elevated element was Al in cosmetics, and the most elevated metals in the food samples were Pb followed by Al; otherwise, the beverage outcomes indicate that the most elevated heavy metals were Cr, followed by Al and then Pb, only in sample 4 (Figure 1, Figure 2, Figure 3 and Figure 4).

As a consequence of the results of daily lifestyles in the current results, the concentration of Cr, Al, Cd, and Pb in all of the different biological samples (serums, nails, and hair) in group 1 was higher than in group 2, where the highest concentration of heavy metals, in descending order, was in the nails, hair, and then serum. In addition, the most elevated heavy metal was Cr in all of the studied biological samples. However, there was an elevation that was greater than the references in group 2 that were not used in the studied samples; this may be that they used other food or daily products that affected the level of the studied heavy metals, and this agrees with previous publications that report food contamination with heavy metals in other selected foods, fresh fruits, vegetables, meat, and fish from local Saudi Arabian market [15,23,24,25,26] or other cosmetics products where heavy metals are widely diffused in pigmented makeup products [14].

The results of the metals in the studied biological samples showed a large diversity of distributions, as evidenced by the high standard deviation and the unsound increases that were greater than the normal range. The reason for this is that the concentration of elements in the biological samples depends on several factors, e.g., the content of metals in hair, nails, and serum may be influenced by age, geographical location, diet, gender, race, number of elements in other used foods, beverages, drinking water, cosmetics, seasonal changes in diet, exposure to different environmental factors, etc. [10,12,27]. Most of these factors also determine the mineral content of nails. The use of hair and nails, unlike blood and urine, reflects the concentration of elements in the organism over several months and provides information that allows for the evaluation of chronic exposure, physiological condition, nutritional status, and body load [11]. The current results for group 1 and group 2 may be due to the daily use of other cosmetics, foods, or drinks, where previous research has indicated various kinds of food contamination and toxicity, including manufactured or fresh products, as mentioned above [23,24,25,26].

Our results also show a significant elevation of cholinesterase activity in the serum of group 1, greater than that of group 2 (Table 2), which agrees with the docking analysis of the current investigation as the effect of the binding affinity of the studied heavy metals with the protein itself and its selected protein receptors of cholinesterase. A previous publication agrees with the current results and indicates that there is a positive relationship between heavy metals and cholinesterase enzyme activity. Otherwise, there is a report that cholinesterase is a bioindicator of heavy metal toxicity [28]. Another publication reported that some heavy metals have significant negative correlations and other positive signs of cholinesterase enzyme activity [29]. In addition to significant negative variation in the results of the CBC in group 1 vs. group 2 compared with the reference ranges (Table 3); this variation, hence, is a result of the elevated levels of heavy metals in the blood, which is a sign of potential health issues. Previous publications have reported that there are relationships between anemia and blood levels of heavy metals in children [30,31].

Heavy metals, including Cr, Al, Cd, and Pb, which were selected for study in this paper, interact with the human body through many proteins, genes, and DNA by losing one or many electrons and manufactured metals cations, which have many affinity sites to various macromolecules, and many tissues and body organs are also affected [6,17]. The current results for groups 1 and 2 (Table 2 and Table 3) may also be due to the daily use of other cosmetics, food, or drinks, and previous research has indicated various kinds of food contamination and toxicity, including those that are manufactured or fresh [14,15,23,25,26,32,33,34,35,36,37]. Hence, the human body’s direct exposure to heavy metals from food, beverages, cosmetics, soil, air, tobacco, dust, and skin can affect internal biological systems depending upon the dose, leading to acute early effective and chronic accumulation, consequently, leading to various macromolecule infection and health issues, including deadly neurological, brain, liver, kidney, bone, skin, and other fetal diseases [1,38].

From this point of view, we studied molecule docking for the predisposition of some proteins that may interact with heavy metals and recorded their highest binding affinity (kcal/mole) with the target (protein, hormone, gene, and their receptors) and the best-enlarged angle for the ligand with the select molecules, as listed in Figure 6 and Table 7. We identified heavy metal interactions with proteins in the lab, which required several special techniques, such as absorption spectroscopy, nuclear magnetic resonance, mobility shift assays spectroscopy, metal-affinity column chromatography, and gel electrophoresis [39].

In the current work, we first conducted AutoDock analyses to predict the interactions between the studied heavy metals and the predisposed toxicity to agree with the ethics, as a replacement, reduction, and refinement, in scientific testing research prior to attempting human and animal investigations. Molecular docking is hypothetical bioinformatics, which reflects the collaboration among molecules (such as DNA, genes, ligands, proteins, and receptors) and expects their binding modes and affinity over a computer bioinformatics system platform. Molecular docking works as satisfactory devices in pharmacology and medicinal study, such as structure-built reasonable drug plans, acknowledged by researchers to a scientific and ethical extent. Numerous primary investigations have been conducted correlating biomolecular interactions for food matrices, toxicity, antibiotics, heavy metals, new drugs, and phytochemistry [40] in recent years. The extraordinary advantages of molecular dockings, such as predicting experiments to help achieve scientific and medical ethics to reduce the number of experimental trials and pass down materials, as well as in elasticity docking for interesting cumulative considerations has potential. By docking complex binding energy, the lowest energy is represented by negative values, and this designates the highest binding affinity as high energy that produces unbalanced conformations that support pharmacological trials and interactions that activates or inhibits the activity of DNA, genes, ligands, proteins, and receptors of any studied molecule, element, phytochemistry, or new drug [41,42,43,44,45,46,47,48,49].

The effect of the current heavy metal concentrations’ interactions in the body beginning with cholinesterase, P53, dopamine, metallothionein, estrogen, keratin, protein kinase enzyme, beta-amyloid, ATPase, albumin, MAO, adrenaline, cortisol, TNF-α, IL-1β, COX-2, and LOX-1 are reported in Table 7 and Appendix A and Figure 6, which agree with previous publications [6,50].

Heavy metals may have a neurological effect, inactivate regulatory molecules, such as P53, and release free radicals that affect detoxification system proteins [51]. Previous publications report a positive relationship between the percent of DNA methylation in breast tumor tissue as a result of heavy metal accumulation and tumor tissues [52], which may be affected by both the quality of the activities of p53–protein and p53–DNA [53].

As a consequence, the p53-Estrogen receptor may also be affected, as previous research has reported [54], which agrees with the current results of the docking analysis, where there was a high binding affinity (kcal/mole) of heavy metals with the studied molecules (P53, estrogen, and dopamine). Moreover, the current results agree with a previous publication that reported relationships between the percent of heavy metals in cancer cells and the progressive sensitivity of breast tumors, preventing the cure and treatment of DNA repair and decrease in the detoxification effect of the body’s enzyme and proteins system [17,52]

Furthermore, a previous publication reported that metallothionein was a metal-binding protein expression in tumor tissues and presented a mutation of P53 in tumor tissue [55], which predicted a relationship between the binding affinity of heavy metals with both metallothionein and P53. Metallothionein is reported as fighting stress [6] and it may have a relationship with mechanisms of stress hormones, such as cortisol, adrenaline, and metallothionein in binding to heavy metals that agree with the current study; hence, we report a high binding affinity (kcal/mole) of heavy metals with metallothionein, cortisol, adrenaline, P53, and other proteins (Figure 6 and Table 7).

Heavy metals enter the mucous membrane in the skin, and the interactions depend on protein–protein interactions that include enzymes of detoxification systems such as GSH [6,56,57]. In addition, heavy metals can reach the blood system through the direct application of cosmetic samples onto the skin and into hair follicles and then sweat pores, entering blood capillaries to the blood and reaching all organs, and bioaccumulating according to daily use, lifestyle, and dose exposure to heavy metals [6,56,57]. A previous publication reported that Pb can interact and affect GSH and other antioxidant enzyme proteins [56], which agrees with the current results.

The current results reflect an interaction with high binding affinity with albumin, ATPase, and protein kinase that agrees with previous publications that report that heavy metals, such as cadmium and lead, can also combat and challenge other essential metals such as calcium to bind many proteins, showing their toxicity through interactions that inhibit calcium-ATPase and protein kinase enzyme activity because of their biochemical similarities to calcium [6,58,59].

Furthermore, IL-1β and TNF-α can express acute inflammation, infection, and cancer [6]. The current results report a high binding affinity between IL-1β and TNF-α and the studied heavy metals, which agrees with previous publications [6,17]. The interaction of heavy metals with detoxification systems affects the balance of ROS in the human body and affects the level of inflammatory markers and inflammatory enzymes, such as interleukin−1β (IL-1β), phosphorylated protein kinase B (PKB), and cyclooxygenase 2 (COX-2). Also, previous publications report that there are interactions among heavy metals and interleukin−1β (IL-1β), phosphorylated protein kinase B (PKB), and cyclooxygenase 2 (COX-2), which agrees with the current results. Docking analysis results report high binding affinity energy among COX-2, LOX, IL-1β, TNF-α, and protein kinase enzyme [17,60,61].

Previous publications report a relationship between heavy metals and beta-amyloid, as well as Alzheimer’s disease [62,63,64]. This may be due to the effect of heavy metals on homeostasis through the formation, accumulation, deposition, and proteolytic scavenger process and the activity of enzymes that are responsible for removing beta-amyloid in homeostasis balances. Heavy metals may also be enhancing, surrounding, and protecting the plaque’s beta-amyloid protein in Alzheimer’s disease in the brain. This agrees with the current results of the selected heavy docking analysis, which reflects high binding energy with the beta-amyloid [62,63,64].

Keratin is a cysteine that contains protein and can interact with heavy metals [65,66]; hence, it may be the first way to interact with heavy metals and enter the human body, which agrees with current results and explains the high binding energy with keratin in the selected heavy metals in the docking analysis. Previous publications report that heavy metals can accumulate in the human body as a result of lifestyle exposure and entrance via the blood into the brain; they also find a relationship between measured heavy metals in humans and mono amino oxidase and various neurotransmitters mediators, relationships, and interactions that are reflected in the current docking analysis results, where there was high binding energy with the selected heavy metals in this study and mono amino oxidase with target receptors and enzymes [67,68,69,70,71,72,73,74,75]. Our result also showed binding energy with adrenaline target receptors and molecules with Cr, Cd, Pb, and Al, which explains the results of the effect of heavy metals and neurological disturbances of many previous findings [6,76]. Furthermore, previous publications report that heavy metals have abilities to accumulate in the adrenal gland adrenal capsular, adrenal decapsular, and Leydig cells of the testis and other essential organs in the body leading to increasing body stress levels [76].

The relationship between stress and heavy metals was confirmed in a previous publication. The investigation reported that there was a positive relationship between the measured heavy metals and cortisol hormone levels in plasma [77,78], which agrees with the current results, as this position also represents the interaction with cortisol and receptor proteins; moreover, the current results report the binding energy among Cr, Cd, Pb, and Al and cortisol, which may also lead to increases in stress and inflammatory markers and enzymes, as mentioned in our previous publications [79,80,81,82]. Also, other previous publication report relationships among heavy metals and COX-2 and other inflammatory markers, such as LOX and increased oxidative stress [61,83,84,85,86], and this agrees with our current results, where there is a high binding affinity between COX-2 and LOX.

In summary, natural ecosystem environmental factors, inheritance genomes, and human lifestyles can interact with the human body, including food, beverage, and cosmetic additives and metals. Heavy metals are considered toxic health issues and can enter the human body through environmental factors, such as soil, volcanics, and wastewater that are diffused from various manufacturing industries. The World Health Organization has approved the maximum and minimum concentrations of heavy metals in the food, beverage, cigarette, cosmetic, and other industries; however, there are many health problems caused by tiny particles of heavy metals that lead to skin inflammation, tumors, digestion, nausea, inflammation, and toxicity, which agrees with the current results. We conclude that the contamination of the selected samples in this study, as well as the volunteers that used the selected samples in group 1 showed elevated concentrations of heavy metals, cholinesterase, and CBC-negative varieties more than those that did not use the samples in group 2.

In summary, heavy metals compete at genes, proteins, and enzyme sites and receptors, consequently affecting their function quality, activity, and scavenger degradation process, leading to various health issues, which agrees with the current results on the potential ligands of cholinesterase, P53, dopamine, metallothionein, estrogen, keratin, protein kinase enzyme, beta-amyloid, ATPase, albumin, MAO, adrenaline, cortisol, TNF-α, IL-1β, COX-2, and LOX-1, as well as confirms the biochemical analysis. Future studies should be emphasized at all levels and solve all of these problems related to heavy metal contamination and related health issues in Saudi Arabia in progress.

## 4. Materials and Methods

### 4.1. Study Population

The current study included 100 females with a body mass index (BMI) of 32.58 ± 3.5. Volunteers with a history of any disease were excluded from the study. Obese females, pregnant females, smokers, and children were also excluded. The selected female volunteers were 18–45 years of age. The appropriate institution involved in the current work approved and accepted our protocol, which satisfied the Declaration of Helsinki as revised in 2013. We collected consent from all of the female volunteers. We grouped the female volunteers into two groups according to the questionnaire results: Group 1 (*n* = 71), where they used cosmetic, food, and drink samples; Group 2 (*n* = 29), where they did not use cosmetics, food, and drink samples that were included in this investigation [1,21].

### 4.2. Biological Samples Collection

Before we collected the hair and nail samples, we asked the female volunteers to clean and wash their hands and nails with medical soap that was metal free and to dry and remove external dirt and contamination using sterile tissue paper, which may avoid external contamination of heavy metal. We collected blood samples after fasting for 9 h. We prepared the serum for the heavy metals and biochemical analyses as per a previous publication [82].

### 4.3. Non-Biological Samples Collection

We chose not to reveal the names of the brands used in this study to avoid any conflict issues. Hence, canned cheese, canned meat, canned fish, synthetic juices, cosmetic products, and water bottles were selected according to the results of the applied questionnaire.

We collected canned cheese, canned meat, canned fish, synthetic juices, cosmetic products, and water bottles from local marks in Saudi Arabia. The selection procedure was performed according to the questionnaire filled out by the volunteers in this study. We applied our previous sample preparation method and analysis for inductively coupled plasma mass spectrometry (ICP–MS; 7500 cx, Agilent Technologies, Santa Clara, CA 95051, USA) [1,21].

### 4.4. The Sample Preparation for Inductively Coupled Plasma Mass Spectrometry ICP-MS and Heavy Metals Analyses

All samples, except water and biological serum samples, were soaked for 4 h in 1 M HNO_3_ by (1:10) (*w*/*v*) prior to performing the microwave digestion method. After digesting all of the samples (canned cheese, canned meat, canned fish, synthetic juices, cosmetic products, and biological samples) with nitric acid (1:5 *v*/*v*) in a microwave digestion system, Ethos 1 (Milestone, Fremont, CA 94539, USA), according to the manufacturer’s instructions. We diluted the samples, including the water samples (1:5 *v*/*v*), with nitric acid to determine the concentrations of lead (Pb), cadmium (Cd), aluminum (Al), and chromium (Cr). We repped our recent method for inductively coupled plasma mass spectrometry (ICP MS; 7500 cx, Agilent Technologies, Santa Clara, CA 95051, USA) [1,21].

### 4.5. Biochemical Assays

We analyzed the level of acetylcholinesterase (AChE) in the serum samples according to our previous publication [86,87,88,89,90,91,92]. We determined the activity of AChE with the kinetic colorimetric method according to Deutsche (1992) with the ARCHITECT systems [87]. We used butyrythiocholine as the specific substrate for cholinesterase (CHE), which catalyzes the degradation of the substrate to butyrate and thiocholine. Thiocholine reacted and transformed hexacyanoferrate (III) to hexacyanoferrate (II). We recorded the absorbance reading directly proportional to the CHE activity in the sample using the ARCHITECT systems at 37 °C [87].

We analyzed the complete blood count (CBC) in blood using the Sysmex K−4500 Haematology Analyzer TOA SYSMEX, Kobe, Japan [88,89].

### 4.6. AutoDock Analysis

#### 4.6.1. Preparation of the Modeled Receptor from PDB: Heavy Metals as Ligand Components for Docking

The AutoDock analysis section used potential ligands and receptors to provide more considerable proof of their therapeutic potential [44]. We obtained the enzyme protein or their receptors from the PDB portal site (https://www.rcsb.org/) (accessed on 1 January 2023) (Appendix A), and we obtained the ligand from the Cr (Compound CID: 23976), Al (Compound CID: 5359268), Cd (Compound CID: 23973), and Pb (Compound CID: 5352425) from PubChem portal site [44].

#### 4.6.2. AutoDock Analysis

AutoDock Vina software includes devices for optimizing proteins and ligands, such as allowing for atomic charges to make proteins more polar [44]. We used the protein–ligand docking method from the RPBS WEB PORTAL. The protein–ligand method on this website (https://mobyle.rpbs.univ-paris-diderot.fr/cgi-bin/portal.py#welcome) (accessed on 1 January 2023) is based on MTiOpenScreen via AutoDock Vina. We used this docking method based on the Vina software basic procedure [92,93,94,95].

The protein–ligand docking procedure for the molecular modeling of the proteins and ligands was used to charge and rotate the bonds. The reflection of the energy influenced the desolations through the ligand binding on the protein and the previous identification of the grid maps on the protein surface for the synergy ligands using the auto grid. The outstanding apparatuses increased the rapidity and precision. RPBS WEB PORTAL, based on AutoDock Vina docking, was used with select scoring dimensions, actual optimization, and multithreading of molecular docking [44,93,95,96,97].

### 4.7. Statistical Analyses

We analyzed the data using IBM SPSS software package version 20.0. (Armonk, NY, USA: IBM Corp). Continuous data were tested for normality with the Kolmogorov–Smirnov and Shapiro–Wilk tests. Quantitative outcomes are described as ranges (minimum and maximum), means, standard deviations, and medians for normally distributed quantitative variables. The Student’s *t*-test was utilized to correspond to the two groups. Furthermore, for non-normally distributed quantitative variables, the Mann–Whitney U test was used to compare the two groups. The significance of the obtained results was judged at the 5% level [1].

## 5. Conclusions

We conclude that there was heavy metal contamination in the selected samples, including Cr, Al, Cd, and Pb. Comparisons of the results from human volunteers who used the selected samples in their daily life with volunteers who did not use them revealed contamination with heavy metals. Volunteers that used the selected samples showed elevated Cr, Al, Cd, and Pb levels greater than the volunteers that did not use the selected samples of the study. The outcomes of the docking analysis confirmed the biochemical analysis. Further studies involving more screening and the biochemical potential health consequences of heavy metals are in progress.

## Figures and Tables

**Figure 1 molecules-28-05582-f001:**
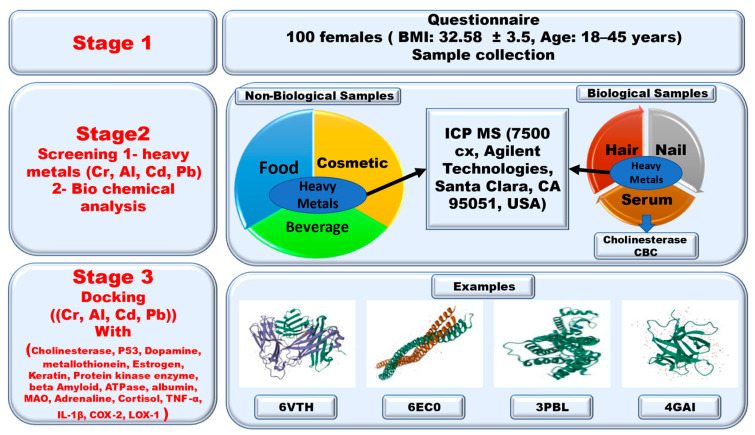
Experimental design.

**Figure 2 molecules-28-05582-f002:**
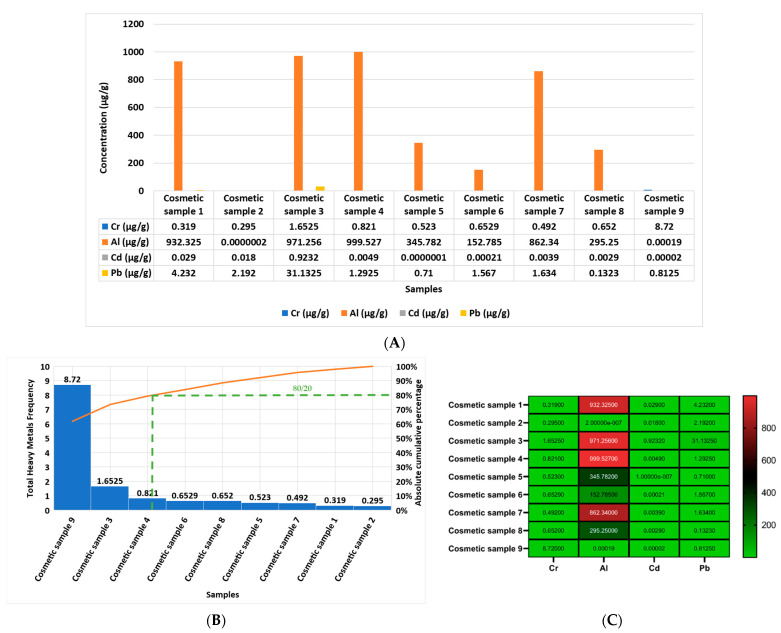
Comparison of the heavy metal concentrations of Cr, Al, Cd, and Pb in each cosmetic product analyzed separately in μg/g in the current study: (**A**) cosmetics samples—sample 1 = hair cream, sample 2 = skin cream, sample 3 = shampoo, sample 4 = shower gel, sample 5 = body lotion, sample 6 = body lotion, sample 7 = handwash, sample 8 = skin cream, and sample 9 = toothpaste; (**B**) Pareto curve of cosmetic samples 1 to 9 according to the cumulative frequency of the rule (80/20) of the Pareto curve in descending order (orange line: represent the cumulative effect of studied samples in descending order), green color represents the slope: where according to Pareto principle or the law of the vital few and trivial many) states that, for many heavy metals concentrations, roughly 80% of the toxicity effects come from 20% of the samples of the left of the slope, this means that cosmetic samples 9 and 3 are the most contaminated with the heavy metals; (**C**) heat map of the concentrations of Cr, Al, Cd, and Pb in each cosmetic (1 to 9) product separately.

**Figure 3 molecules-28-05582-f003:**
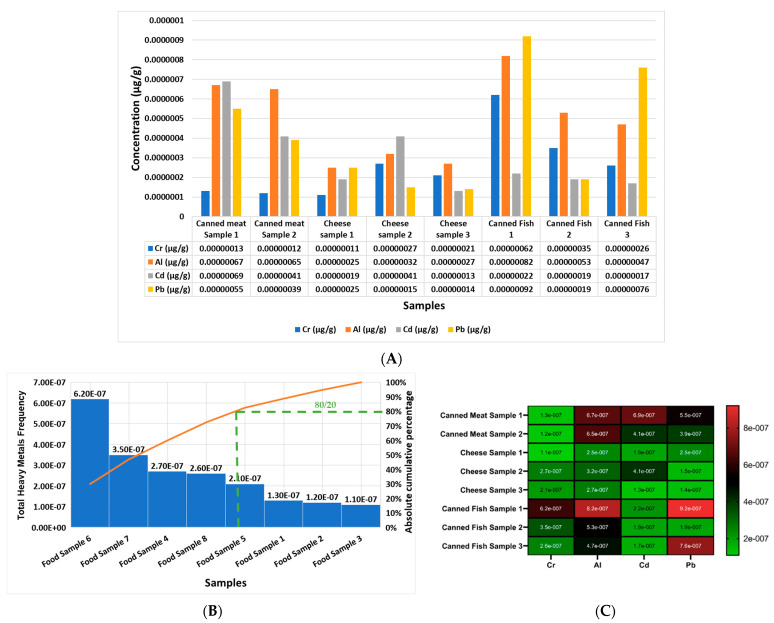
Comparison of the heavy metal concentrations of Cr, Al, Cd, and Pb in each food product analyzed separately in μg/g in the current study in μg/g: (**A**) food samples—sample 1 = canned meat, sample 2 = canned meat, sample 3 = cheese sample 1, sample 4 = cheese sample 2, sample 5 = cheese sample 3, sample 6 = canned fish 1, sample 7 = canned fish 2, and sample 8 = canned fish 3; (**B**) Pareto curve of food samples 1 to 8 according to the cumulative frequency of the rule (80/20) of the Pareto curve in descending order, orange line: represent the cumulative effect of studied samples in descending order), green color represents the slope: where, according to Pareto principle or the law of the vital few and trivial many) states that, for many heavy metals concentrations, roughly 80% of the toxicity effects come from 20% of the samples in the left of the slope, this means that food samples 6, 7, 4, and 8 are the most contaminated with the heavy metals; (**C**) heat map of the concentrations of Cr, Al, Cd, and Pb in each food (1 to 8) product separately.

**Figure 4 molecules-28-05582-f004:**
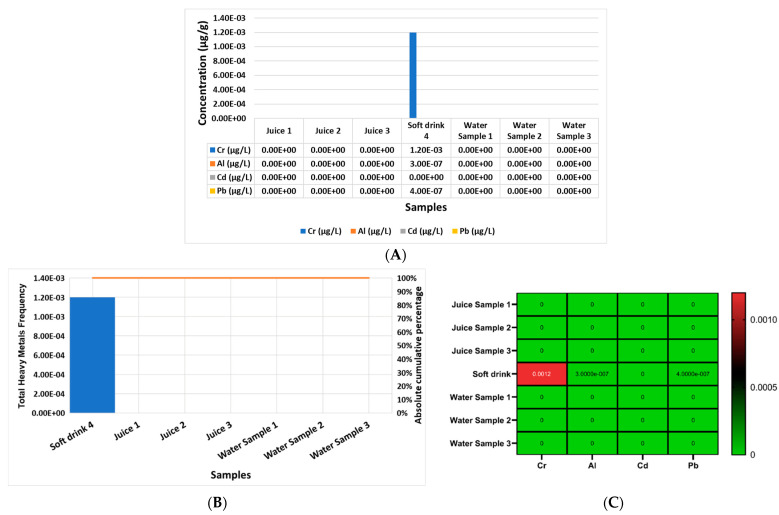
Comparison of the heavy metal concentrations of Cr, Al, Cd, and Pb in each beverage and water product analyzed separately in the current study, from sample one to sample seven: (**A**) beverage and water samples—sample 1 = manufacturer juice 1 (mango), sample 2 = manufacturer juice 2 (orange), sample 3 = manufacturer juice 3 (guava), sample 4 = manufacturer soft drink 4, sample 5 = water sample 1 brand, sample 6 = water sample 2 brand, and sample 7 = water sample 3 brand; (**B**) Pareto curve of each beverage and water samples from 1 to 7 according to the cumulative frequency of the rule (80/20) of the Pareto curve in descending order; (**C**) heat map of the concentrations of Cr, Al, Cd, and Pb in the beverage and water (1 to 7) products separately.

**Figure 5 molecules-28-05582-f005:**
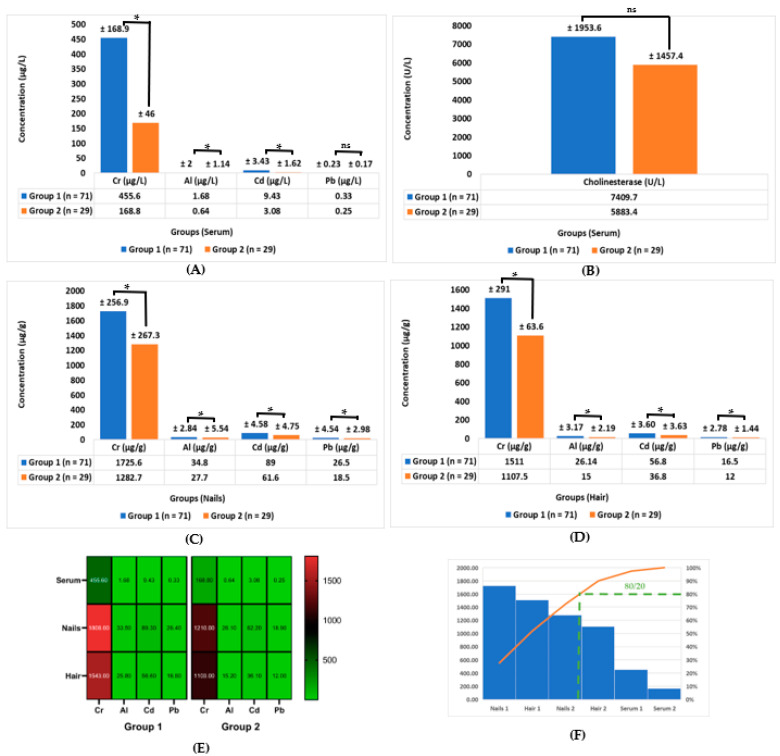
Biochemical comparison of the two groups’ investigation: (**A**) comparison of the heavy metal concentrations of Cr, Al, Cd, and Pb in serum in μg/L and (**B**) comparison of the concentration of cholinesterase in U/L in serum; (**C**) comparison of heavy metal concentrations of Cr, Al, Cd, and Pb in μg/g in nails; (**D**) comparison of heavy metal concentrations of Cr, Al, Cd, and Pb in μg/g in hair; (**E**) heat map of the concentrations of Cr, Al, Cd, and Pb in three different biological samples (serums, nails, and hair) in group 1 vs. group 2 separately; (**F**) Pareto curve of three different biological samples (serums, nails, and hair) in group 1 vs. group 2 according to the cumulative frequency of the rule (80/20) of the Pareto curve in descending order, (orange line: represent the cumulative effect of studied samples in descending order), green color represents the slope: where, according to Pareto principle or the law of the vital few and trivial many) states that, for many heavy metals concentrations, roughly 80% of the toxicity effects come from 20% of the samples in the left of the slope, this means that biological samples Nails 1, Hair 1, and Nails 2 are the most contaminated with the heavy metals. Group 1 (*n* = 71) used cosmetic, food, and drink samples according to questionnaire results that were analyzed and included in this investigation—Group 2 (*n* = 29) did not use cosmetics, food, and drink samples that were analyzed and included in this investigation according to questionnaire results. Data are given as the mean ± SD. Statistical significance was set at * *p* ≤ 0.05; the means reported with a star are significantly different; ns means non-significant.

**Figure 6 molecules-28-05582-f006:**
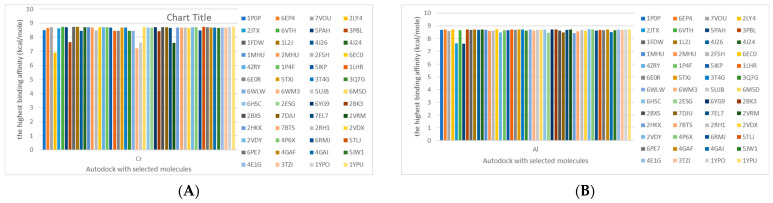
Comparison between the highest binding energy of the studied ligands and target receptor: (**A**) AutoDock of Cr with selected studied receptor; (**B**) AutoDock of Al with selected studied receptor; (**C**) AutoDock of Cd with selected studied receptor; (**D**) AutoDock of Pb with selected studied receptor.

**Table 2 molecules-28-05582-t002:** Comparison between the two studied groups according to biological samples with normal range.

	Biological Samples	Group 1 (*n* = 71)	Group 2 (*n* = 29)	Test of Sig.	*p*
Serum	Cr (μg/L) (0.1–2.1)	0 (0%)	0 (0%)	–	–
	Mean ± SD	455.6 ± 168.9	168.8 ± 46	U = 26.0 *	<0.001 *
	Median (Min.–Max.)	408.1 (128–962)	172 (101–273)		
	Al (μg/L) (0–9)	70 (98.6%)	29 (100%)	χ^2^ = 0.413	^FE^*p* = 1.000
	Mean ± SD	1.68 ± 2	0.64 ± 1.14	U = 768.0 *	0.042 *
	Median (Min.–Max.)	1.29 (0–9.73)	0.20 (0–5.16)		
	Cd (μg/L) (0–4.9)	7 (9.9%)	26 (89.7%)	χ^2^ = 59.97 *	<0.001 *
	Mean ± SD	9.43 ± 3.43	3.08 ± 1.62	U = 75.0 *	<0.001 *
	Median (Min.–Max.)	8.90 (3.40–17.2)	3.10 (0–7.20)		
	Pb (μg/L) (0–190)	71 (100%)	29 (100%)	–	–
	Mean ± SD	0.33 ± 0.23	0.25 ± 0.17	U = 791.0	0.069
	Median (Min.–Max.)	0.32 (0–0.96)	0.20 (0–0.60)		
	Cholinesterase (U/L) (4389–10,928)	66 (93%)	23 (79.3%)	χ^2^ = 3.917	^FE^*p* = 0.074
	Mean ± SD	7409.7 ± 1953.6	5883.4 ± 1457.4	U = 523.50 *	<0.001 *
	Median (Min.–Max.)	7786.6 (1628.7–10,289.5)	6044 (3938.2–9452)		
Nails	Cr (μg/g) (0.01–1.4)	0 (0%)	0 (0%)	–	–
	Mean ± SD	1725.6 ± 256.9	1282.7 ± 267.3	t = 7.733 *	<0.001 *
	Median (Min.–Max.)	1808 (1105–2015)	1210 (1003–2318)		
	Al (μg/g) (0.5–69)	71 (100%)	29 (100%)	–	–
	Mean ± SD	34.8 ± 2.84	27.7 ± 5.54	t = 6.524 *	<0.001 *
	Median (Min.–Max.)	33.5 (30.1–39.8)	26.1 (17–39.1)		
	Cd (μg/g) (0.001–0.14)	0 (0%)	0 (0%)	–	–
	Mean ± SD	89 ± 4.58	61.6 ± 4.75	t = 26.945 *	<0.001 *
	Median (Min.–Max.)	89.3 (80.9–99.3)	62.2 (50.2–69.2)		
	Pb (μg/g) (0.02–2)	0 (0%)	0 (0%)	–	–
	Mean ± SD	26.5 ± 4.54	18.5 ± 2.98	t = 8.720 *	<0.001 *
	Median (Min.–Max.)	26.4 (15.1–35.9)	18.9 (10.9–24.1)		
Hair	Cr (μg/g) (0.1–1.5)	0 (0%)	0 (0%)	–	–
	Mean ± SD	1511 ± 291	1107.5 ± 63.6	t = 11.056 *	<0.001 *
	Median (Min.–Max.)	1543 (1003–1983)	1103 (1003–1301)		
	Al (μg/g) (0.2–14.6)	0 (0%)	11 (37.9%)	χ^2^ = 30.260	^FE^*p* < 0.001 *
	Mean ± SD	26.14 ± 3.17	15 ± 2.19	t = 17.243 *	<0.001 *
	Median (Min.–Max.)	25.8 (20.1–35.1)	15.2 (10.7–18.1)		
	Cd (μg/g) (0.004–0.17)	0 (0%)	0 (0%)	–	–
	Mean ± SD	56.8 ± 3.60	36.8 ± 3.63	t = 25.151 *	<0.001 *
	Median (Min.–Max.)	56.6 (50.2–66)	36.1 (30.6–44.8)		
	Pb (μg/g) (0.1–5.2)	0 (0%)	0 (0%)	–	–
	Mean ± SD	16.5 ± 2.78	12 ± 1.44	t = 10.675 *	<0.001 *
	Median (Min.–Max.)	16.8 (10.8–26.3)	12 (10–17.3)		

SD: standard deviation; t: Student’s *t*-test; U: Mann–Whitney U test; χ^2^: Chi-square test; FE: Fisher’s exact test; *p*: *p*-value for comparing between the two studied groups. * Statistically significant at *p* ≤ 0.05.

**Table 3 molecules-28-05582-t003:** Comparison between the two studied groups according to CBC with normal range.

CBC	Group 1 (*n* = 71)	Group 2 (*n* = 29)	Test of Sig.	*p*
Hemoglobin (g/dL) (12–15)	8 (11.3%)	29 (100.0%)	χ^2^ = 69.547 *	<0.001 *
Mean ± SD	10.2 ± 1.15	12.8 ± 0.51	t = 15.543 *	<0.001 *
Median (Min.–Max.)	10.1 (8.20–13)	12.7 (12.1–13.9)		
Hematocrit (%) (37–47)	60 (84.5%)	29 (100.0%)	χ^2^ = 5.048 *	^FE^*p* = 0.031 *
Mean ± SD	38.3 ± 1.28	42.35 ± 1.29	t = 14.130 *	<0.001 *
Median (Min.–Max.)	38.4 (35.2–40.5)	42.3 (40–45.9)		
RBCs (×10^12^/L) (3.8–4.8)	22 (31.0%)	29 (100.0%)	χ^2^ = 39.243 *	<0.001 *
Mean ± SD	5.04 ± 0.46	4.11 ± 0.20	t = 13.992 *	<0.001 *
Median (Min.–Max.)	5.12 (4.12–5.85)	4.12 (3.82–4.65)		
MCV (fl) (80–100)	30 (42.3%)	29 (100.0%)	χ^2^ = 28.384 *	<0.001 *
Mean ± SD	78.9 ± 3.93	83.2 ± 1.23	t = 8.173 *	<0.001 *
Median (Min.–Max.)	79.1 (70.9–86.5)	82.9 (80.2–85.4)		
MCH (pg) (27–32)	7 (9.9%)	29 (100.0%)	χ^2^ = 72.613 *	<0.001 *
Mean ± SD	26 ± 0.60	27.98 ± 0.64	t = 14.596 *	<0.001 *
Median (Min.–Max.)	26 (25–27.5)	27.9 (27.1–29.2)		
MCHC (g/dL) (32–36)	0 (0.0%)	29 (100.0%)	χ^2^ = 100.00 *	<0.001 *
Mean ± SD	29.5 ± 0.65	33.8 ± 0.63	t = 30.260 *	<0.001 *
Median (Min.–Max.)	29.2 (28.2–30.9)	33.9 (32.5–34.9)		
RDW (%) (13–15)	4 (5.6%)	29 (100.0%)	χ^2^ = 82.928 *	<0.001 *
Mean ± SD	15.7 ± 0.63	13.77 ± 0.36	t = 19.585 *	<0.001 *
Median (Min.–Max.)	15.6 (14.2–17)	13.8 (13.1–14.6)		
Platelet (×10^9^/L) (150–400)	71 (100.0%)	29 (100.0%)	–	–
Mean ± SD	233.5 ± 23	215 ± 13.3	t = 5.034 *	<0.001 *
Median (Min.–Max.)	227 (200–292)	215.(184–241)		
MPV (fl) (7.5–12)	24 (33.8%)	29 (100.0%)	χ^2^ = 36.221	<0.001 *
Mean ± SD	7.35 ± 0.28	8.50 ± 0.38	t = 16.746 *	<0.001 *
Median (Min.–Max.)	7.25 (7.01–7.95)	8.48 (7.56–9.12)		
WBCs (×10^9^/L) (4.5–11)	71 (100.0%)	29 (100.0%)	–	–
Mean ± SD	6.40 ± 0.36	5.42 ± 0.27	t = 13.258 *	<0.001 *
Median (Min.–Max.)	6.35 (5.20–6.98)	5.40 (5.10–5.90)		
Basophils (×10^9^/L) (0.02–0.1)	71 (100.0%)	29 (100.0%)	–	–
Mean ± SD	0.05 ± 0.02	0.04 ± 0.01	U = 576.0 *	<0.001 *
Median (Min.–Max.)	0.05 (0.02–0.09)	0.04 (0.02–0.06)		
Eosinophil’s (×10^9^/L) (0.2–0.5)	71 (100.0%)	29 (100.0%)	–	–
Mean ± SD	0.32 ± 0.06	0.29 ± 0.05	t = 2.719 *	0.008 *
Median (Min.–Max.)	0.32 (0.21–0.47)	0.28 (0.24–0.41)		
Neutrophils (×10^9^/L) (2–7)	71 (100.0%)	29 (100.0%)	–	–
Mean ± SD	4.09 ± 0.69	4.12 ± 0.19	t = 0.370	0.712
Median (Min.–Max.)	3.98 (3.01–5.98)	4.15 (3.65–4.53)		
Lymphocytes (×10^9^/L) (1–3)	71 (100.0%)	29 (100.0%)	–	–
Mean ± SD	2.22 ± 0.49	2.03 ± 0.17	t = 2.866 *	0.005 *
Median (Min.–Max.)	2.38 (1.12–2.96)	2.03 (1.68–2.31)		
Monocytes (×10^9^/L) (0.2–1)	70 (98.6%)	29 (100.0%)	χ^2^ = 0.413	^FE^*p* = 1.000
Mean ± SD	0.66 ± 0.10	0.51 ± 0.03	t = 11.397 *	<0.001 *
Median (Min.–Max.)	0.68 (0.45–1.01)	0.51 (0.45–0.57)		

SD: standard deviation; t: Student’s *t*-test; U: Mann–Whitney U test; χ^2^: Chi square test; FE: Fisher’s Exact test; *p*: *p*-value comparing between the two studied groups. * Statistically significant at *p* ≤ 0.05. WBCs: white blood cells; RBCs: red blood cells; HGB: hemoglobin; HTC: hematocrit; MCV: mean corpuscular volume; MCH: mean corpuscular hemoglobin; MCHC: mean corpuscular hemoglobin concentration; RDW: red cell distribution width; MPV: mean platelet volume.

**Table 4 molecules-28-05582-t004:** Correlation between biological samples in each group.

	Biological Samples					
	Serum (μg/L) vs. Nails (μg/g)		Serum (μg/L) vs. Hair (μg/g)		Nails (μg/g) vs. Hair (μg/g)	
	r_s_	*p*	r_s_	*p*	r	*p*
Group 1 (*n* = 71)						
Cr	0.014	0.908	0.075	0.532	−0.185	0.122
Al	−0.083	0.490	−0.083	0.494	−0.017	0.886
Cd	0.279 *	0.018 *	−0.028	0.814	−0.070	0.559
Pb	−0.245 *	0.039 *	0.081	0.501	−0.038	0.751
Group 2 (*n* = 29)						
Cr	0.273	0.152	−0.133	0.492	0.142	0.463
Al	−0.070	0.717	−0.297	0.117	0.103	0.594
Cd	0.125	0.518	−0.075	0.700	0.226	0.239
Pb	0.161	0.404	0.023	0.906	0.252	0.186

r_s_: Spearman coefficient; r: Pearson coefficient. * Statistically significant at *p* ≤ 0.05.

**Table 5 molecules-28-05582-t005:** Correlation between cholinesterase (U/L) and the serum biological samples in each group.

Cholinesterase (U/L) vs.	Group 1 (*n* = 71)		Group 2 (*n* = 29)	
	r_s_	*p*	r_s_	*p*
Serum Biological Samples				
Cr (μg/L)	0.299	0.011 *	0.101	0.602
Al (μg/L)	−0.275	0.020 *	−0.370	0.048 *
Cd (μg/L)	0.118	0.328	0.250	0.191
Pb (μg/L)	0.021	0.860	0.201	0.295

r_s_: Spearman coefficient. * Statistically significant at *p* ≤ 0.05.

**Table 6 molecules-28-05582-t006:** Correlation between serum and the CBC in each group.

		Serum									
		Cr (μg/L)		Al (μg/L)		Cd (μg/L)		Pb (μg/L)		Cholinesterase (U/L)	
		Group 1 (*n* = 71)	Group 2 (*n* = 29)	Group 1 (*n* = 71)	Group 2 (*n* = 29)	Group 1 (*n* = 71)	Group 2 (*n* = 29)	Group 1 (*n* = 71)	Group 2 (*n* = 29)	Group 1 (*n* = 71)	Group 2 (*n* = 29)
Hemoglobin (g/dL)	r_s_	−0.243 *	−0.096	0.154	−0.218	0.024	0.148	−0.034	0.234	−0.092	−0.028
	*p*	0.041 *	0.622	0.198	0.255	0.840	0.444	0.779	0.222	0.445	0.884
Hematocrit (%)	r_s_	0.522 *	0.107	−0.161	−0.019	0.041	−0.116	−0.246 *	0.244	0.409 *	−0.297
	*p*	<0.001 *	0.580	0.179	0.921	0.736	0.548	0.038 *	0.203	<0.001 *	0.118
RBCs (×10^12^/L)	r_s_	−0.044	0.057	−0.084	0.006	−0.119	−0.291	0.116	0.104	−0.027	0.186
	*p*	0.715	0.767	0.487	0.975	0.321	0.126	0.337	0.590	0.823	0.334
MCV (fl)	r_s_	0.338 *	−0.083	0.026	−0.166	0.208	0.153	−0.163	0.108	0.206	0.171
	*p*	0.004 *	0.668	0.830	0.391	0.082	0.427	0.174	0.575	0.085	0.376
MCH (pg)	r_s_	0.023	−0.067	0.010	0.092	−0.109	−0.110	0.119	0.043	−0.313 *	−0.228
	*p*	0.852	0.729	0.932	0.635	0.365	0.571	0.322	0.825	0.008 *	0.234
MCHC (g/dL)	r_s_	−0.201	−0.137	0.248 *	0.324	0.072	−0.250	0.208	−0.255	−0.134	−0.483 *
	*p*	0.093	0.479	0.037 *	0.086	0.548	0.190	0.082	0.182	0.264	0.008 *
RDW (%)	r_s_	0.041	−0.201	0.019	−0.065	−0.087	−0.031	0.074	0.077	−0.160	−0.078
	*p*	0.734	0.295	0.872	0.736	0.472	0.875	0.539	0.692	0.182	0.687
Platelet (×10^9^/L)	r_s_	−0.156	−0.380 *	−0.333 *	−0.028	0.191	0.302	0.056	−0.384 *	0.033	−0.215
	*p*	0.193	0.042 *	0.005 *	0.887	0.110	0.111	0.644	0.040 *	0.782	0.264
MPV (fl)	r_s_	−0.048	−0.052	−0.183	−0.004	−0.095	0.079	0.131	−0.074	−0.053	0.116
	*p*	0.691	0.791	0.127	0.983	0.430	0.683	0.278	0.702	0.662	0.549
WBCs (×10^9^/L)	r_s_	0.092	0.100	−0.159	0.198	0.150	−0.180	0.026	−0.033	0.240 *	−0.016
	*p*	0.444	0.605	0.186	0.304	0.212	0.349	0.833	0.866	0.044 *	0.936
Basophils (×10^9^/L)	r_s_	−0.008	0.039	−0.040	0.041	0.129	0.046	0.139	0.175	0.129	0.015
	*p*	0.948	0.841	0.738	0.833	0.285	0.812	0.246	0.365	0.282	0.938
Eosinophil’s (×10^9^/L)	r_s_	−0.001	0.139	0.170	0.165	0.036	−0.371 *	−0.268 *	−0.237	0.018	−0.367
	*p*	0.990	0.472	0.156	0.393	0.763	0.047 *	0.024 *	0.216	0.878	0.051
Neutrophils (×10^9^/L)	r_s_	−0.139	0.000	−0.022	0.045	0.209	−0.257	−0.072	−0.086	0.011	0.161
	*p*	0.246	0.999	0.854	0.817	0.080	0.179	0.550	0.656	0.930	0.405
Lymphocytes (×10^9^/L)	r_s_	0.057	0.024	−0.080	−0.290	0.131	0.034	−0.166	−0.174	0.168	0.079
	*p*	0.638	0.902	0.508	0.127	0.275	0.860	0.168	0.366	0.163	0.683
Monocytes (×10^9^/L)	r_s_	−0.176	−0.316	0.096	0.355	−0.247 *	0.298	0.209	−0.127	−0.147	−0.486 *
	*p*	0.142	0.095	0.425	0.059	0.038 *	0.117	0.081	0.511	0.222	0.008 *

r_s_: Spearman coefficient. * Statistically significant at *p* ≤ 0.05. WBCs: white blood cells; RBCs: red blood cells; HGB: hemoglobin; HTC: hematocrit; MCV: mean corpuscular volume; MCH: mean corpuscular hemoglobin; MCHC: mean corpuscular hemoglobin concentration; RDW: red cell distribution width; MPV: mean platelet volume; r: Pearson coefficient. * Statistically significant at *p* ≤ 0.05. Evans (1996) suggested the following absolute values for r: 0.00–0.19, “very weak”; 0.20–0.39, “weak”; 0.40–0.59, “moderate”; 0.60–0.79, “strong”; 0.80–1.0, “very strong”.

**Table 7 molecules-28-05582-t007:** Results of the analysis of the studied heavy metals (Cr, Al, Pb, and Cd) with the AutoDock analysis as potential ligands binding to proteins, hormones, genes, and their receptors.

Target (Protein, Hormone, Gene, and Their Receptors)			Global Binding Energy of the Highest Binding Affinity (kcal/mole) of Heavy Metals with the Target (Protein, Hormone, Gene, and Their Receptors) with the Best Enlarged Angle for the Ligand with the Protein			
			Cr Compound CID: 23976	Al Compound CID: 5359268	Cd Compound CID: 23973	Pb Compound CID: 5352425
Cholinesterase	Target Protein	1P0P	−8.49 (Lig. Pos. 3/100) 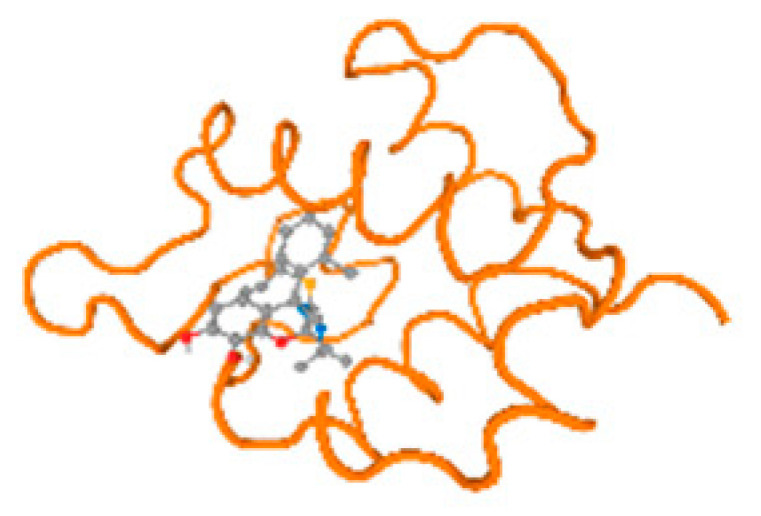	−8.69 (Lig. Pos. 3/100) 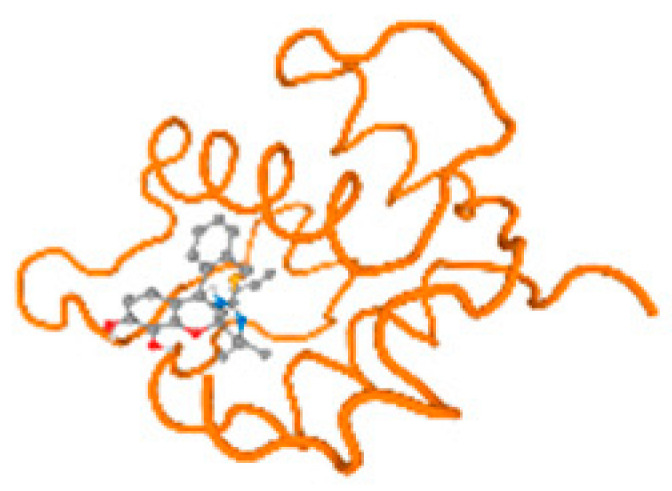	−8.46 (Lig. Pos. 3/100) 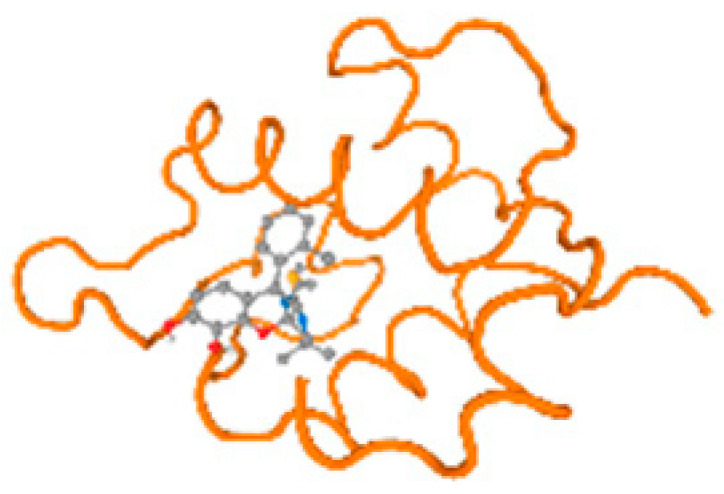	−8.72 (Lig. Pos. 3/60) 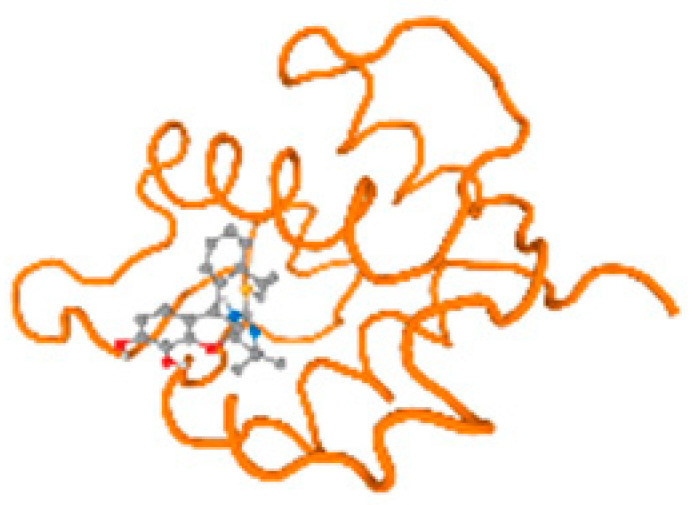
	Target Receptor	6EP4	−8.66 (Lig. Pos. 3/100) 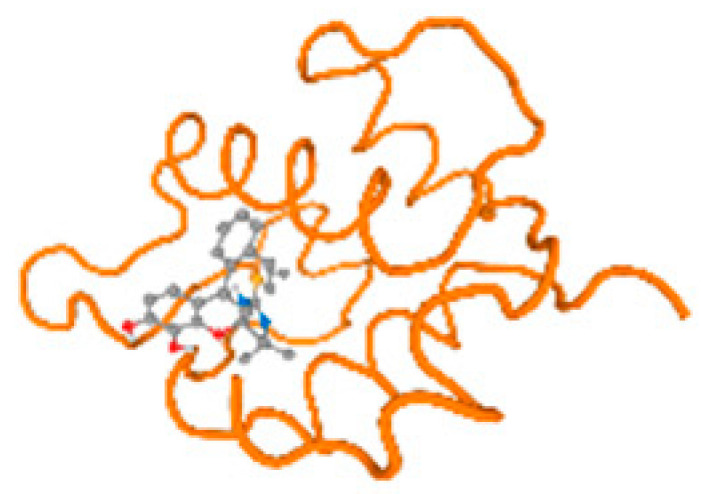	−8.72 (Lig. Pos. 3/70) 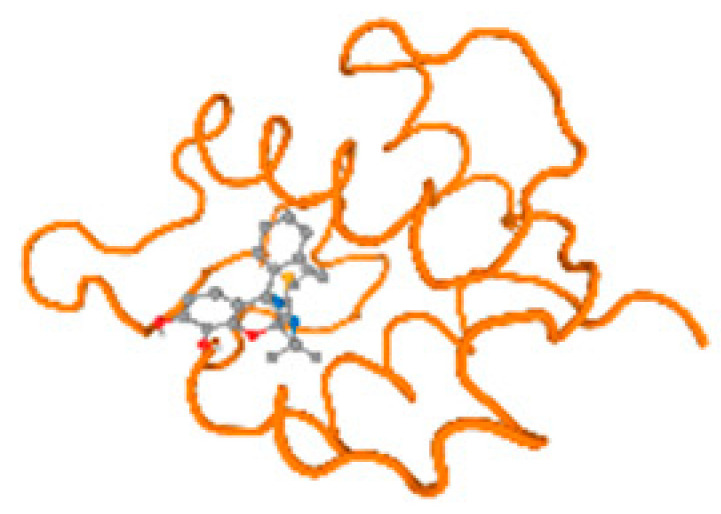	−6.9 (Lig. Pos. 2/20) 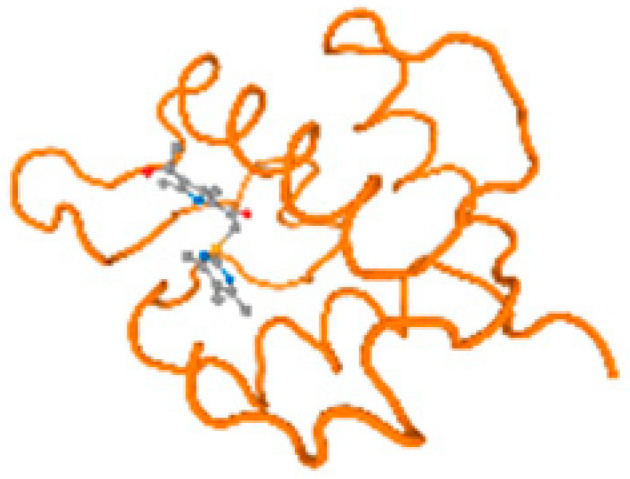	−7.62 (Lig. Pos. 9/40) 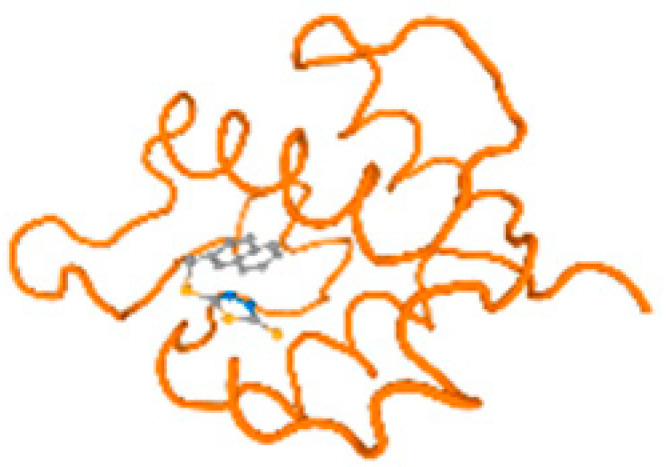
P53	Target Protein	7VOU	−8.72 (Lig. Pos. 3/30) 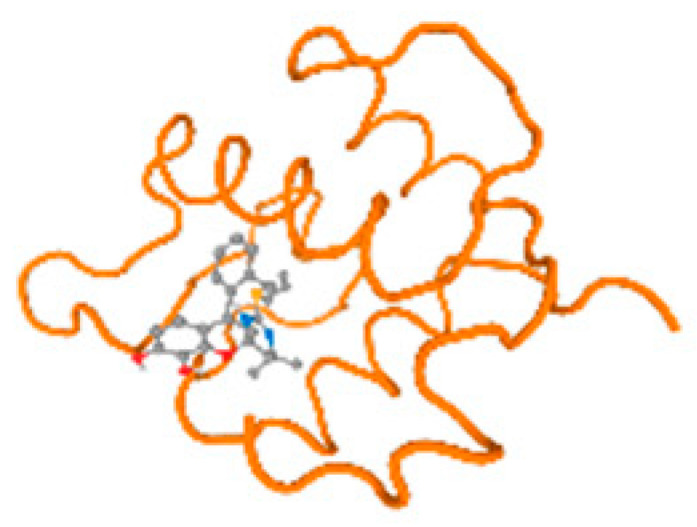	−8.6 (Lig. Pos. 3/30) 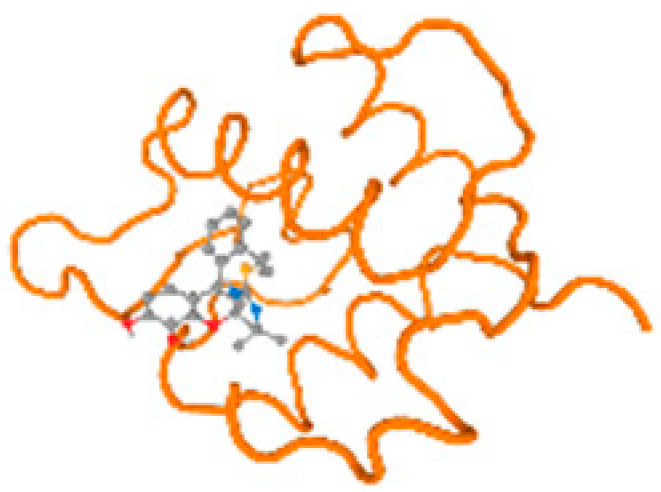	−8.69 (Lig. Pos. 3/100) 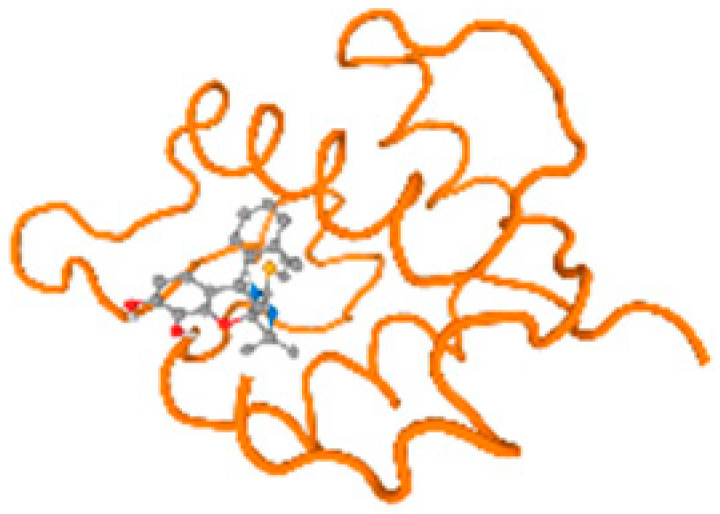	−8.67 (Lig. Pos. 3/40) 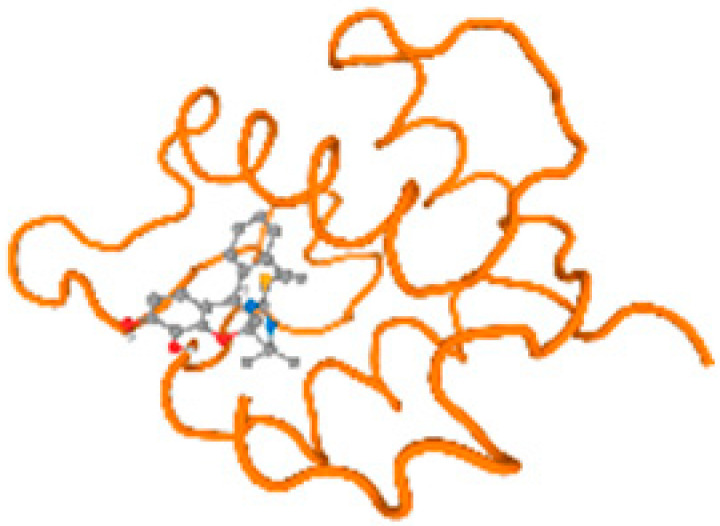
		2LY4	−6.9 (Lig. Pos. 2/20) 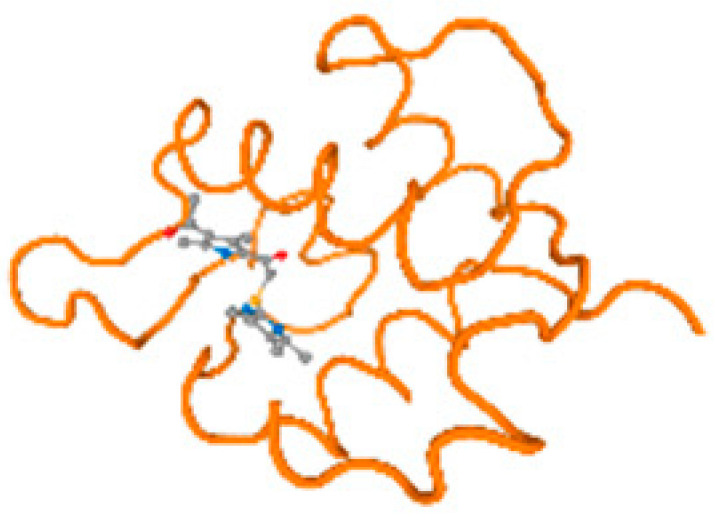	−8.71 (Lig. Pos. 3/100) 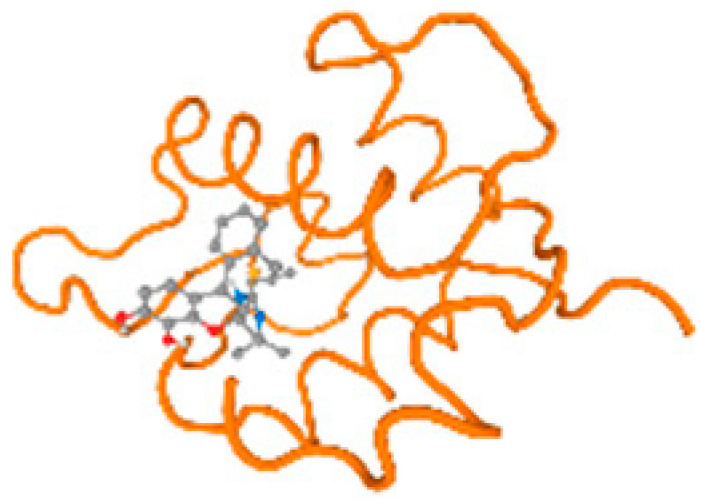	−8.65 (Lig. Pos. 3/50) 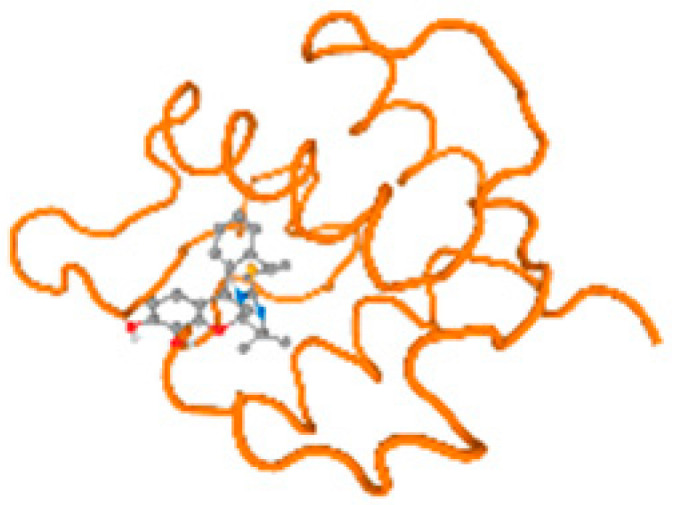	−8.7 (Lig. Pos. 3/30) 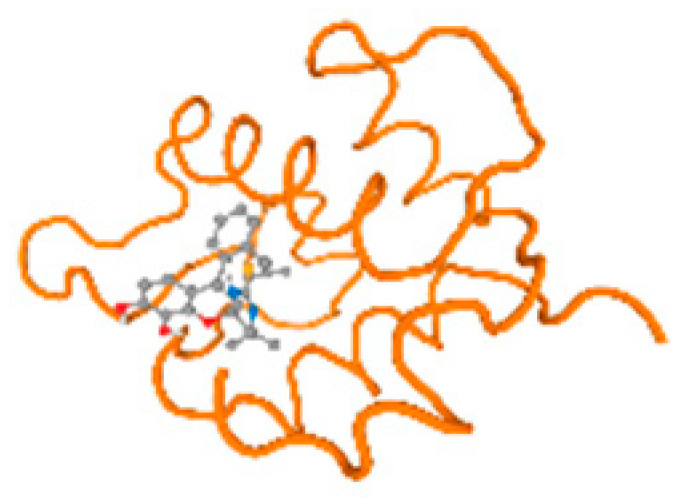
		2JTX	−8.63 (Lig. Pos. 3/100) 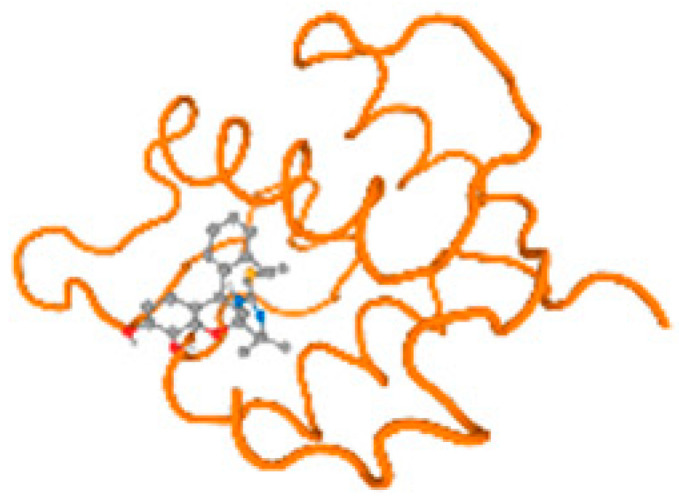	−7.62 (Lig. Pos. 9/70) 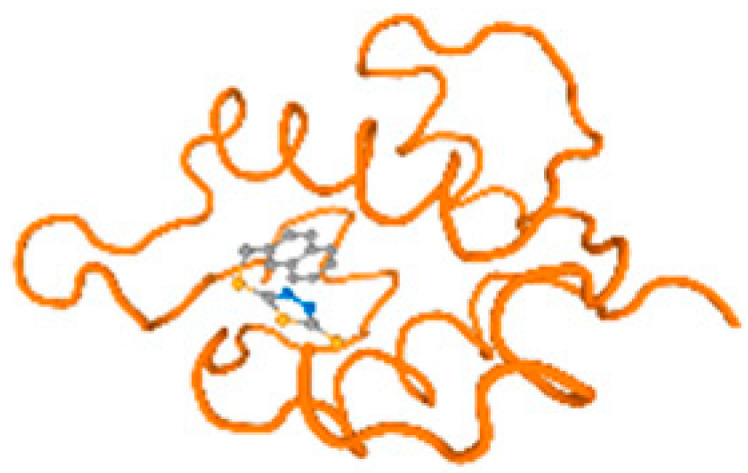	−7.63 (Lig. Pos. 9/70) 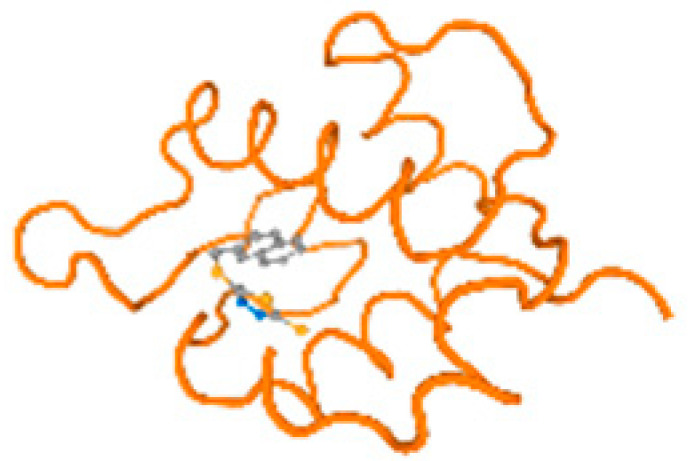	−8.69 (Lig. Pos. 3/100) 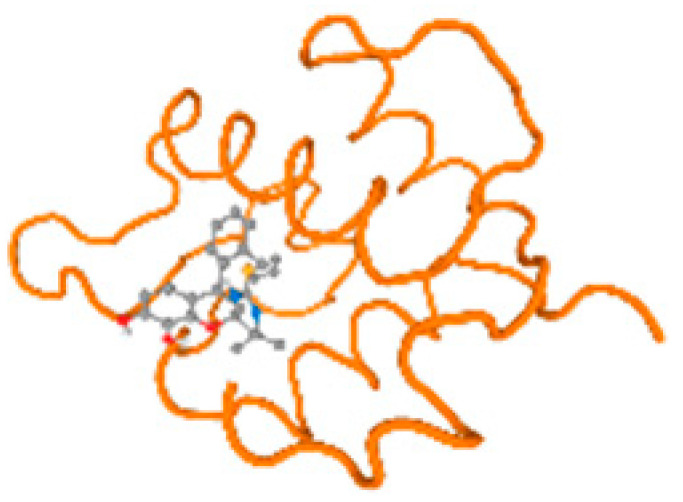
	Target Receptor	6VTH	−8.74 (Lig. Pos. 3/30) 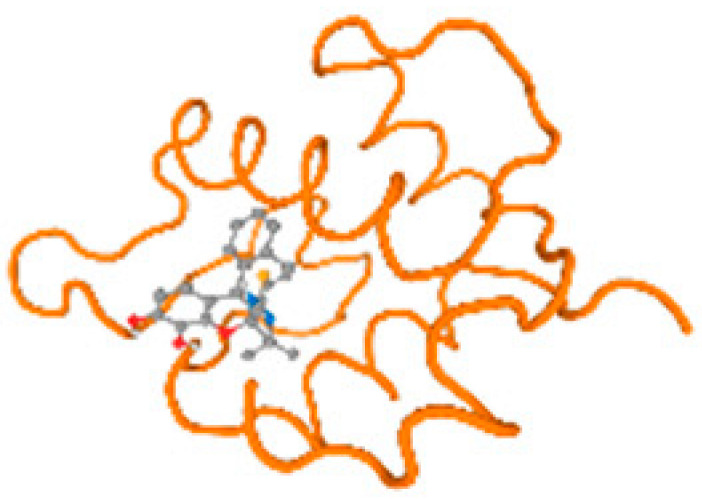	−8.65 (Lig. Pos. 3/90) 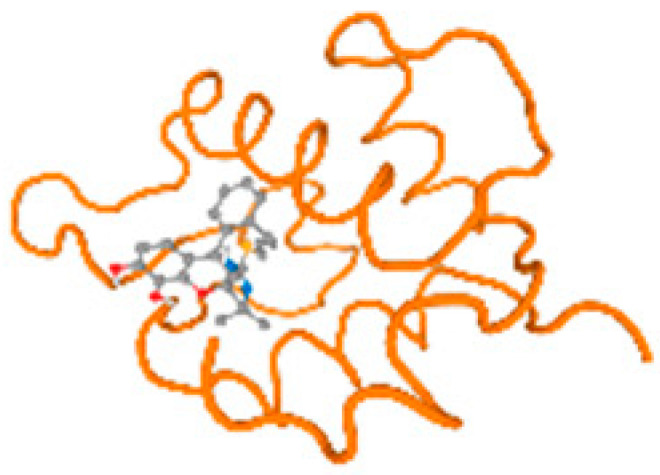	−8.7 (Lig. Pos. 3/100) 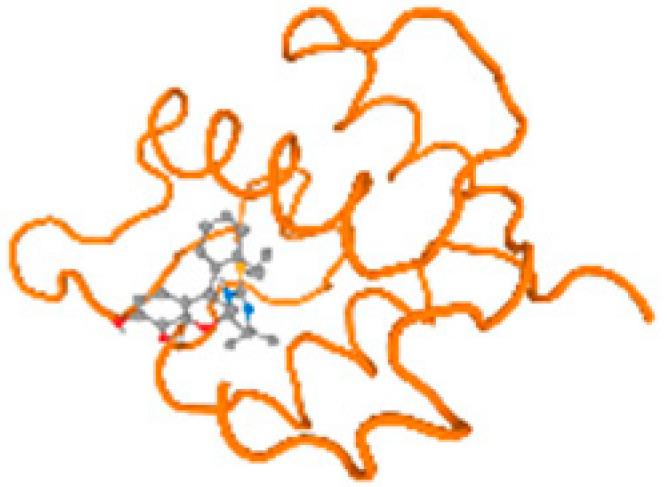	−8.65 (Lig. Pos. 3/100) 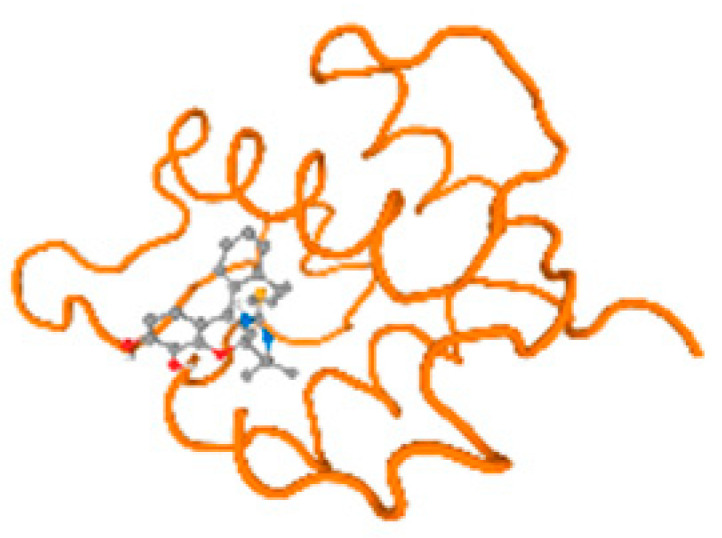
Dopamine	Target Protein	5PAH	−8.71 (Lig. Pos. 3/100) 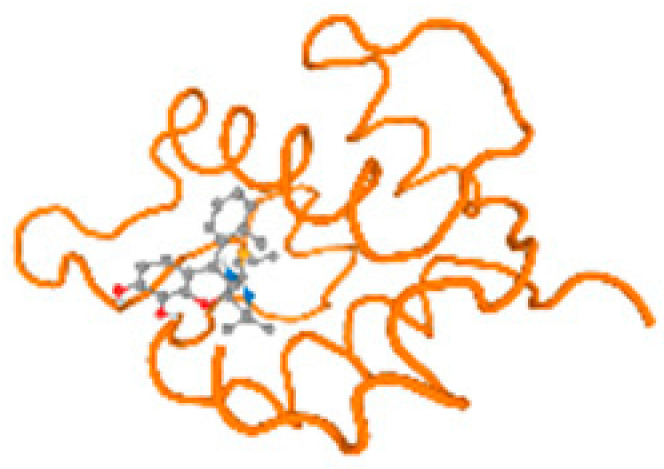	−7.6 (Lig. Pos. 9/40) 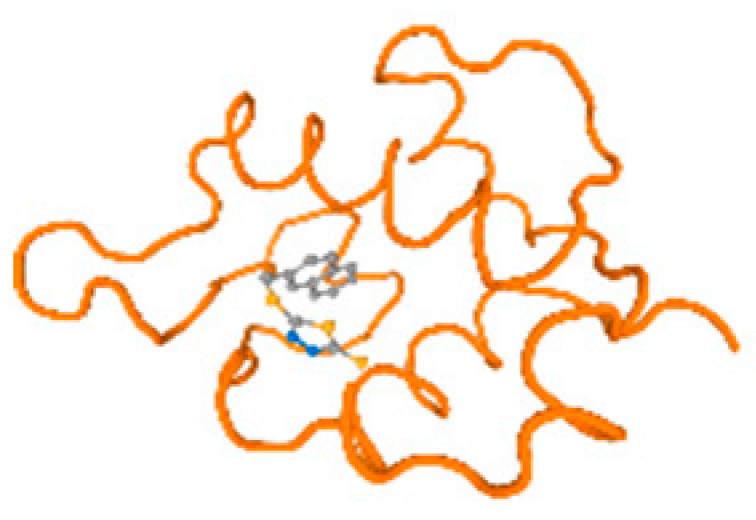	−7.63 (Lig. Pos. 9/30) 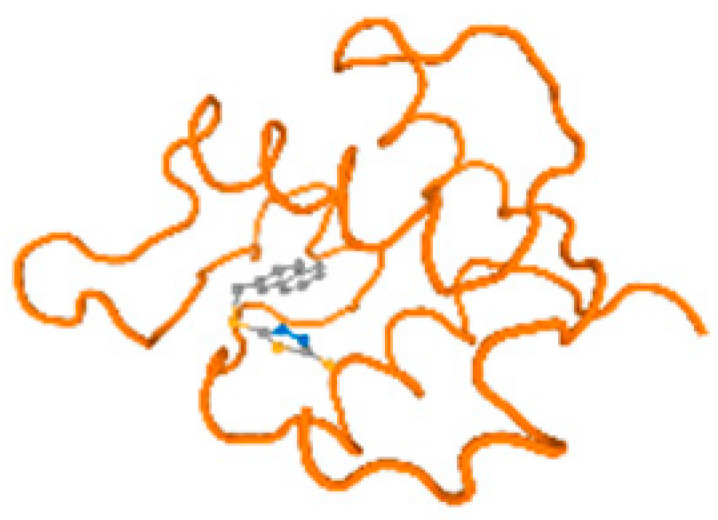	−8.73 (Lig. Pos. 3/100) 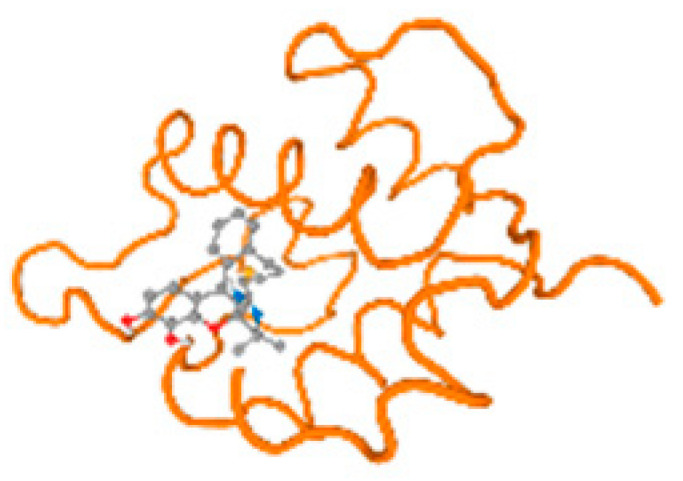
	Target Receptor	3PBL	−7.65 (Lig. Pos. 9/30) 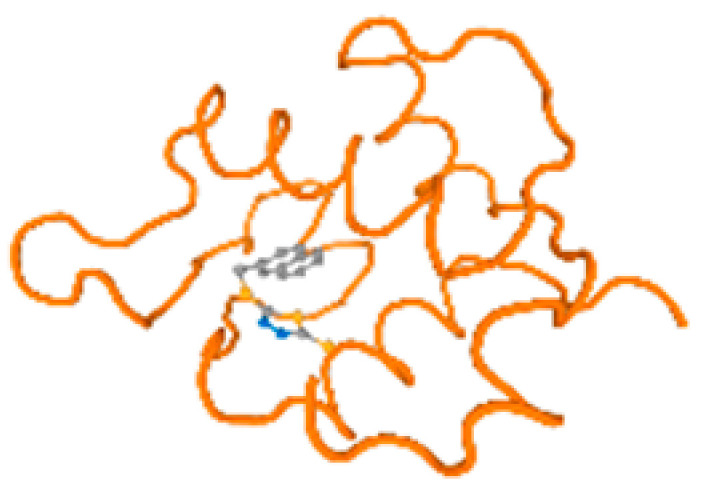	−8.71 (Lig. Pos. 3/100) 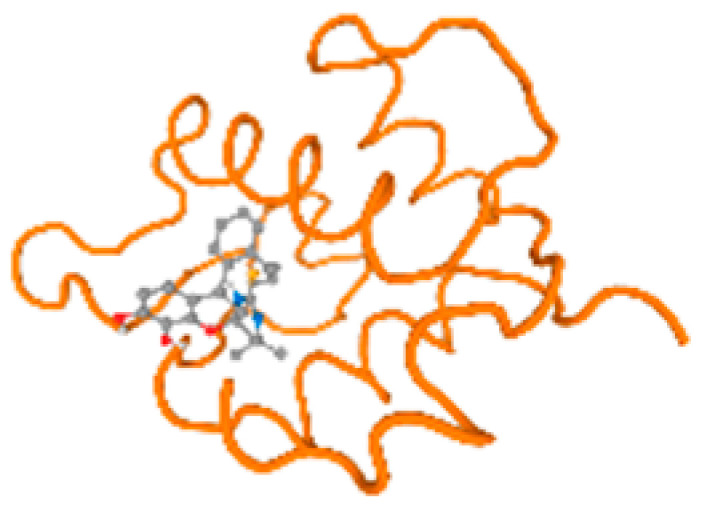	−8.74 (Lig. Pos. 3/100) 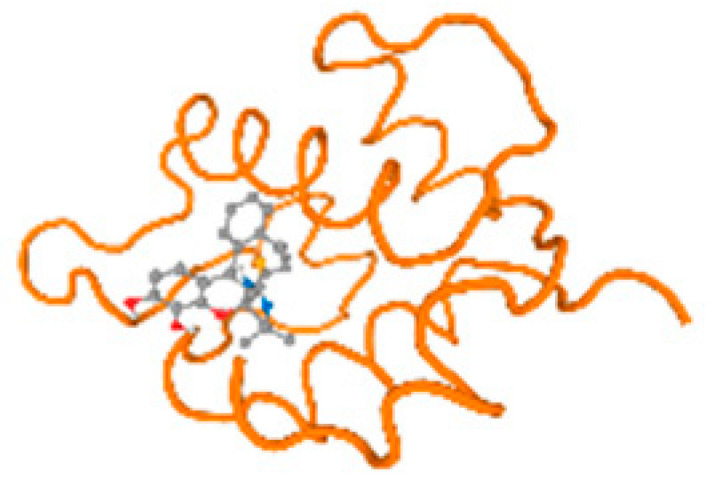	−8.65 (Lig. Pos. 3/100) 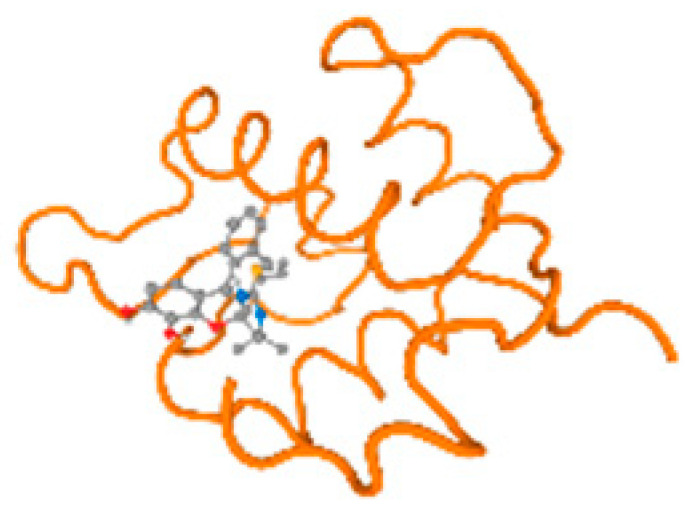
Estrogen	Target Protein	1FDW	−8.73 (Lig. Pos. 3/100) 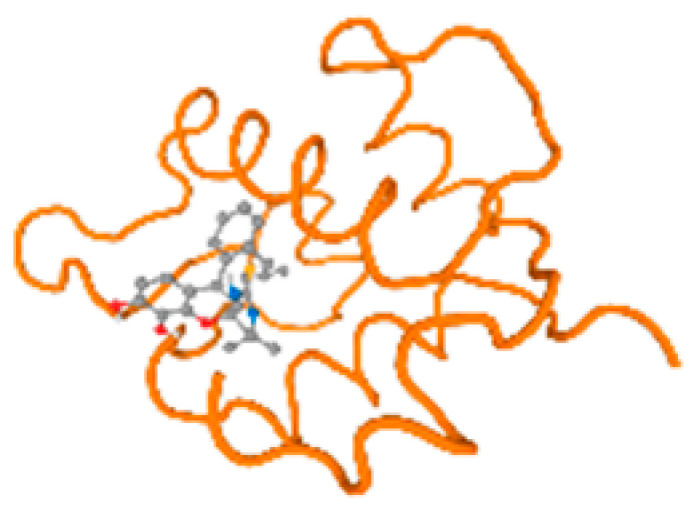	−8.68 (Lig. Pos. 3/100) 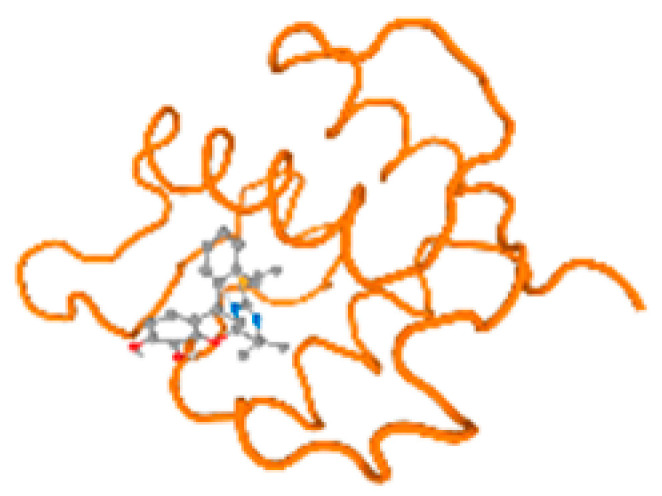	−8.65 (Lig. Pos. 3/90) 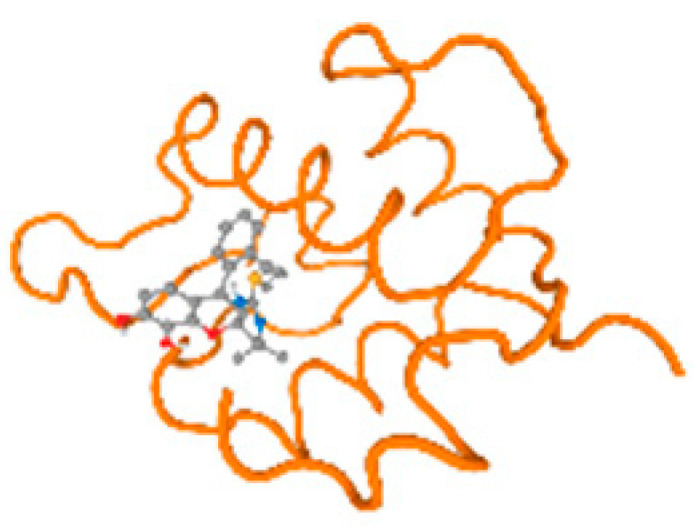	−8.57 (Lig. Pos. 3/100) 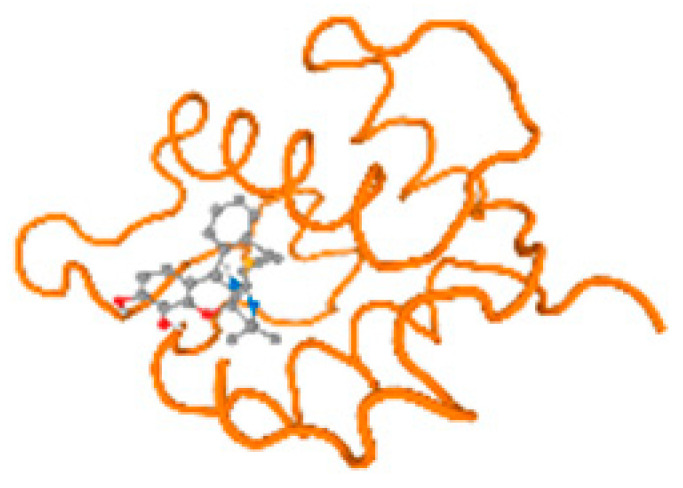
	Target Receptor	1L2J	−8.74 (Lig. Pos. 3/100) 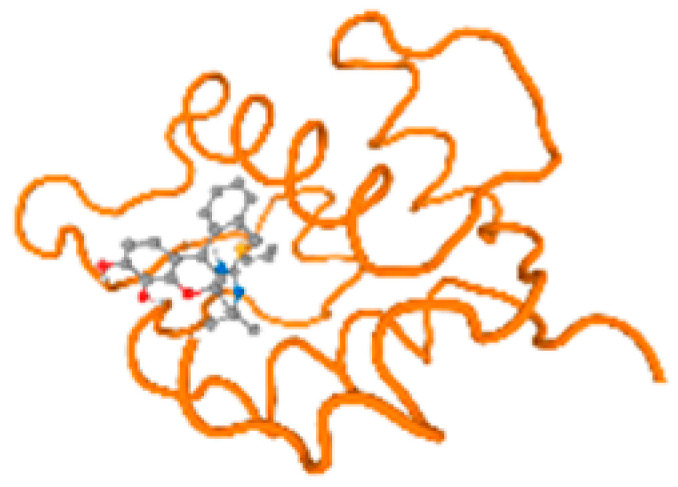	−8.72 (Lig. Pos. 3/100) 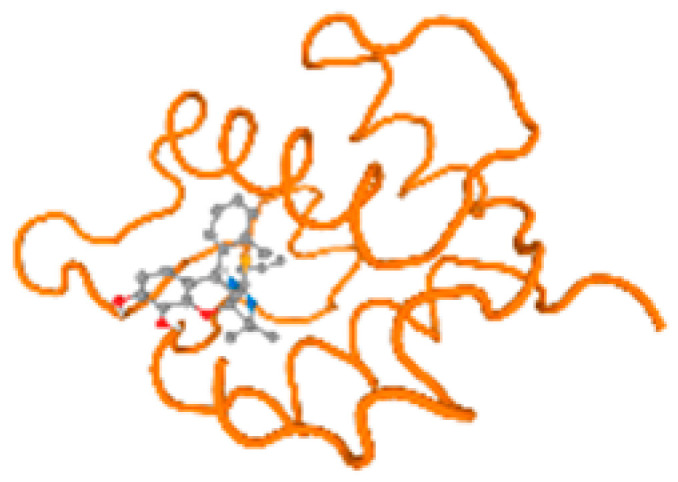	−8.62 (Lig. Pos. 3/100) 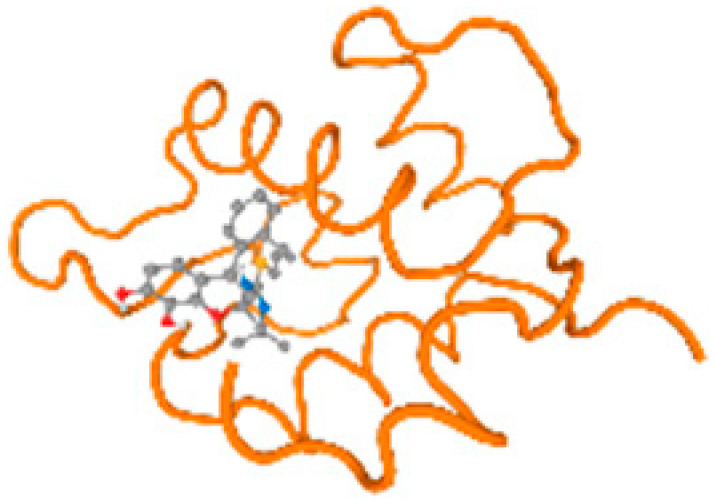	−8.67 (Lig. Pos. 3/100) 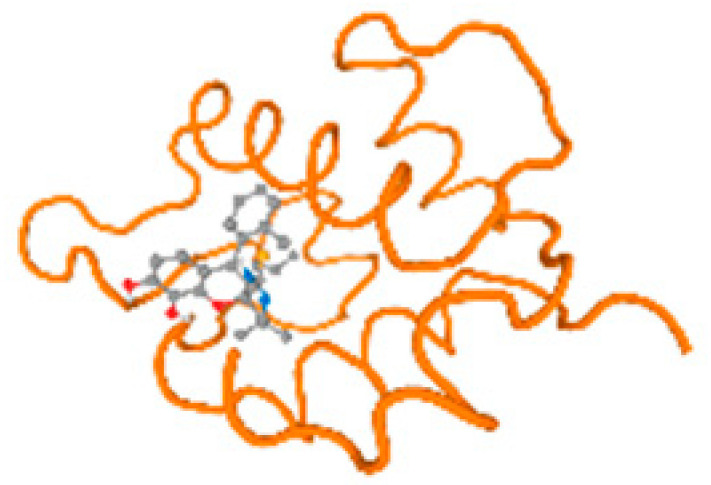
		4J26	−8.45 (Lig. Pos. 3/100) 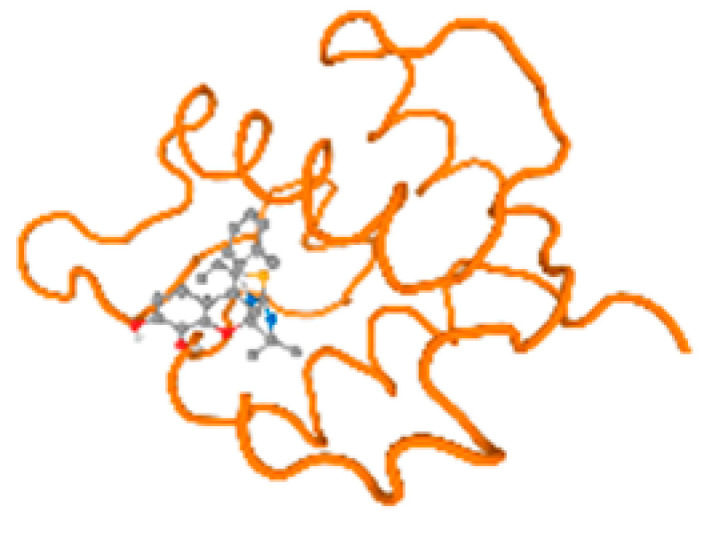	−8.69 (Lig. Pos. 3/100) 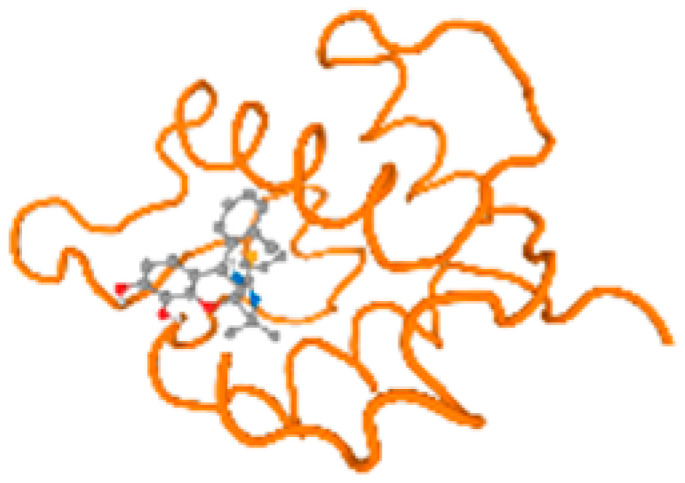	−8.71 (Lig. Pos. 3/100) 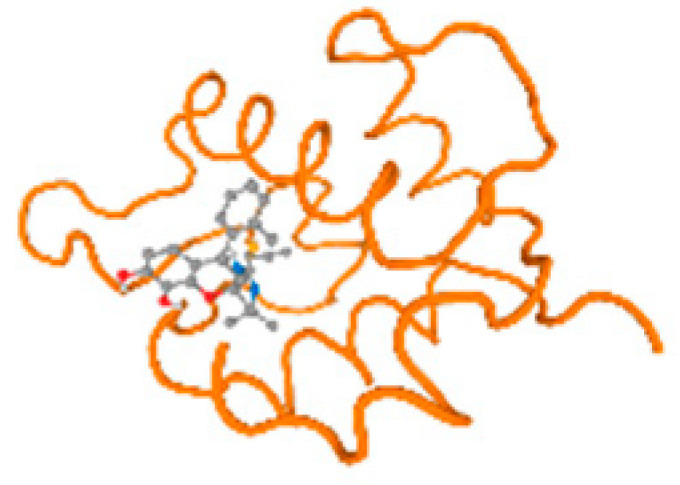	−8.7 (Lig. Pos. 3/100) 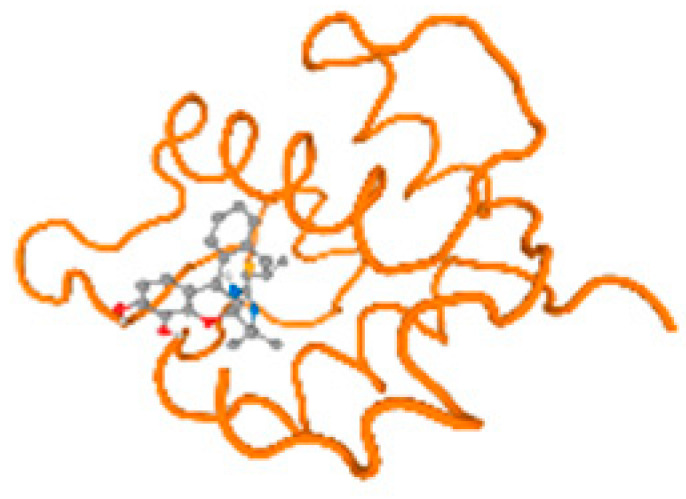
		4J24	−8.71 (Lig. Pos. 3/100) 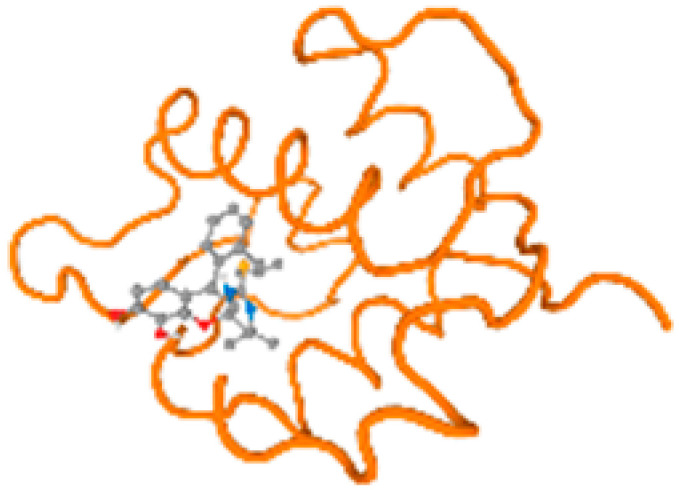	−8.71 (Lig. Pos. 3/100) 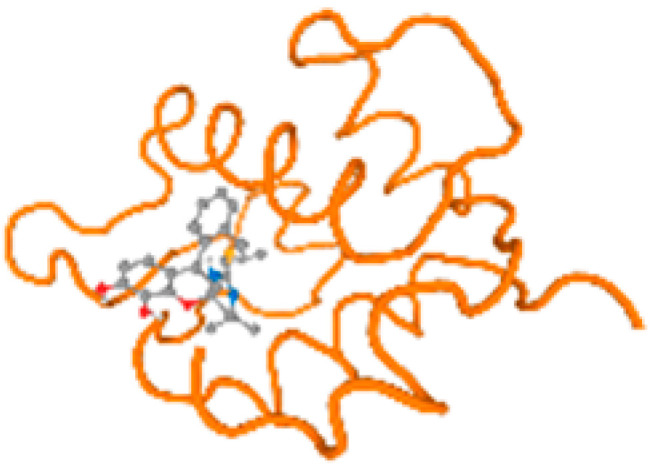	−8.48 (Lig. Pos. 3/100) 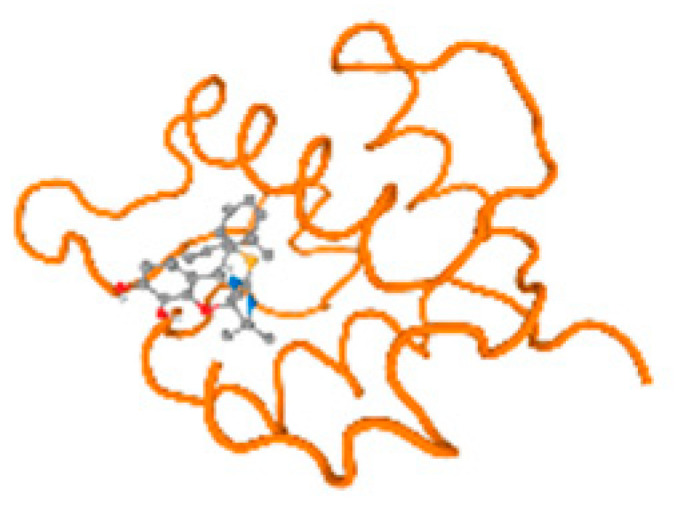	−8.48 (Lig. Pos. 3/100) 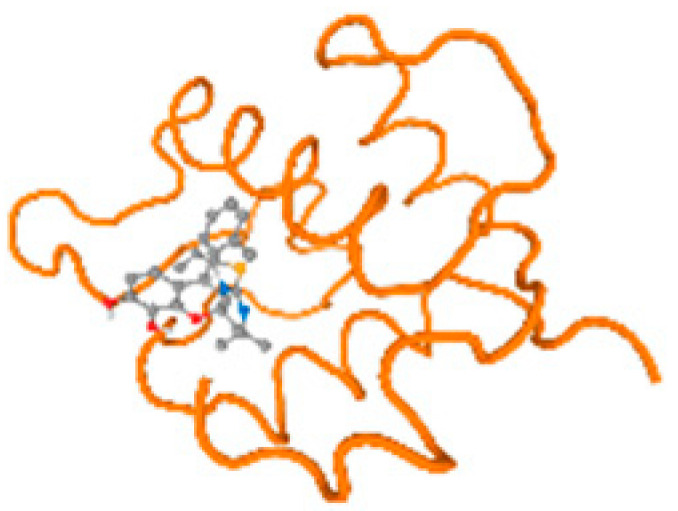
Metallothionein	Target Protein	1MHU	−8.7 (Lig. Pos. 3/100) 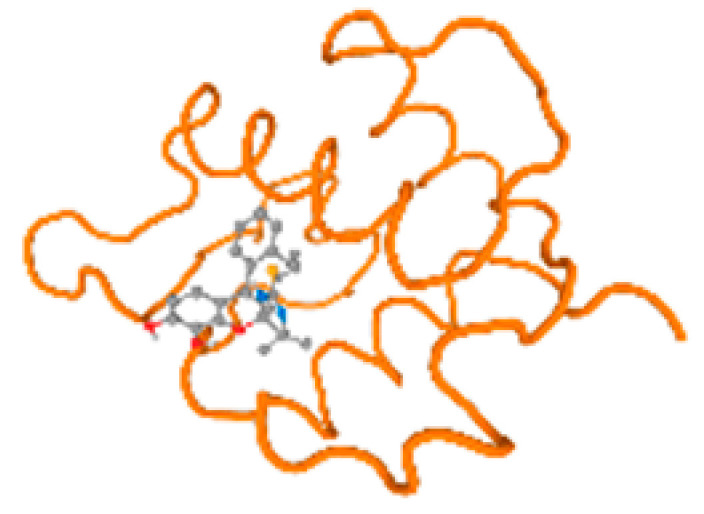	−8.69 (Lig. Pos. 3/100) 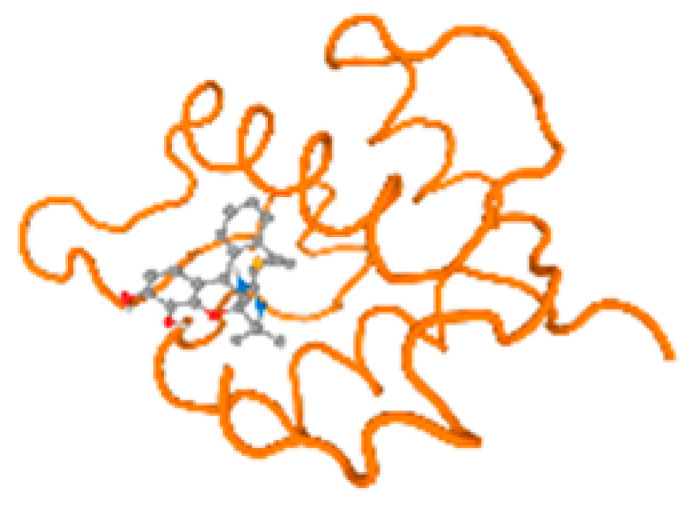	−8.65 (Lig. Pos. 3/100) 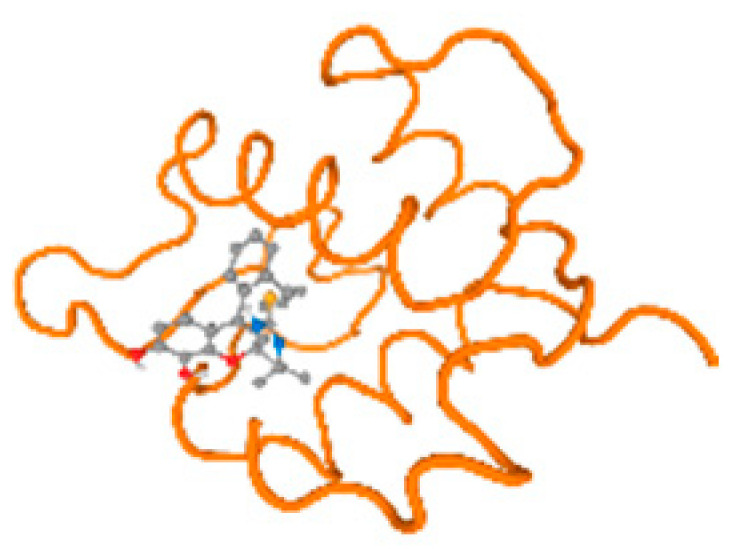	−8.72 (Lig. Pos. 3/60) 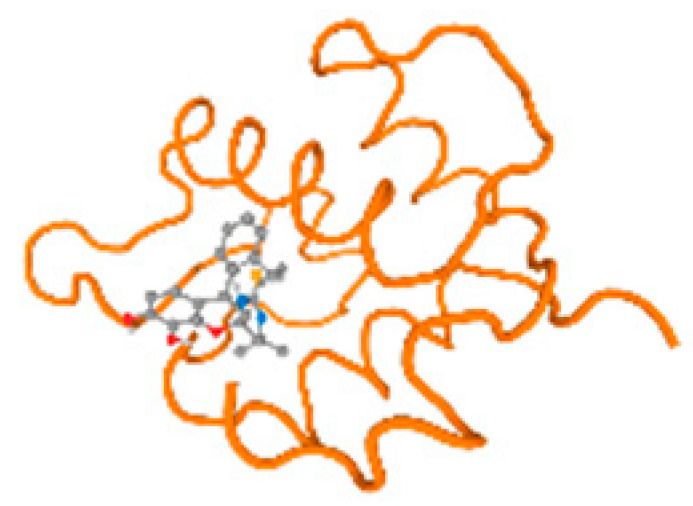
	Target Receptor	2MHU	−8.69 (Lig. Pos. 3/100) 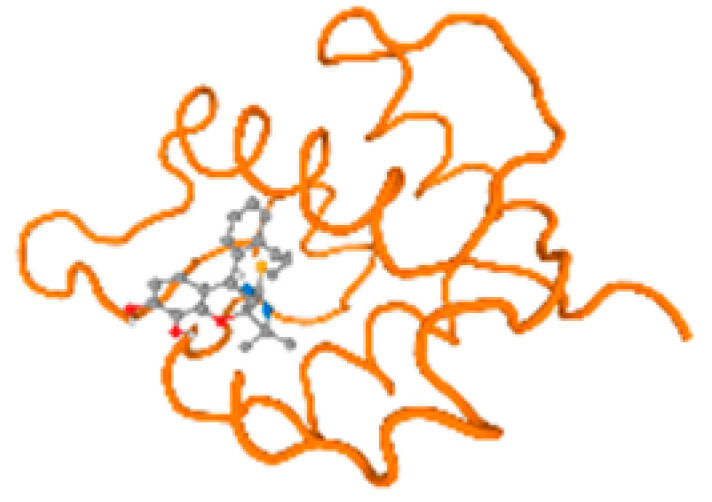	−8.6 (Lig. Pos. 3/100) 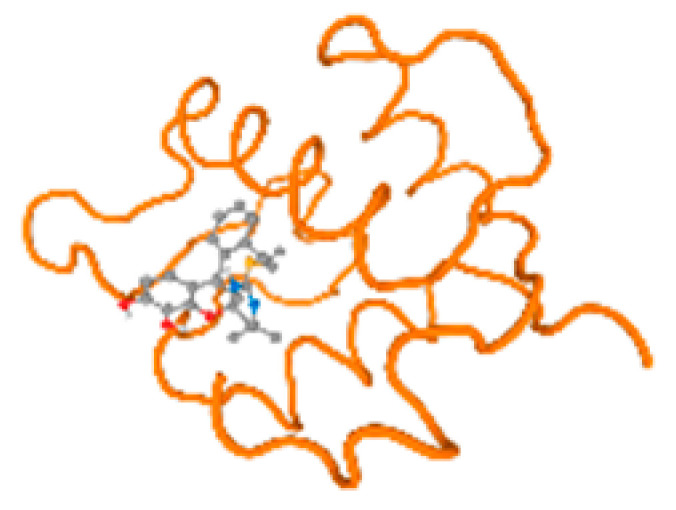	−8.72 (Lig. Pos. 3/100) 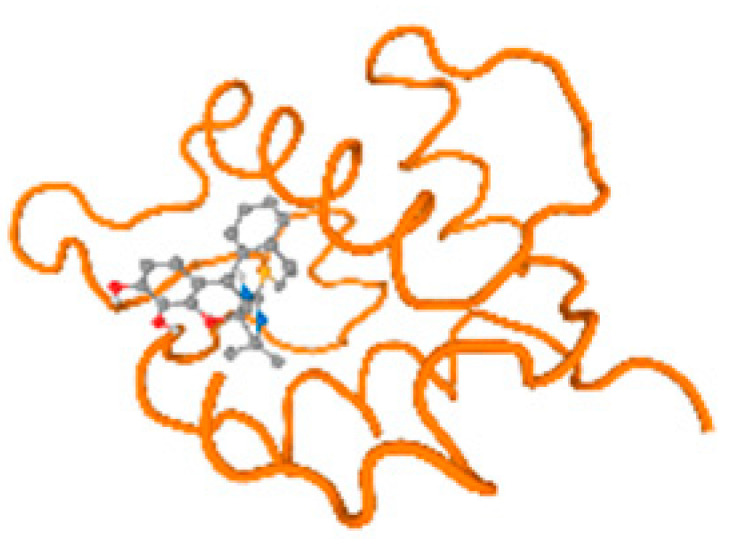	−8.63 (Lig. Pos. 3/100) 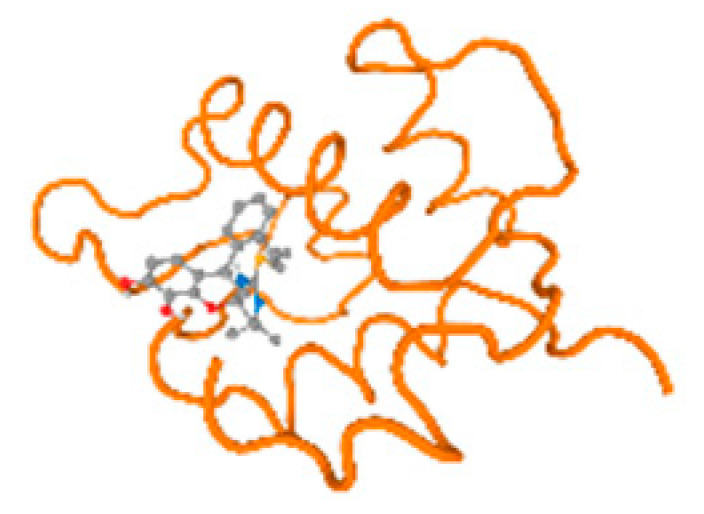
		2F5H	−8.48 (Lig. Pos. 3/100) 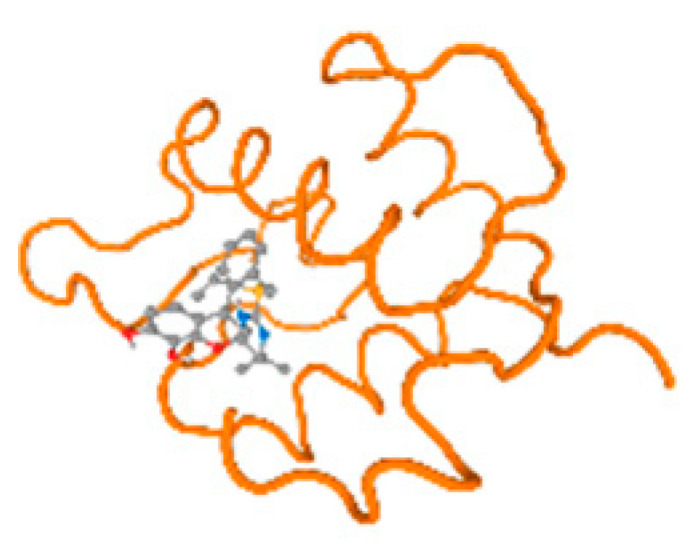	−8.63 (Lig. Pos. 3/100) 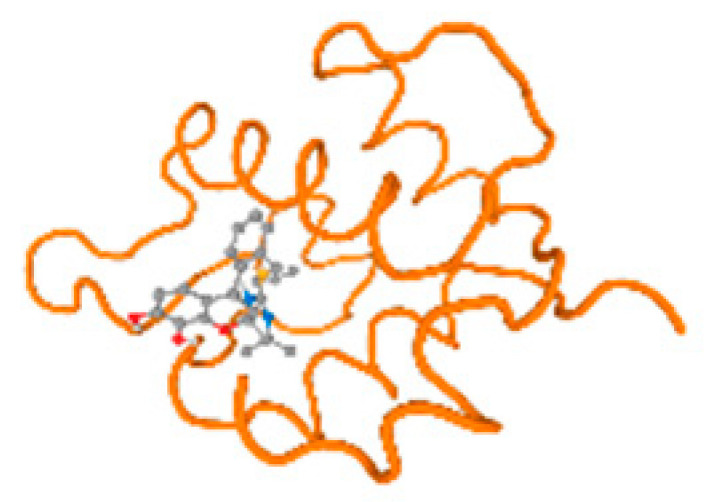	−8.63 (Lig. Pos. 3/100) 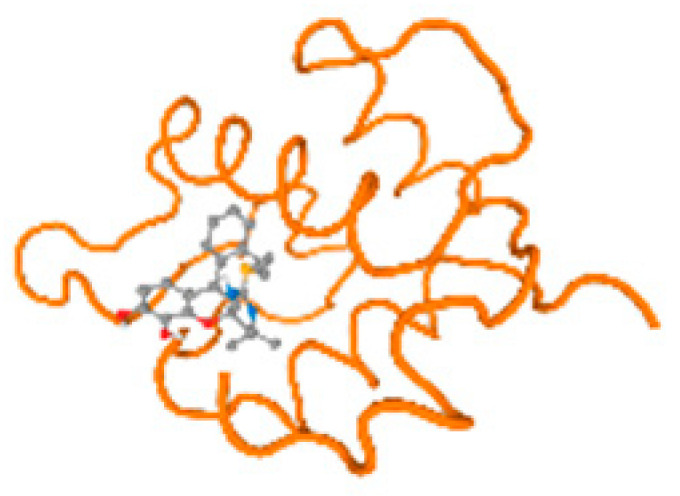	−8.73 (Lig. Pos. 3/100) 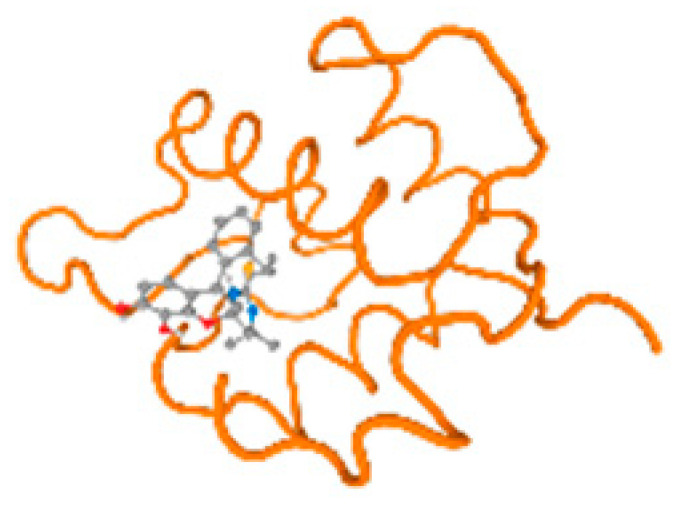
Keratin	Target Protein	6EC0	−8.7 (Lig. Pos. 3/100) 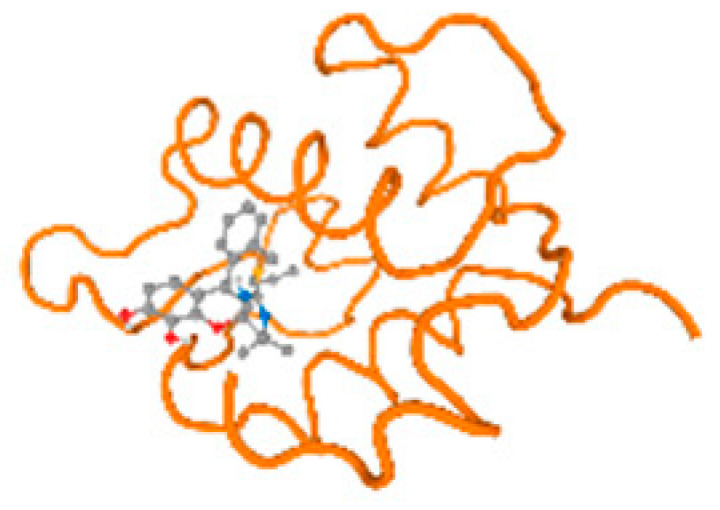	−8.73 (Lig. Pos. 3/90) 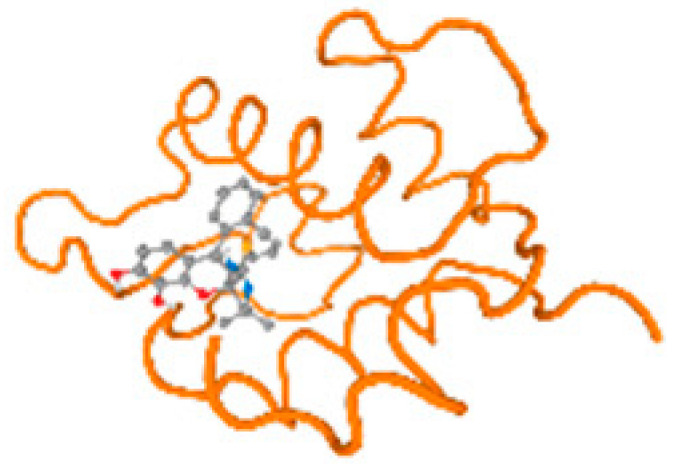	−8.49 (Lig. Pos. 3/100) 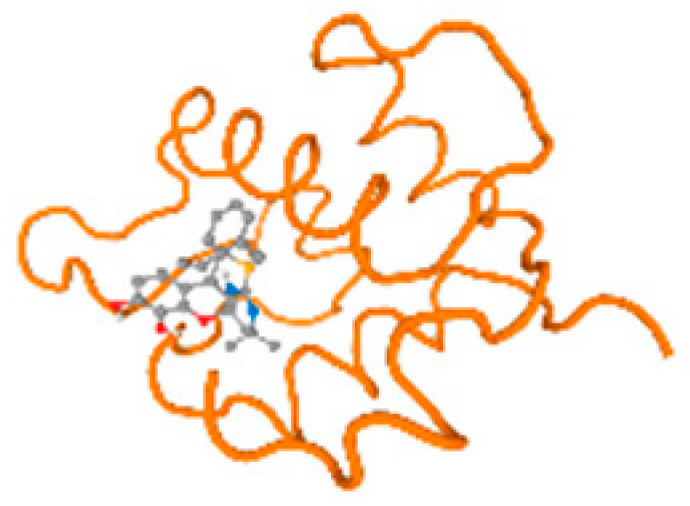	−8.49 (Lig. Pos. 3/100) 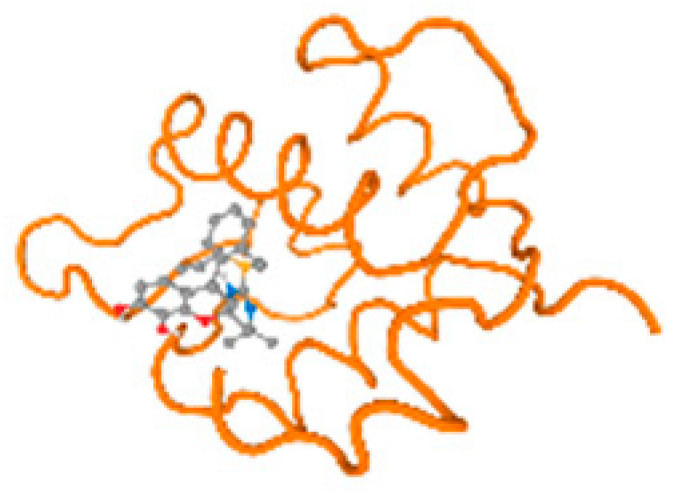
	Target Receptor	4ZRY	−8.7 (Lig. Pos. 3/100) 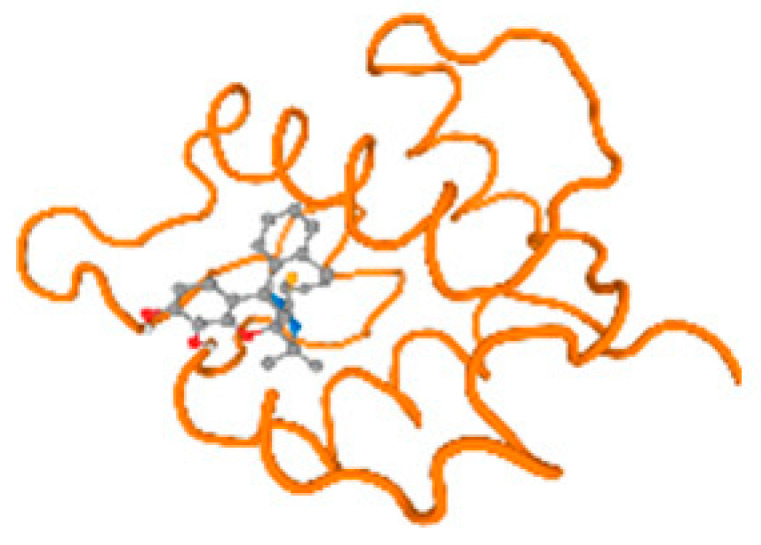	−8.47 (Lig. Pos. 3/100) 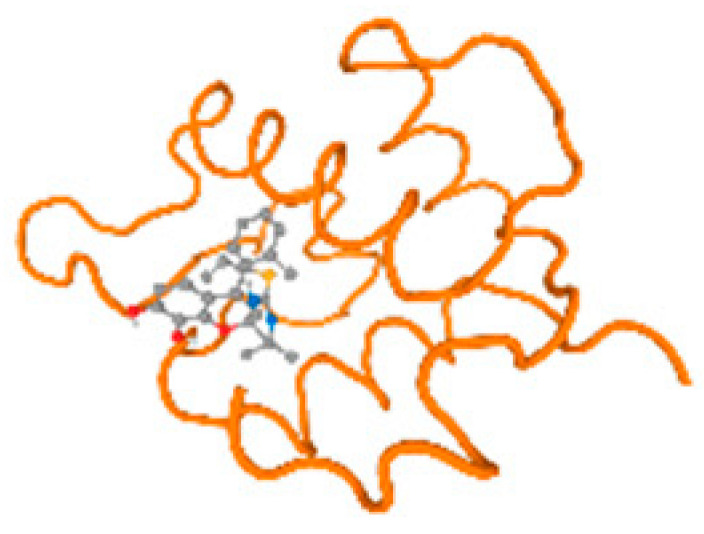	−8.73 (Lig. Pos. 3/100) 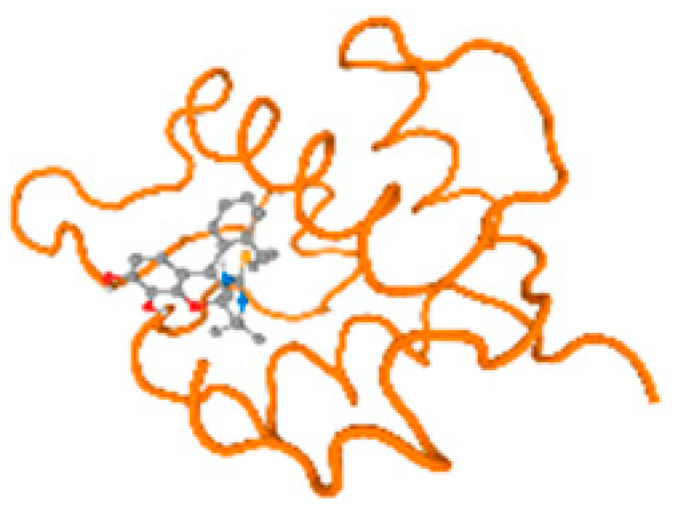	−8.68 (Lig. Pos. 3/100) 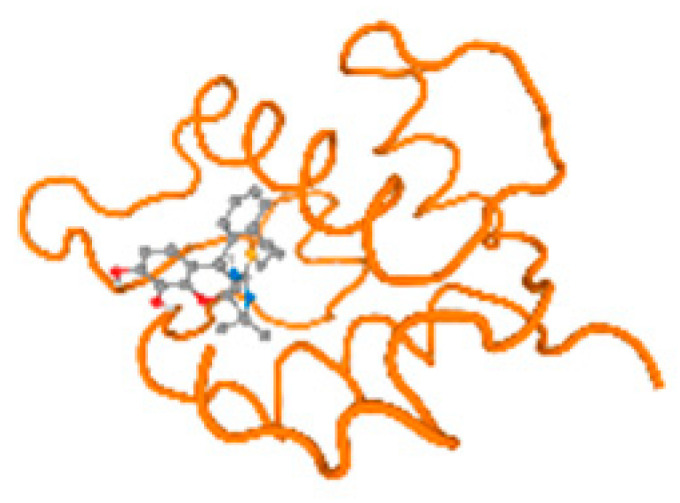
Protein kinase enzyme	Target Protein	1P4F	−8.7 (Lig. Pos. 3/100) 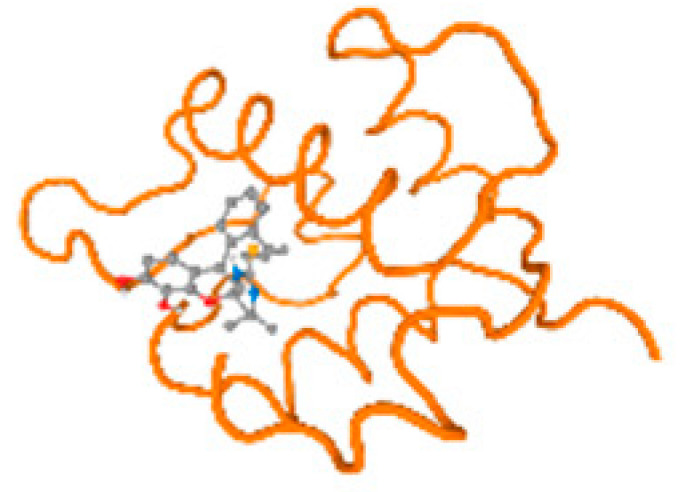	−8.66 (Lig. Pos. 3/100) 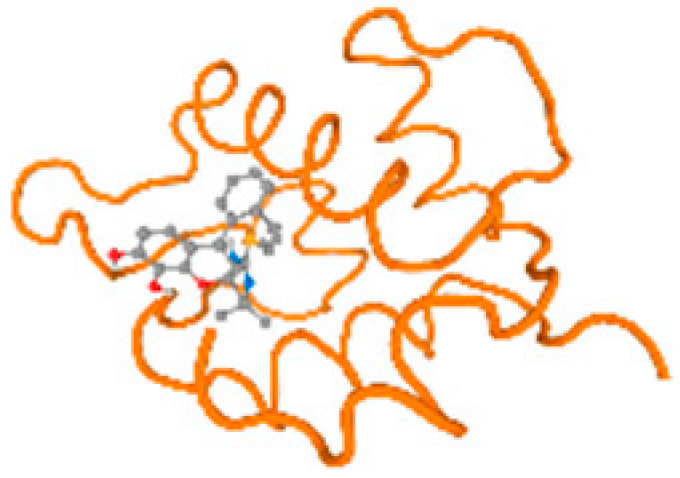	−8.69 (Lig. Pos. 3/100) 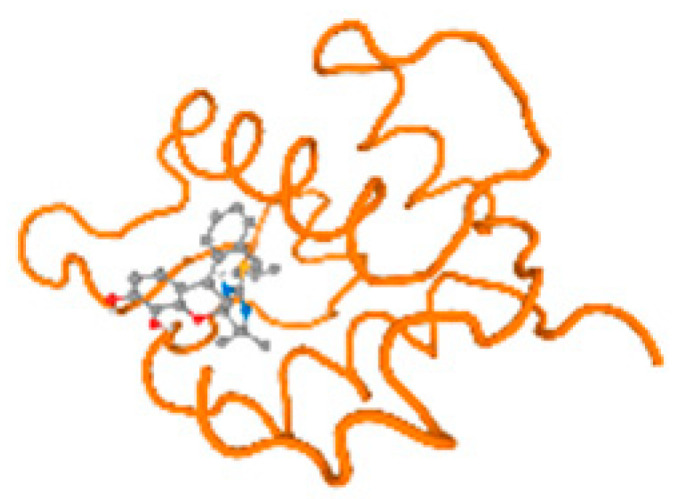	−8.26 (Lig. Pos. 4/100) 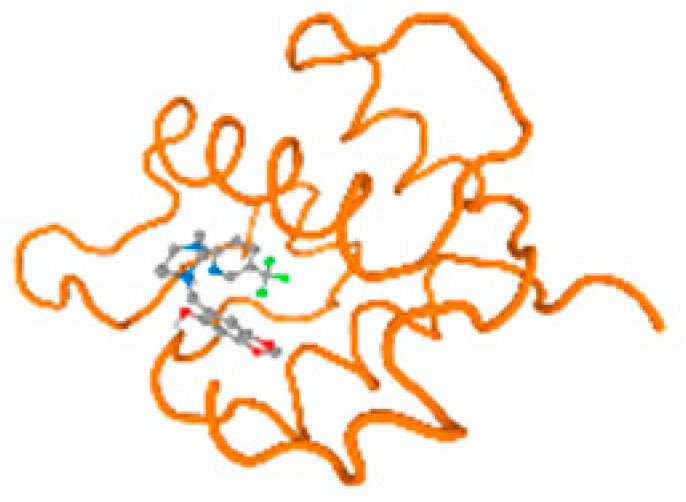
		5IKP	−8.67 (Lig. Pos. 3/100) 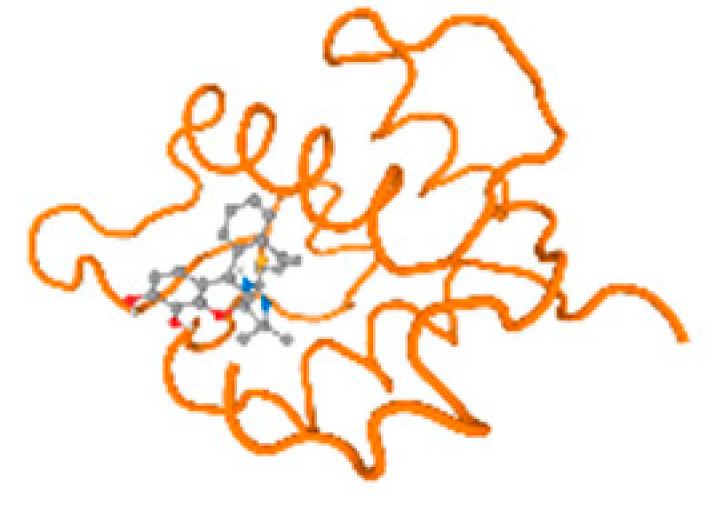	−8.66 (Lig. Pos. 3/100) 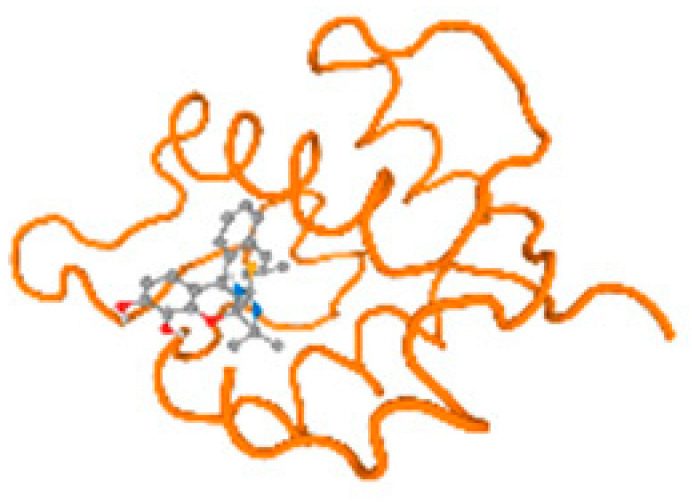	−8.67 (Lig. Pos. 3/100) 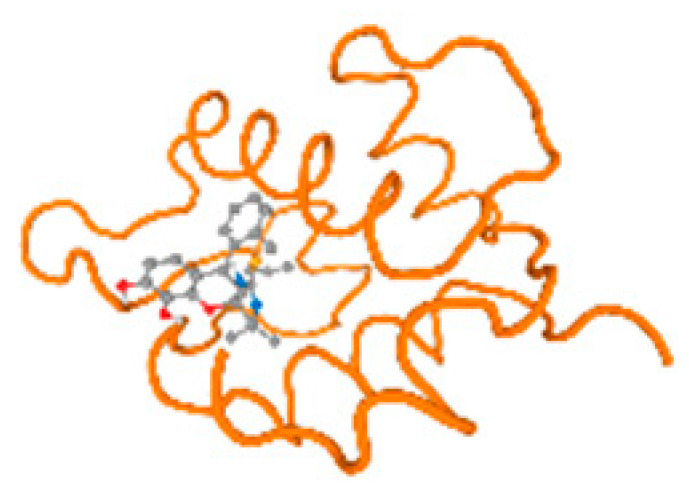	−8.67 (Lig. Pos. 3/100) 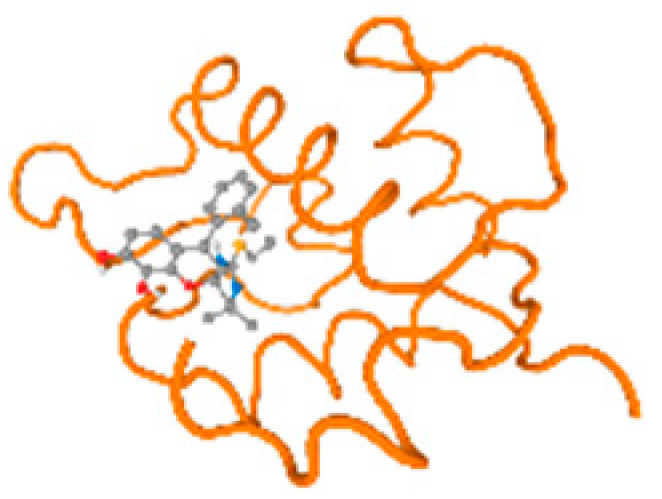
	Target Receptor	1LHR	−8.45 (Lig. Pos. 4/100) 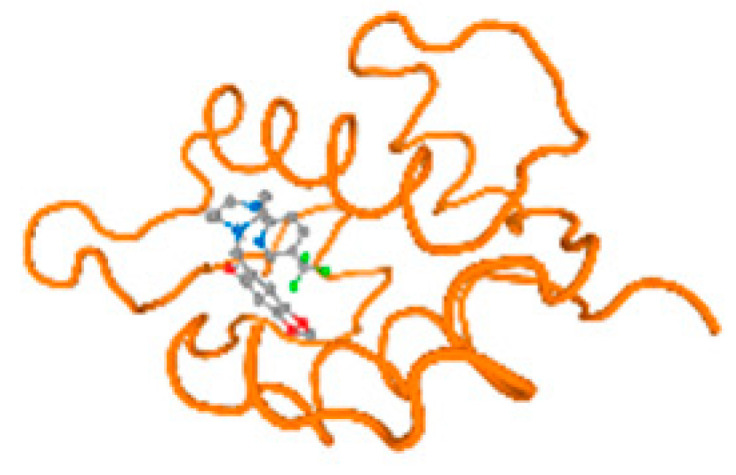	−8.7 (Lig. Pos. 3/100) 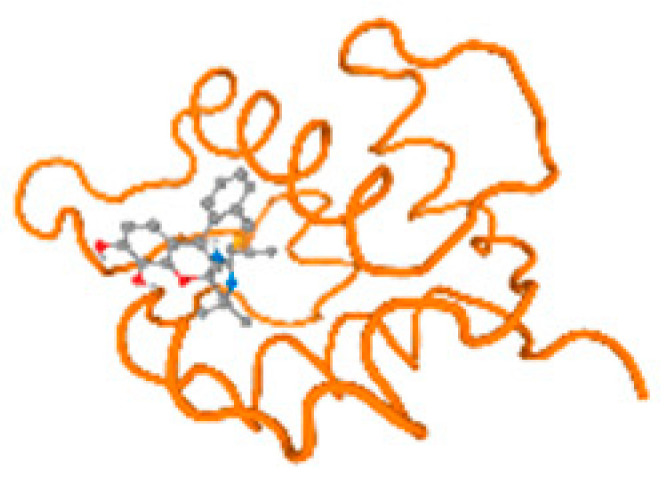	−8.71 (Lig. Pos. 3/100) 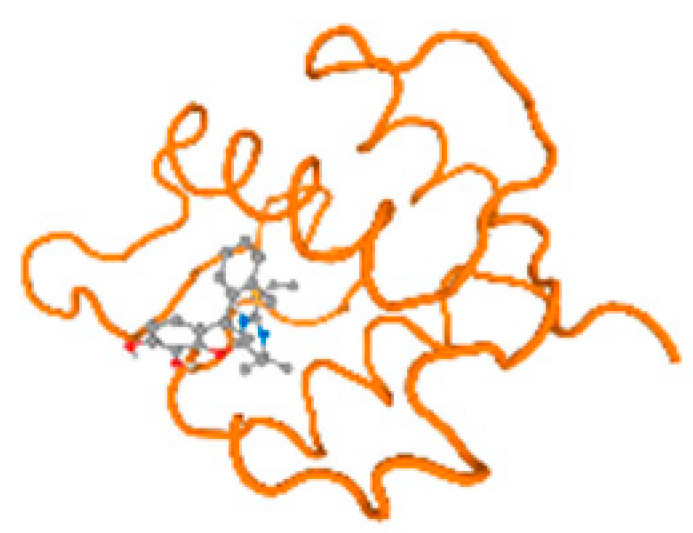	−8.68 (Lig. Pos. 3/100) 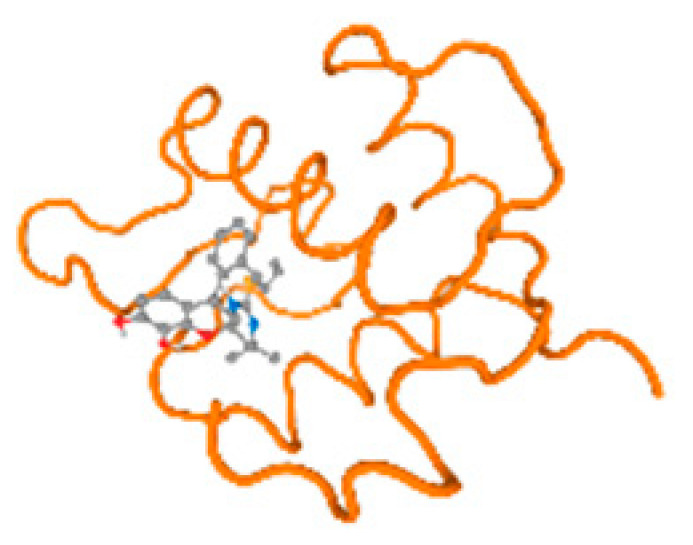
		6E0R	−8.46 (Lig. Pos. 3/100) 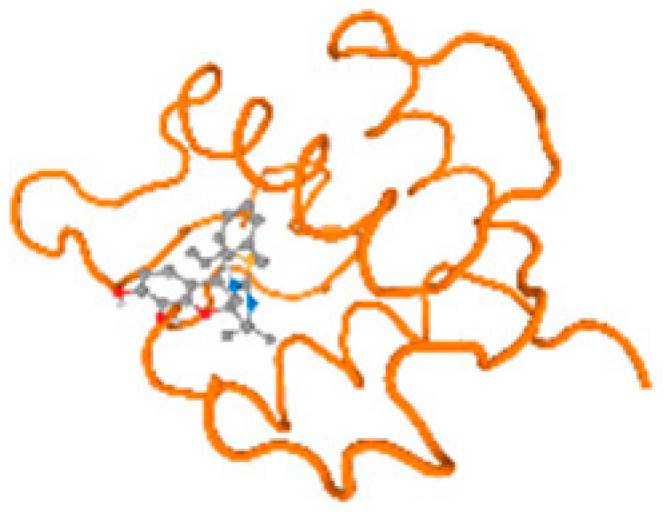	−8.67 (Lig. Pos. 3/100) 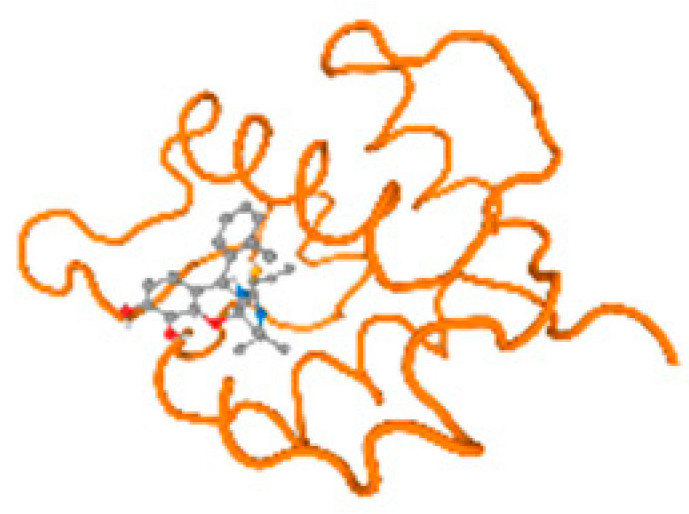	−8.71 (Lig. Pos. 3/90) 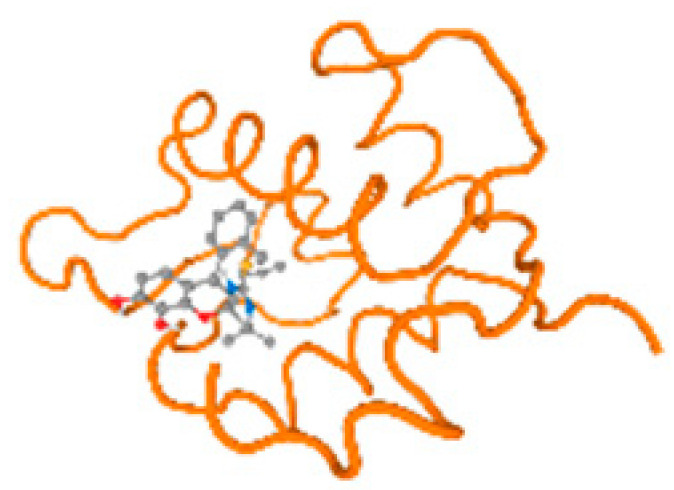	−8.48 (Lig. Pos. 3/80) 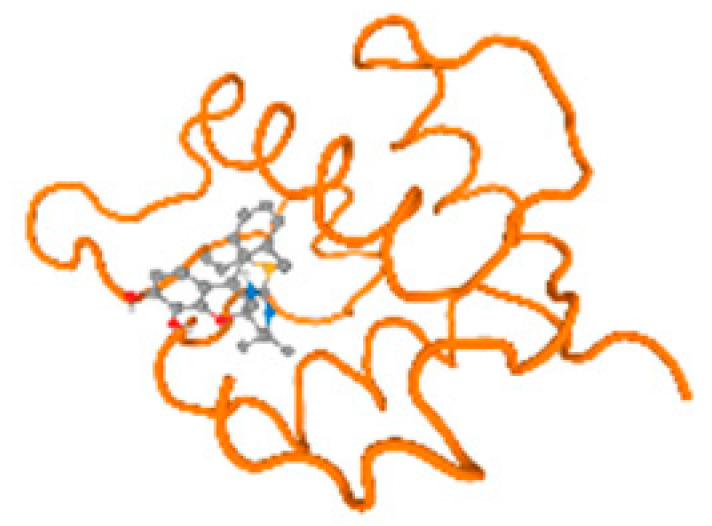
Beta Amyloid	Target Protein	5TXJ	−8.68 (Lig. Pos. 3/100) 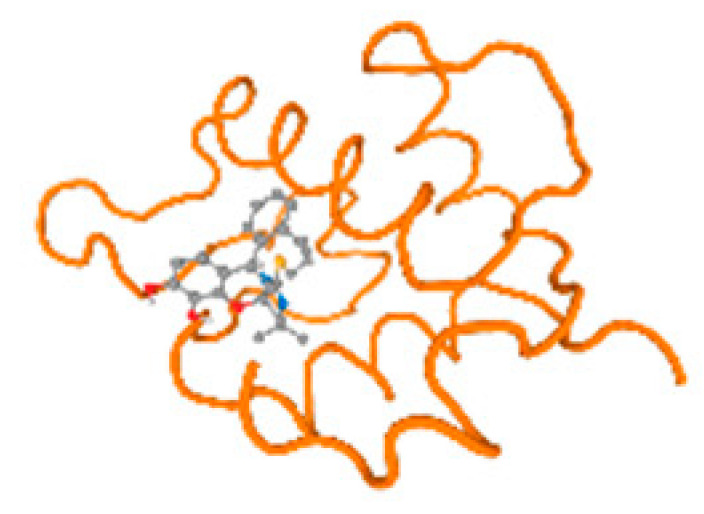	−8.71 (Lig. Pos. 3/100) 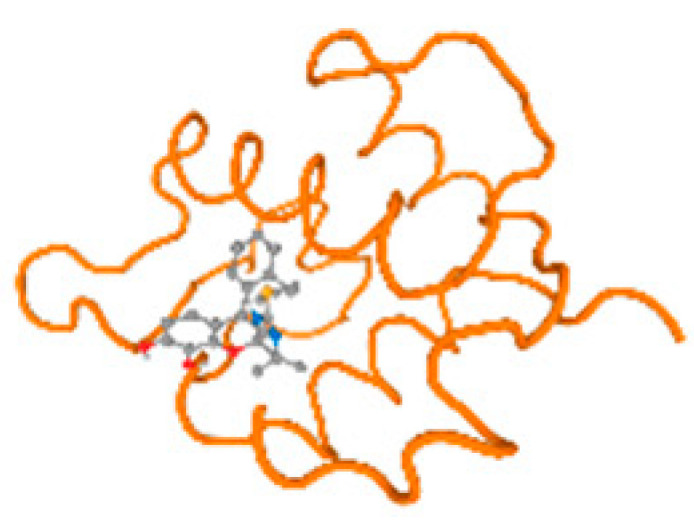	−8.64 (Lig. Pos. 3/100) 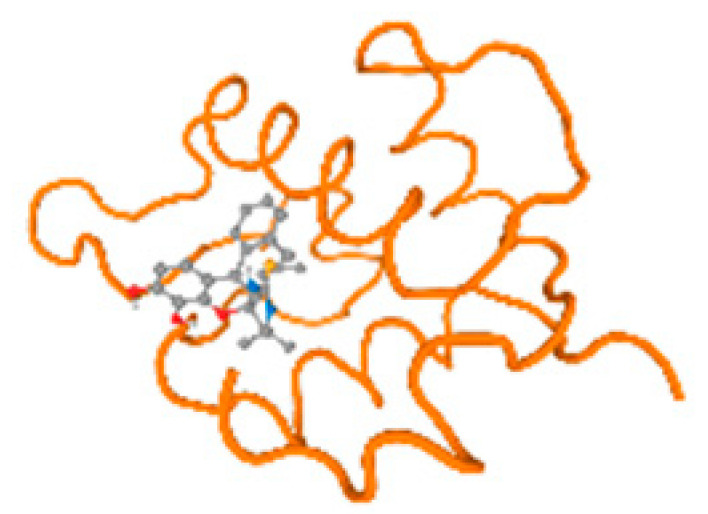	−8.71 (Lig. Pos. 3/100) 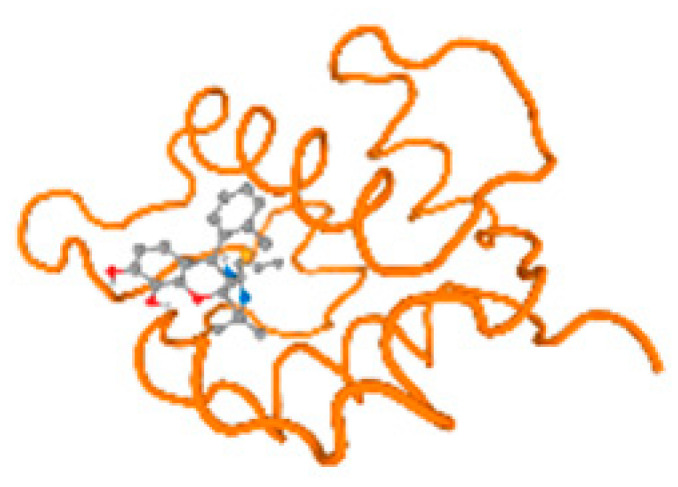
		3T4G	−8.69 (Lig. Pos. 3/90) 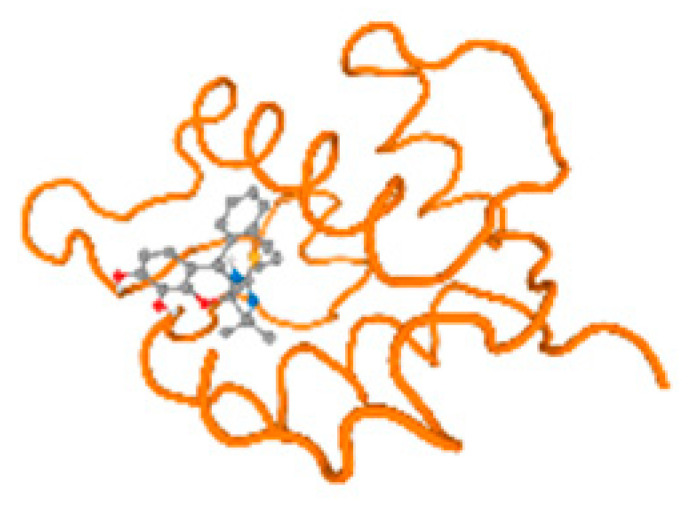	−8.72 (Lig. Pos. 3/70) 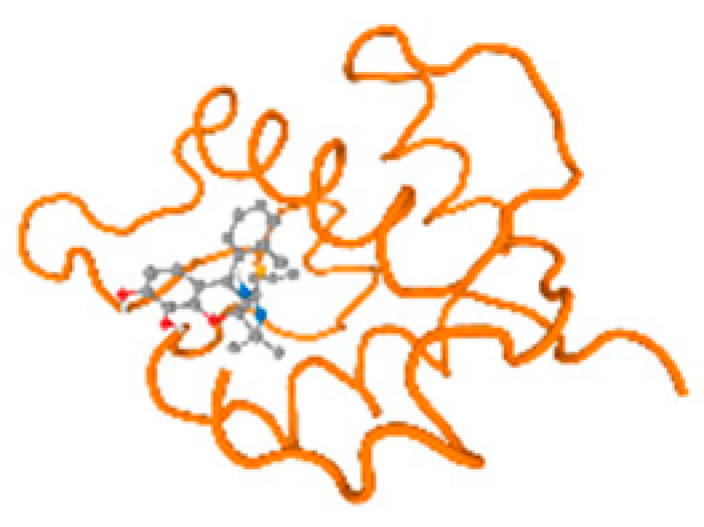	−8.7 (Lig. Pos. 3/90) 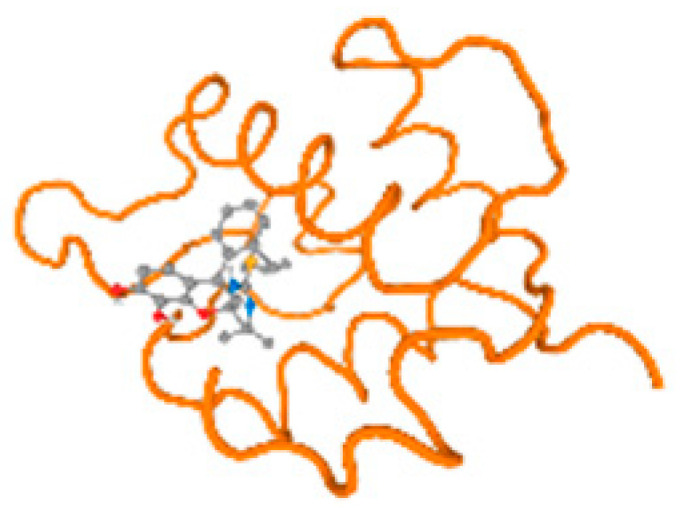	−8.72 (Lig. Pos. 3/100) 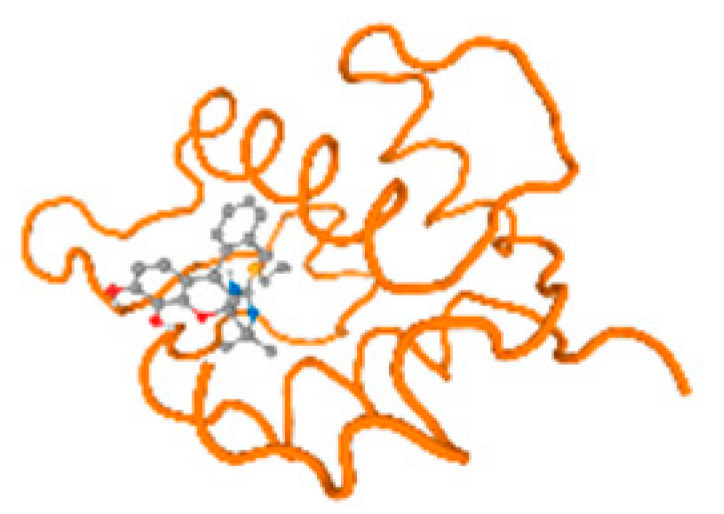
	Target Receptor	3Q7G	−8.46 (Lig. Pos. 3/100) 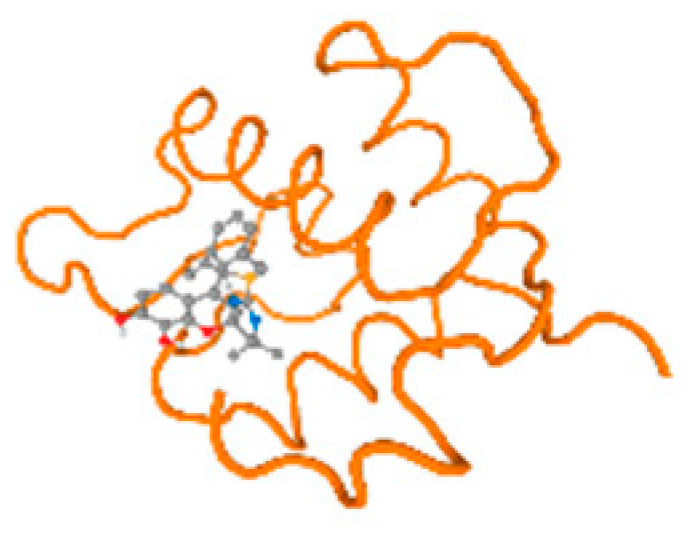	−8.64 (Lig. Pos. 3/100) 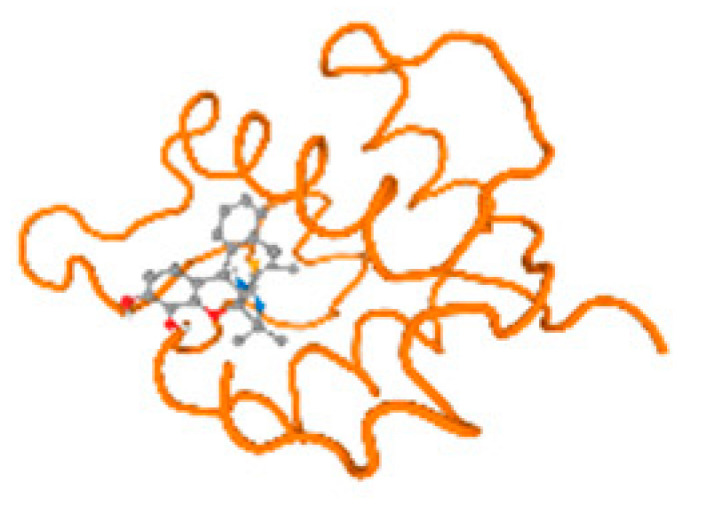	−8.71 (Lig. Pos. 3/100) 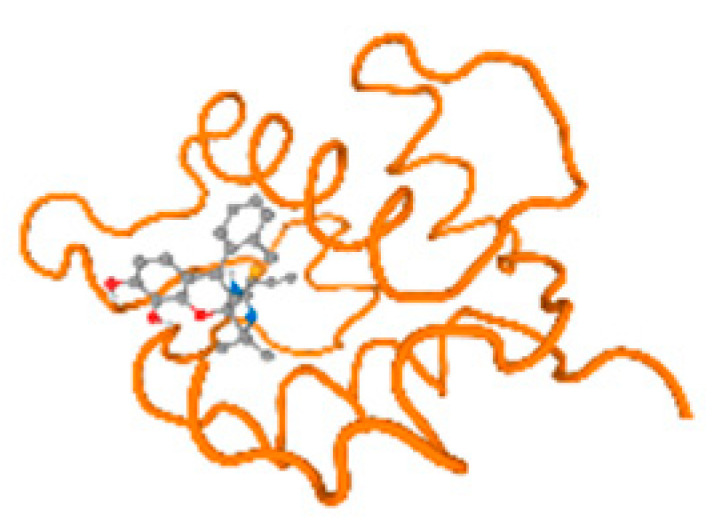	−8.48 (Lig. Pos. 3/100) 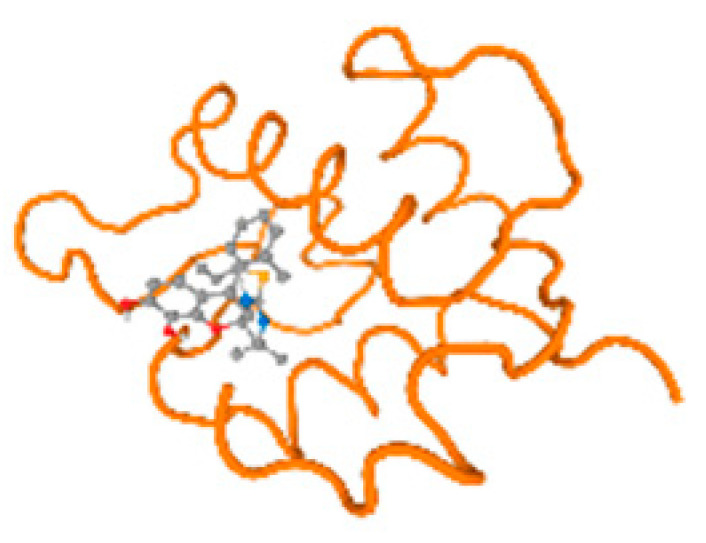
ATPase	Target Receptor	6WLW	−8.46 (Lig. Pos. 3/100) 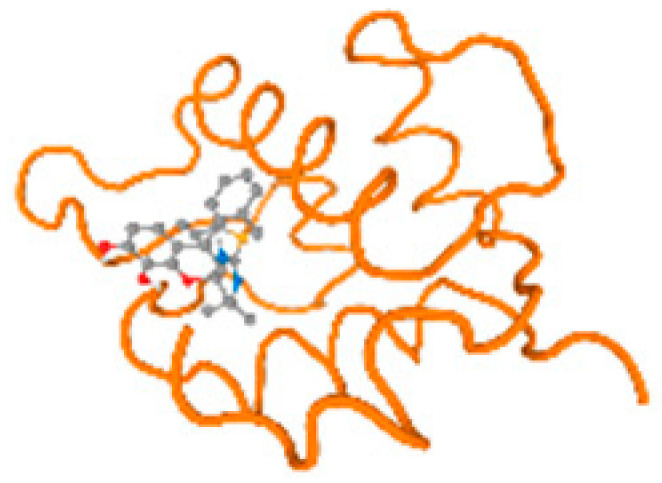	−8.71 (Lig. Pos. 3/100) 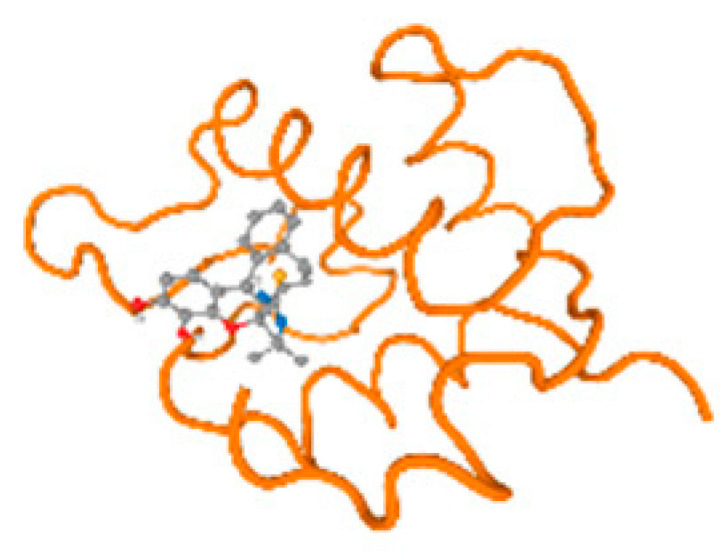	−7.15 (Lig. Pos. 10/30) 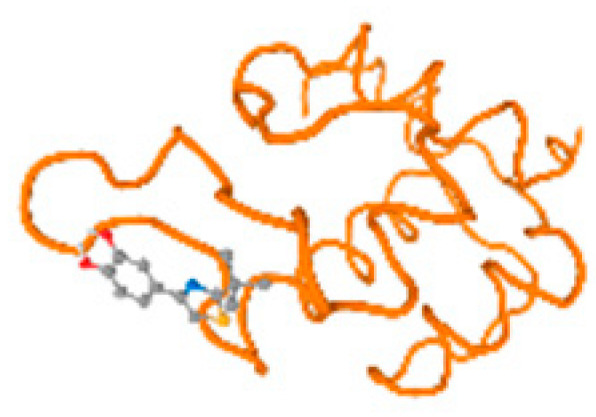	−8.68 (Lig. Pos. 3/90) 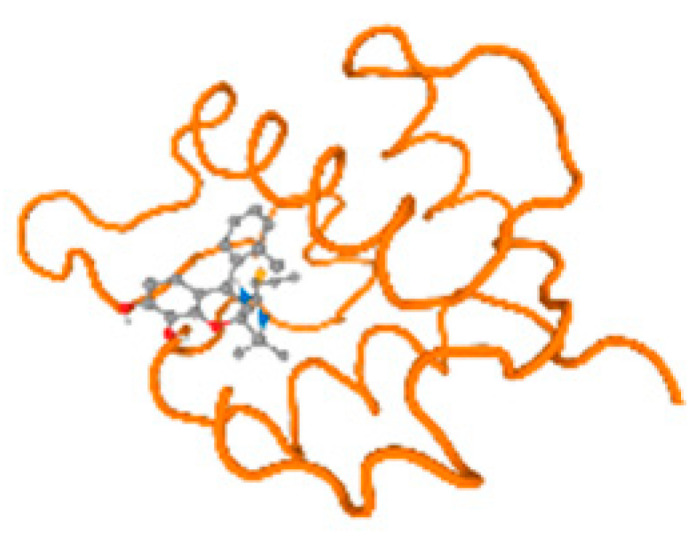
		6WM3	−7.21 (Lig. Pos. 10/20) 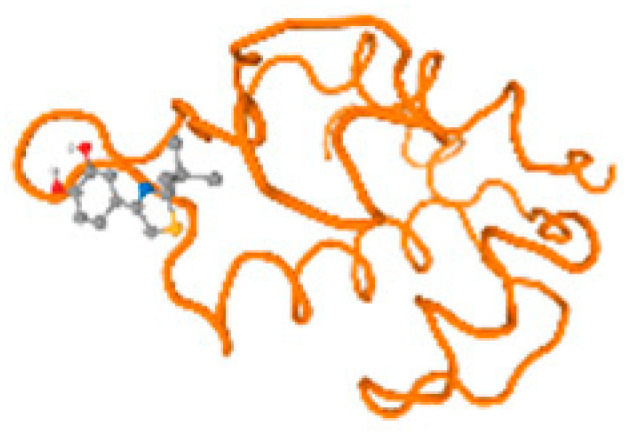	−8.64 (Lig. Pos. 3/100) 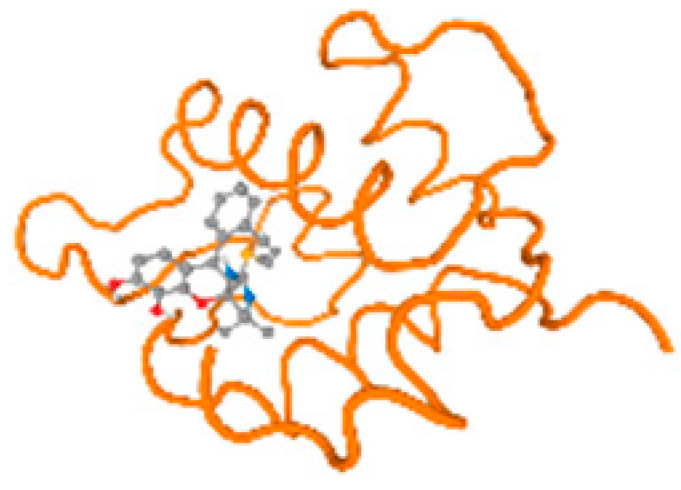	−8.69 (Lig. Pos. 3/90) 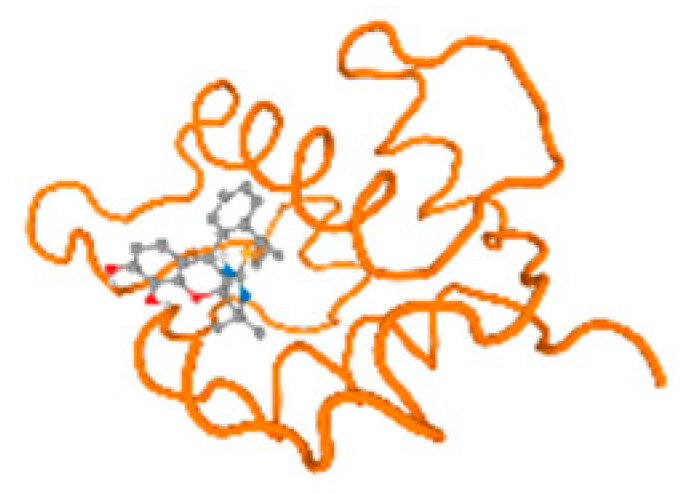	−8.47 (Lig. Pos. 3/100) 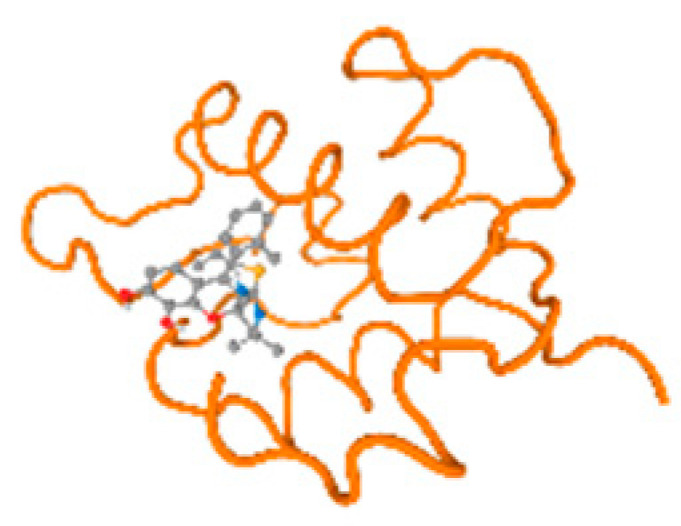
Albumin	Target Protein	5UJB	−7.62 (Lig. Pos. 9/30) 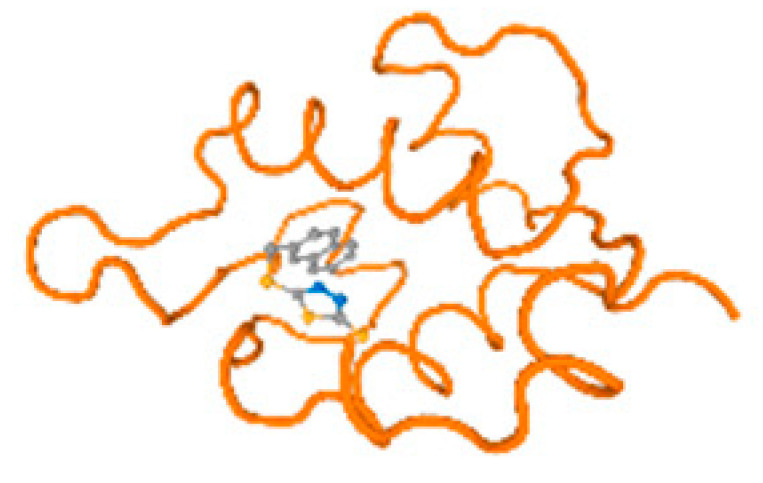	−8.68 (Lig. Pos. 3/100) 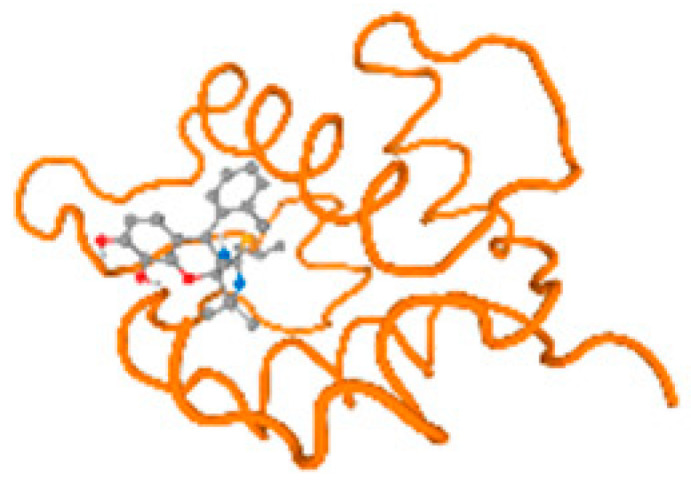	−8.69 (Lig. Pos. 3/100) 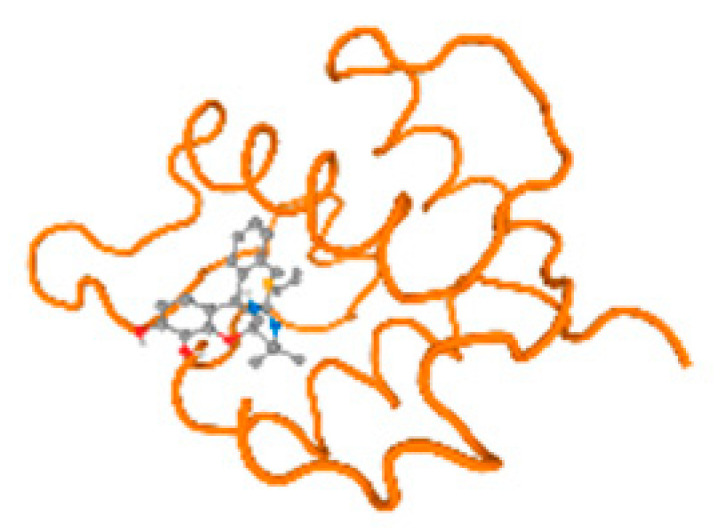	−8.68 (Lig. Pos. 3/80) 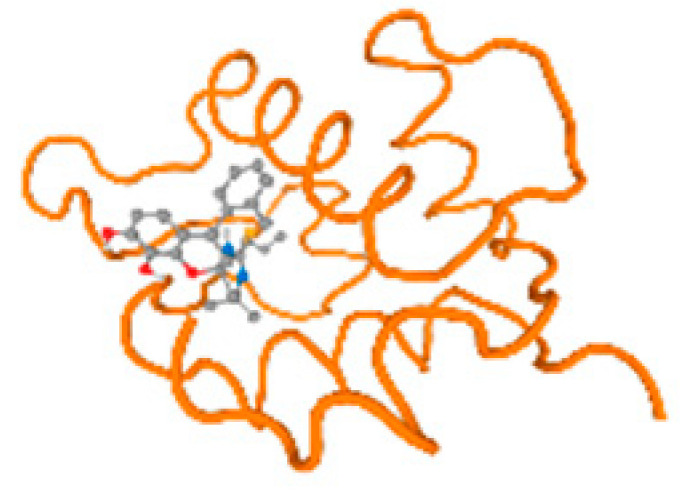
		6M5D	−8.72 (Lig. Pos. 3/100) 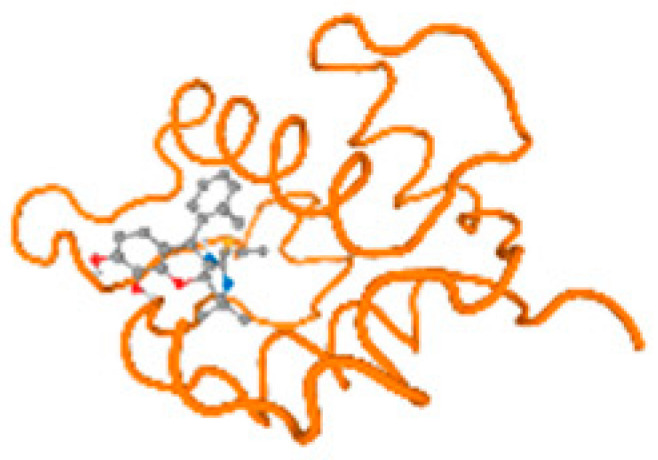	−8.68 (Lig. Pos. 3/100) 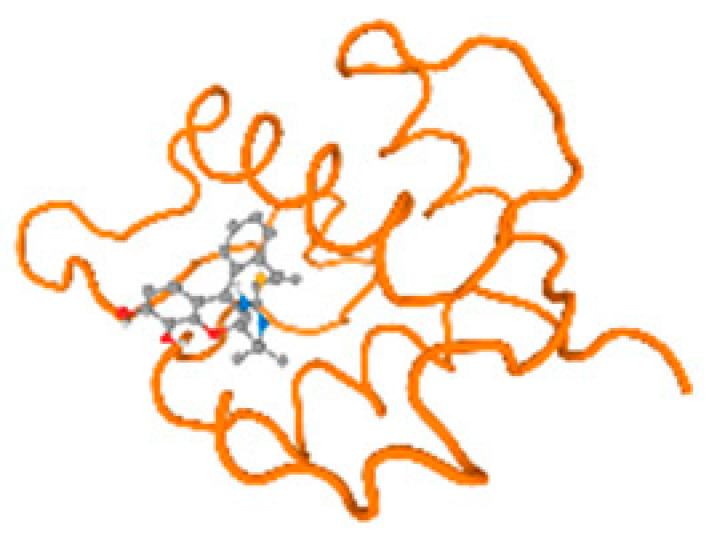	−8.68 (Lig. Pos. 3/100) 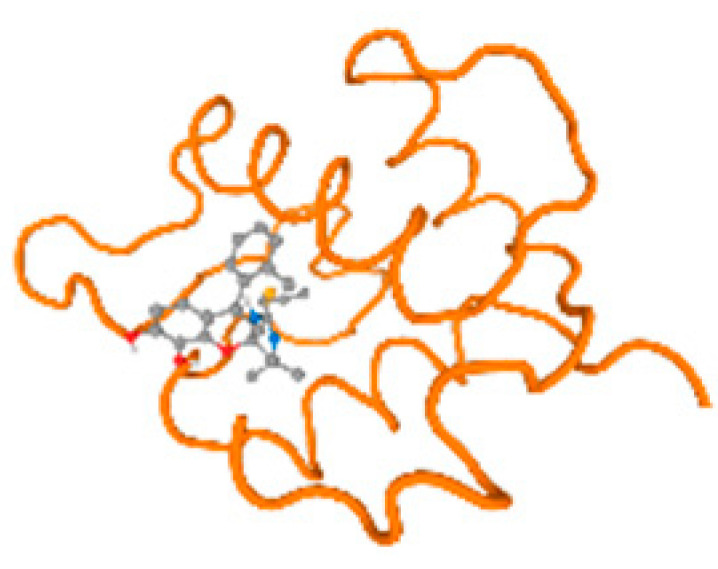	−8.62 (Lig. Pos. 3/90) 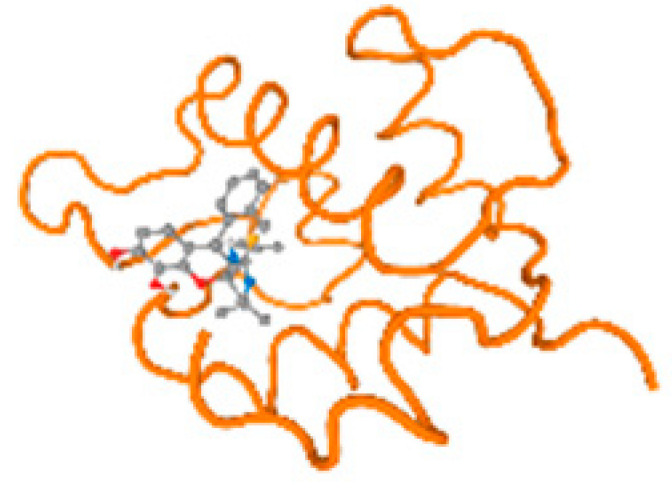
	Target Receptor	6HSC	−8.67 (Lig. Pos. 3/100) 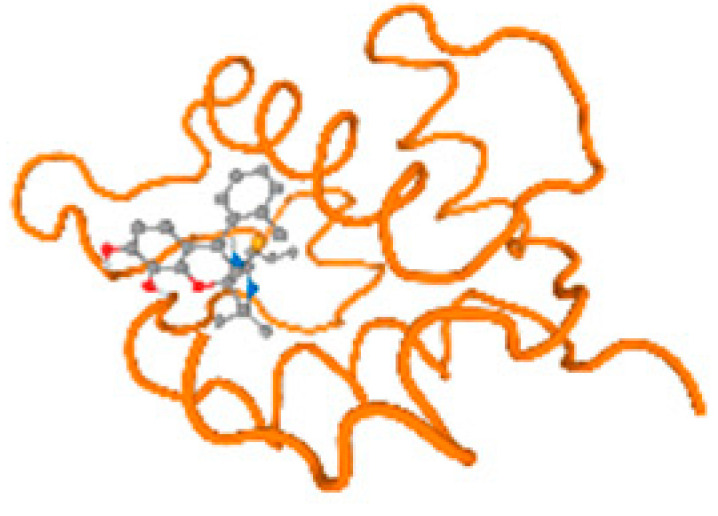	−8.71 (Lig. Pos. 3/100) 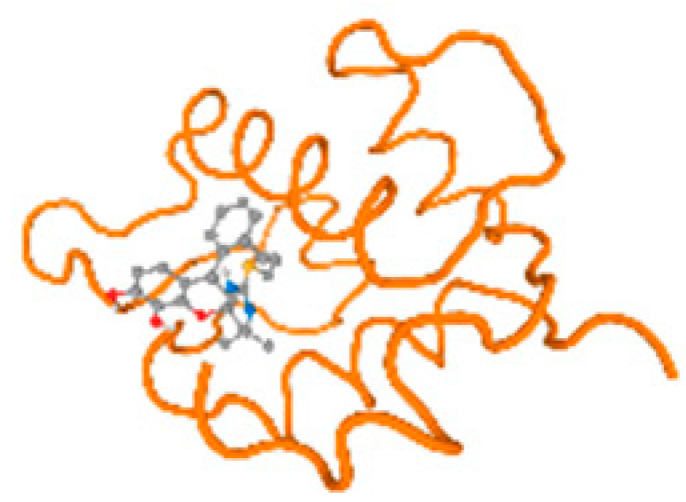	−8.68 (Lig. Pos. 3/100) 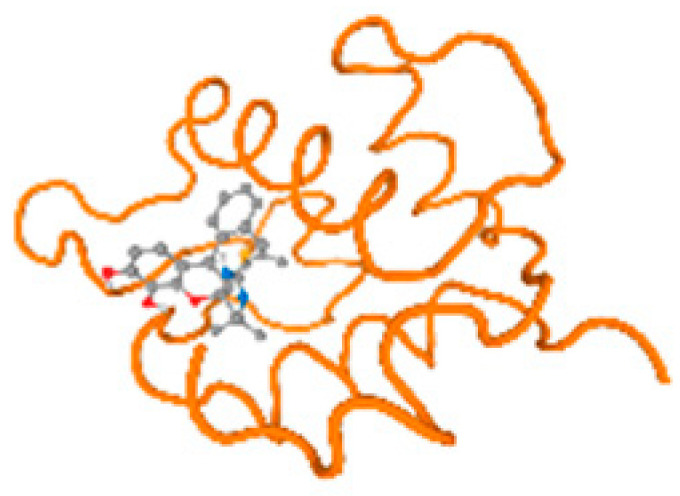	−8.71 (Lig. Pos. 3/80) 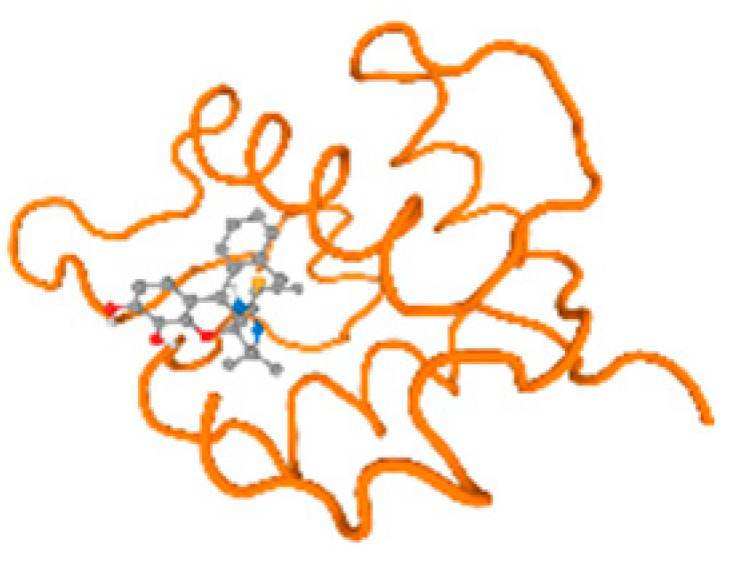
		2ESG	−8.67 (Lig. Pos. 3/100) 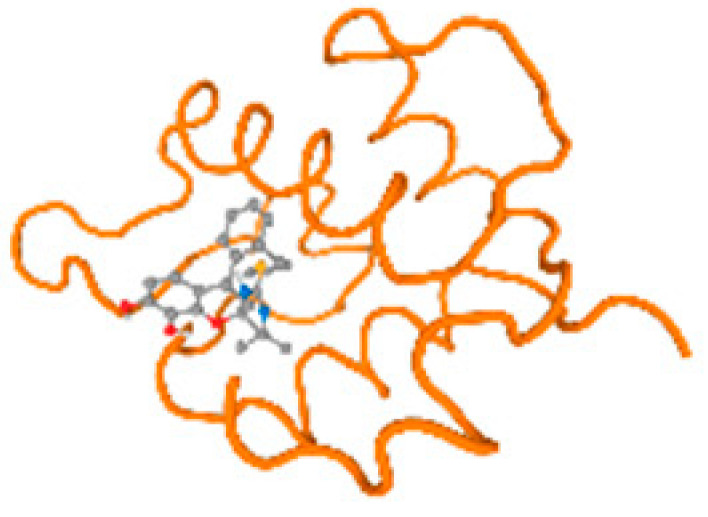	−8.46 (Lig. Pos. 3/90) 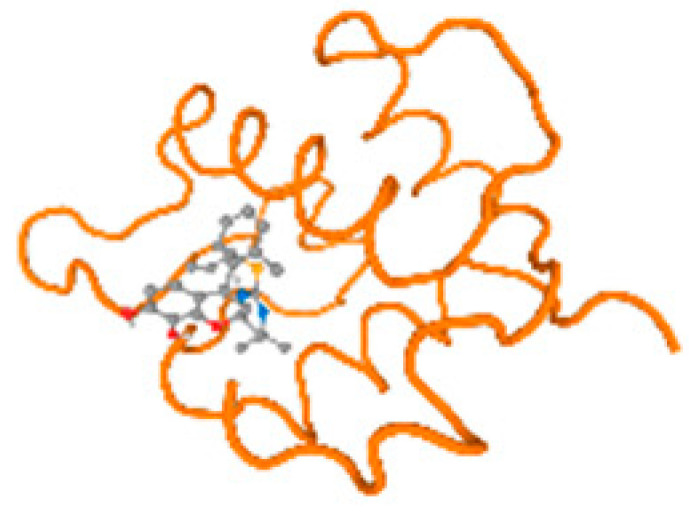	−8.71 (Lig. Pos. 3/100) 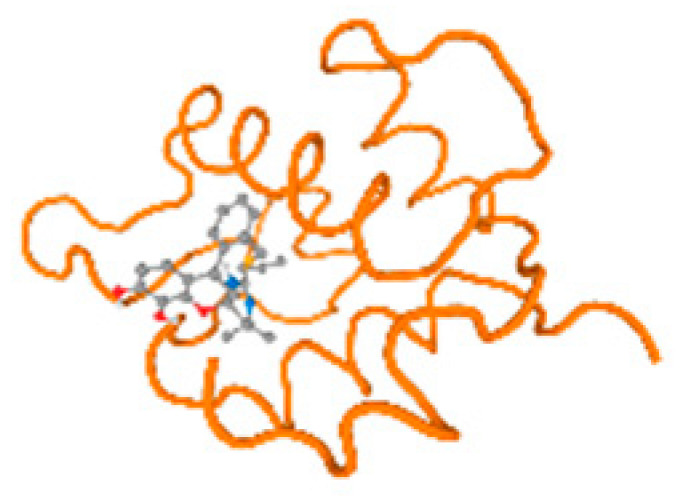	−8.47 (Lig. Pos. 3/100) 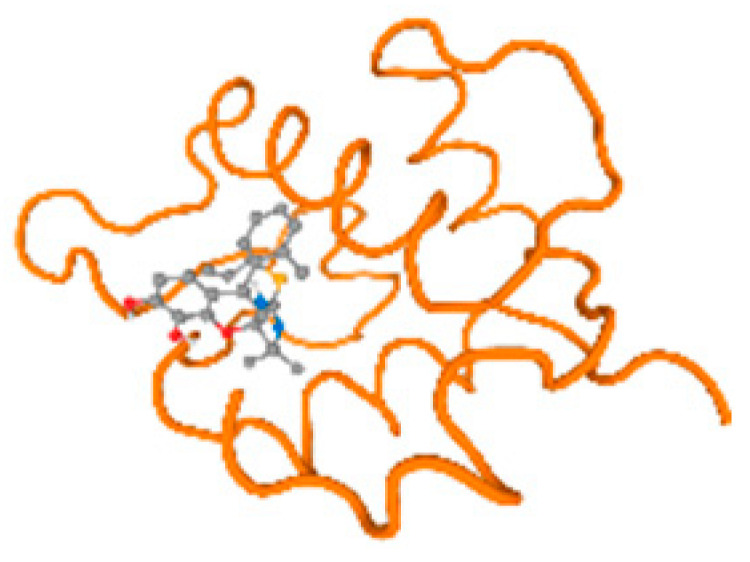
		6YG9	−8.72 (Lig. Pos. 3/100) 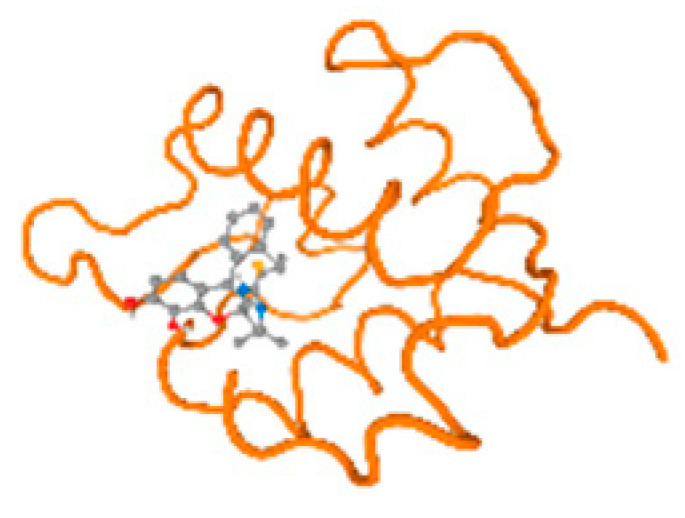	−8.7 (Lig. Pos. 3/80) 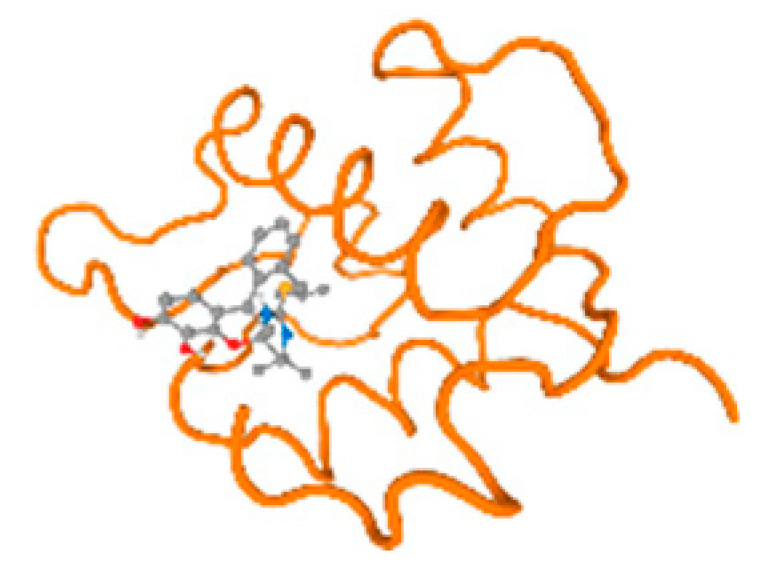	−8.64 (Lig. Pos. 3/100) 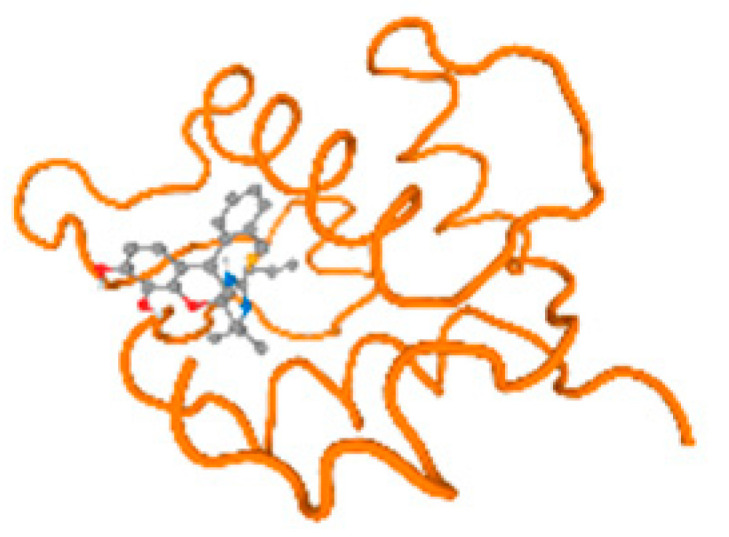	−8.71 (Lig. Pos. 3/80) 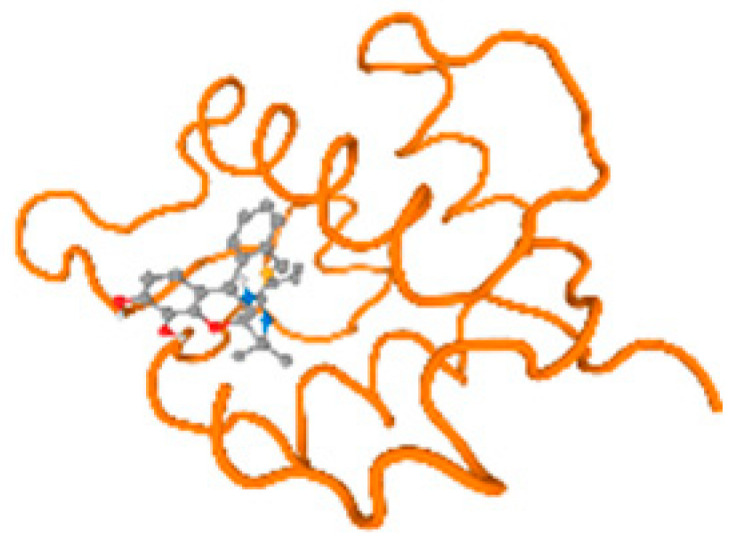
Mono amino oxidase (MAO)	Target Protein	2BK3	−8.42 (Lig. Pos. 3/100) 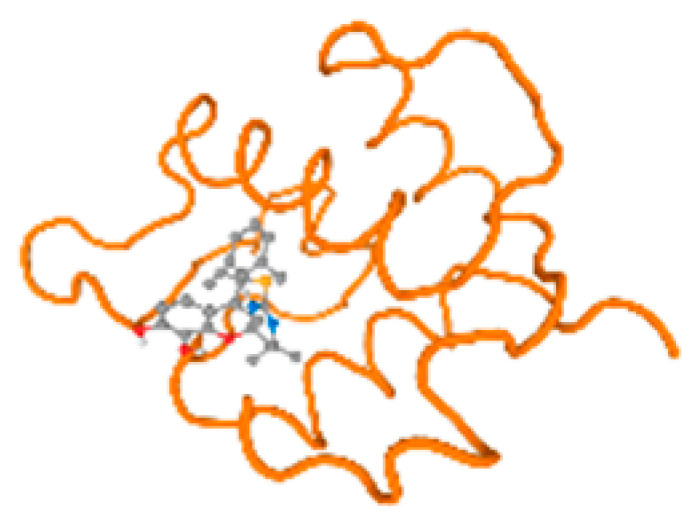	−8.7 (Lig. Pos. 3/100) 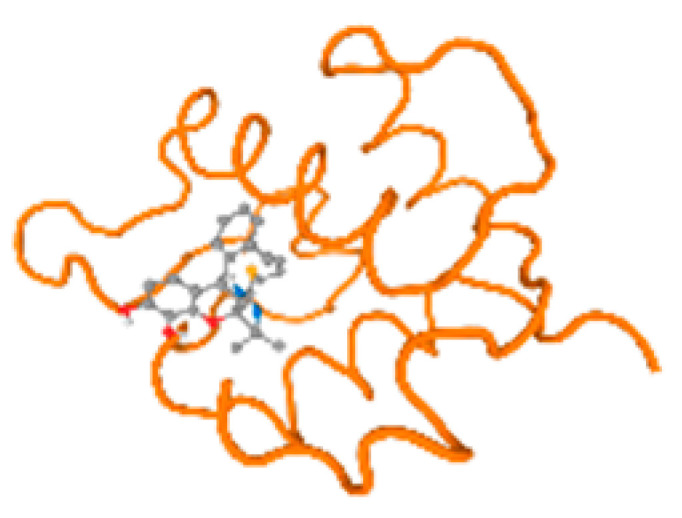	−8.72 (Lig. Pos. 3/100) 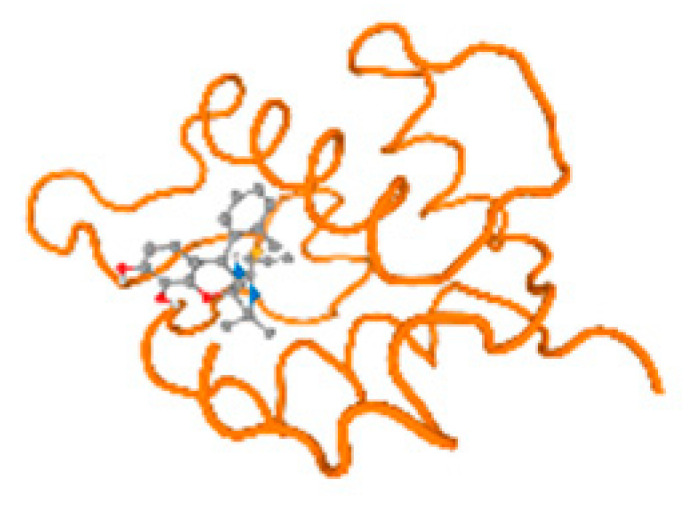	−8.71 (Lig. Pos. 3/100) 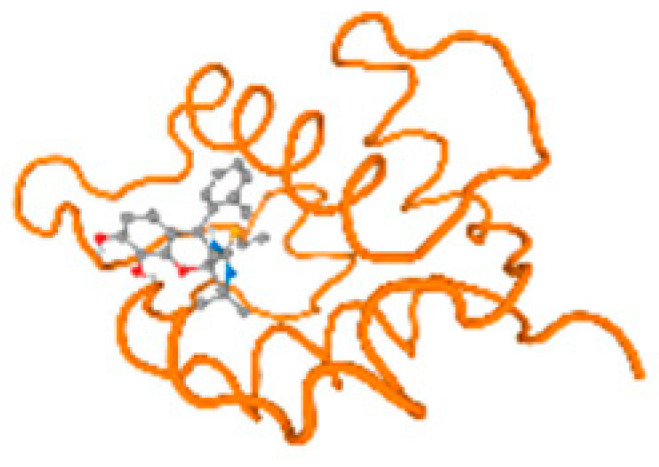
		2BXS	−8.72 (Lig. Pos. 3/100) 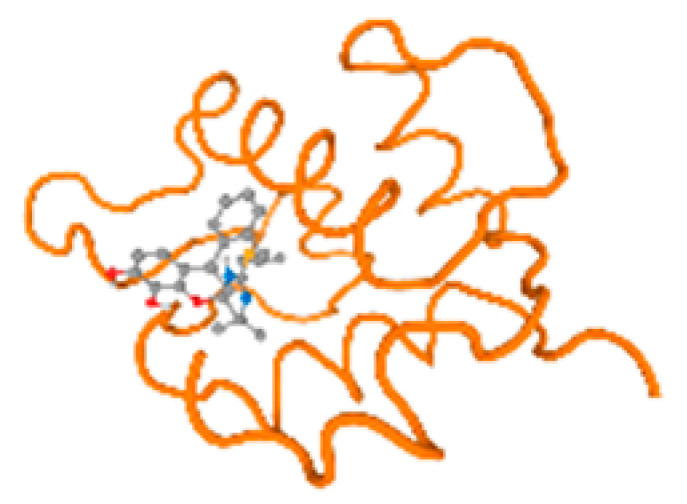	−8.63 (Lig. Pos. 3/100) 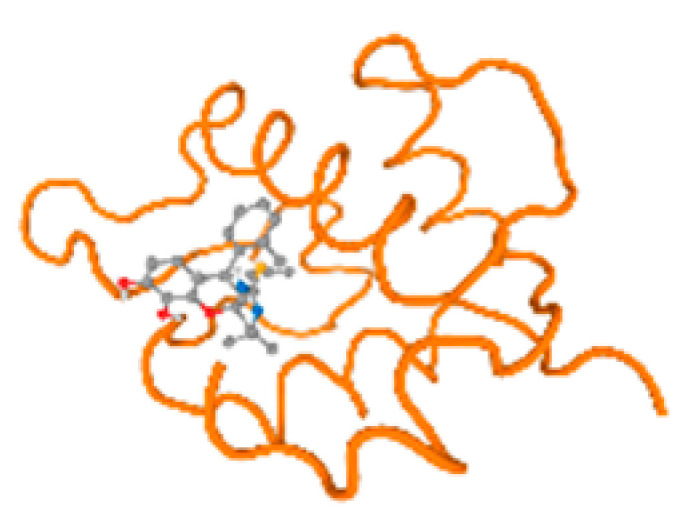	−8.7 (Lig. Pos. 3/100) 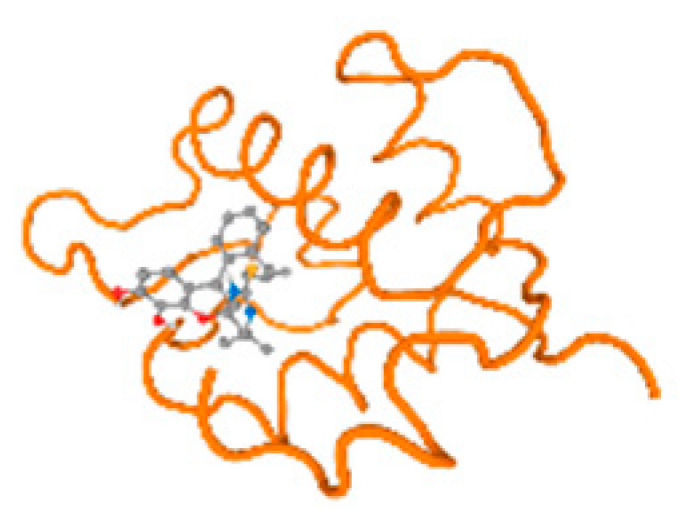	−8.41 (Lig. Pos. 3/100) 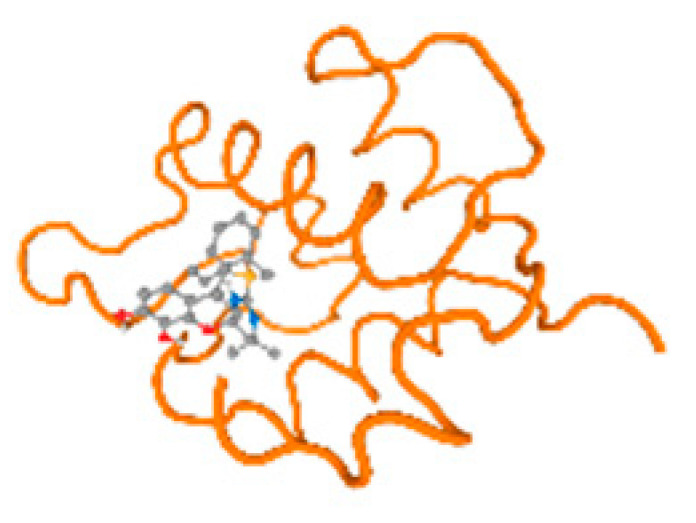
		7DJU	−8.71 (Lig. Pos. 3/100) 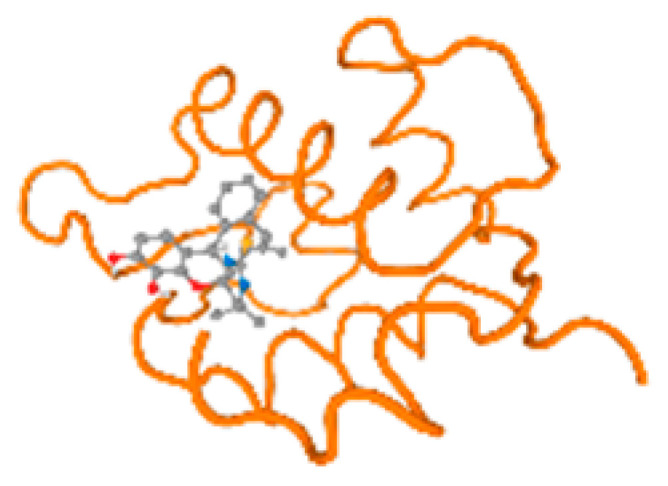	−8.47 (Lig. Pos. 3/100) 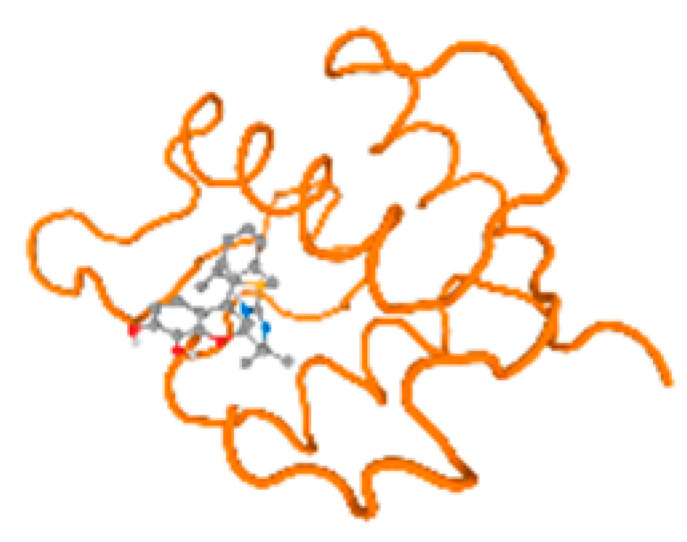	−8.41 (Lig. Pos. 3/80) 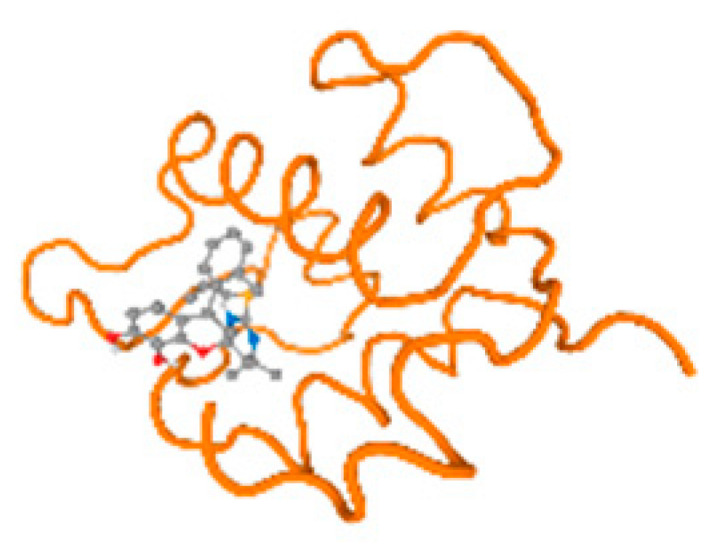	−8.45 (Lig. Pos. 3/100) 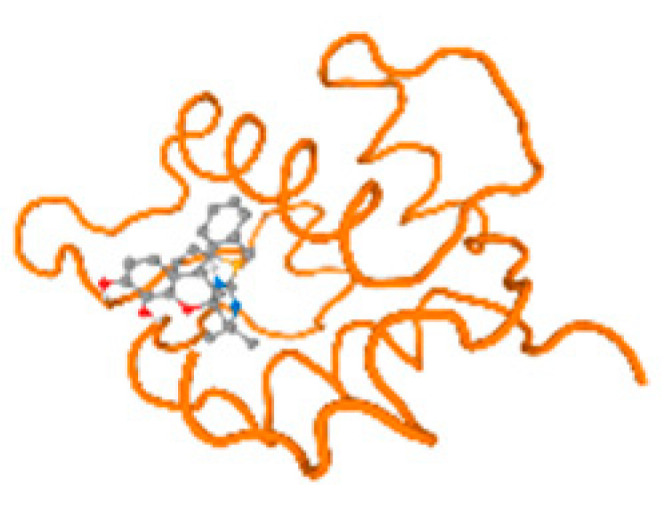
	Target Receptor	7EL7	−8.65 (Lig. Pos. 3/70) 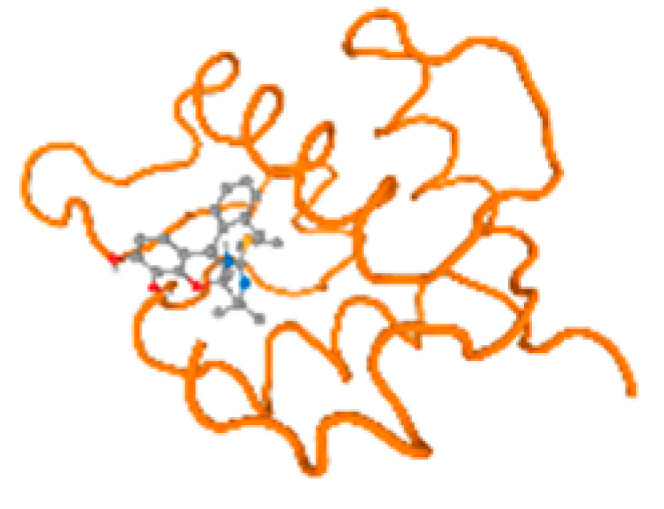	−8.68 (Lig. Pos. 3/100) 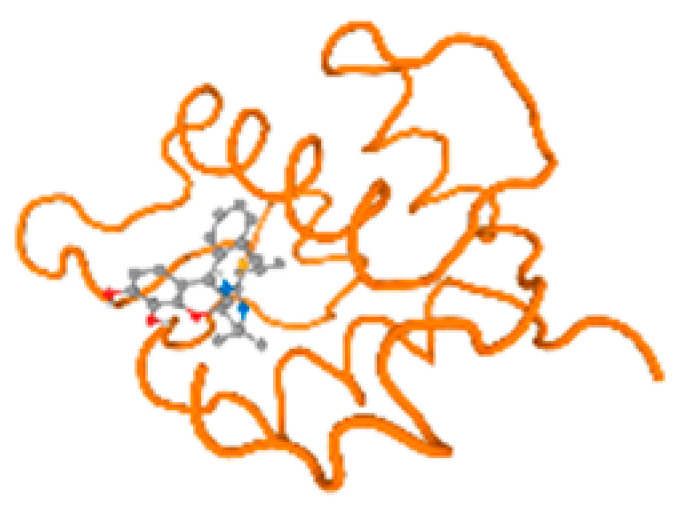	−8.68 (Lig. Pos. 3/90) 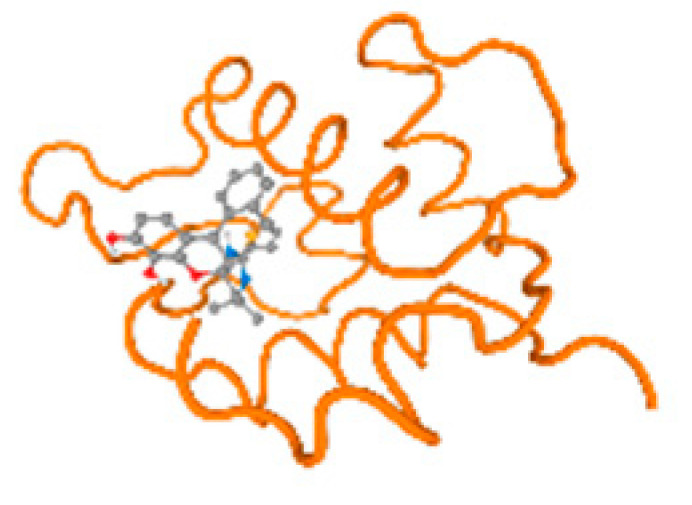	−7.23 (Lig. Pos. 10/20) 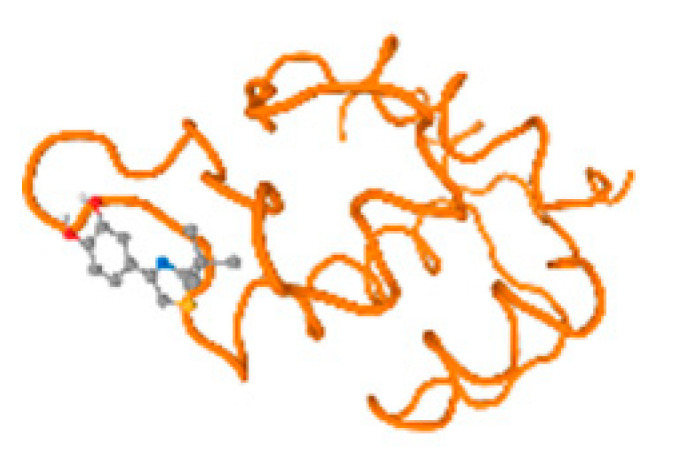
		2VRM	−7.6 (Lig. Pos. 9/50) 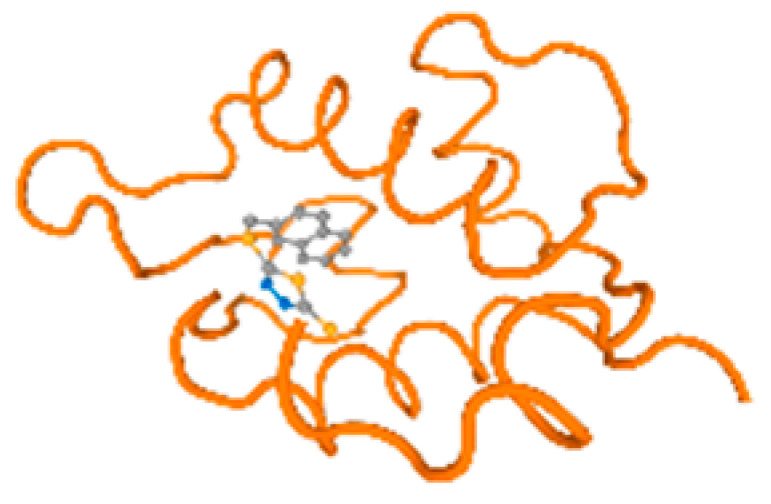	−8.71 (Lig. Pos. 3/30) 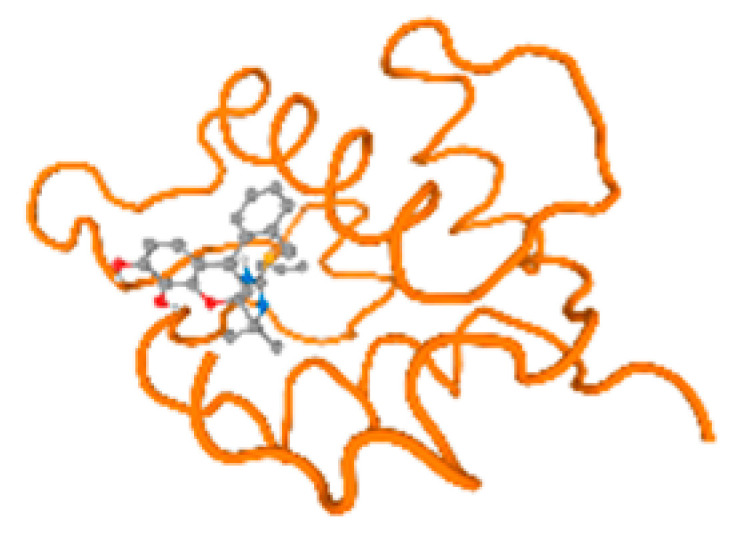	−8.7 (Lig. Pos. 3/30) 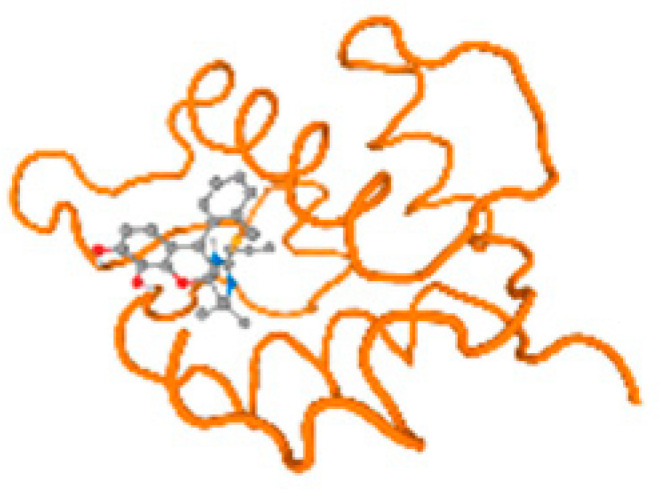	−8.63 (Lig. Pos. 3/70) 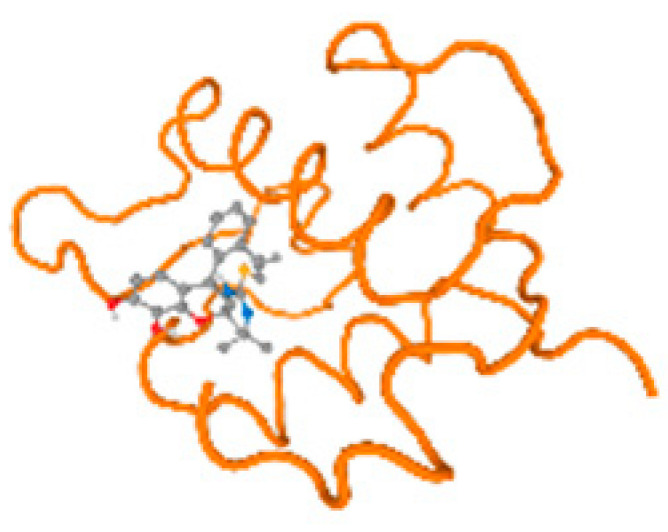
Adrenaline	Target Protein	2HKK	−8.68 (Lig. Pos. 3/100) 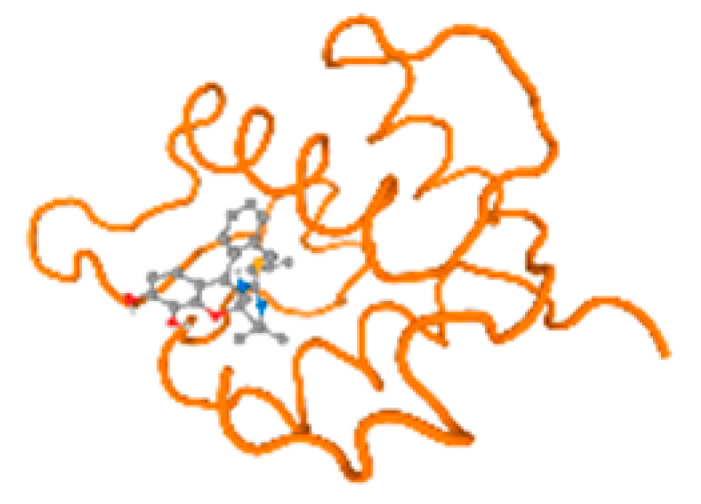	−8.42 (Lig. Pos. 3/100) 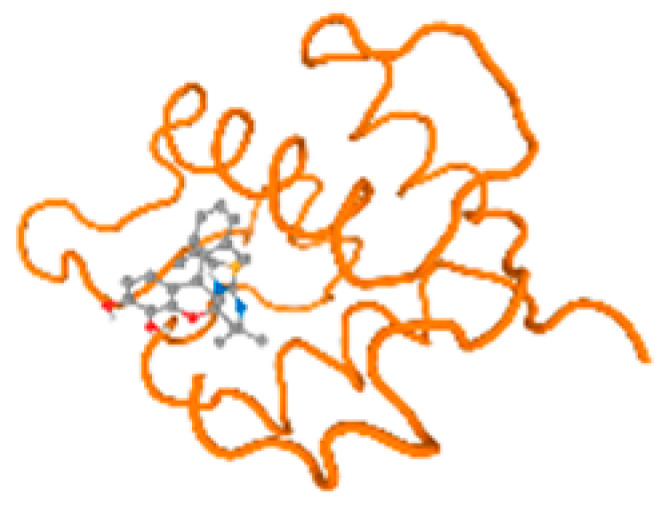	−8.64 (Lig. Pos. 3/50) 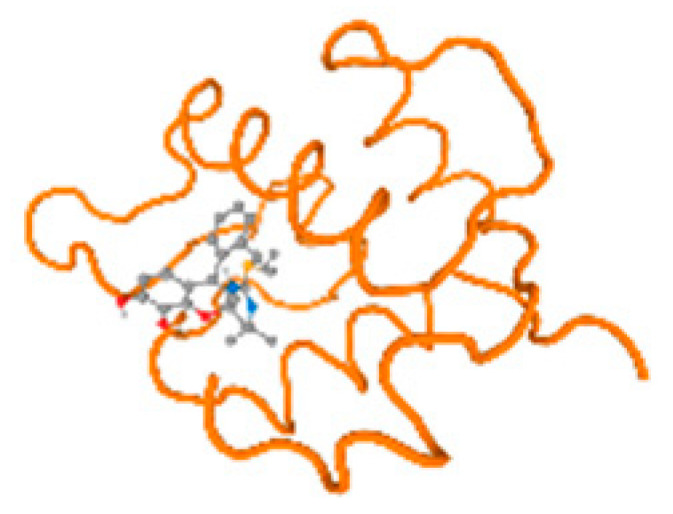	−8.67 (Lig. Pos. 3/100) 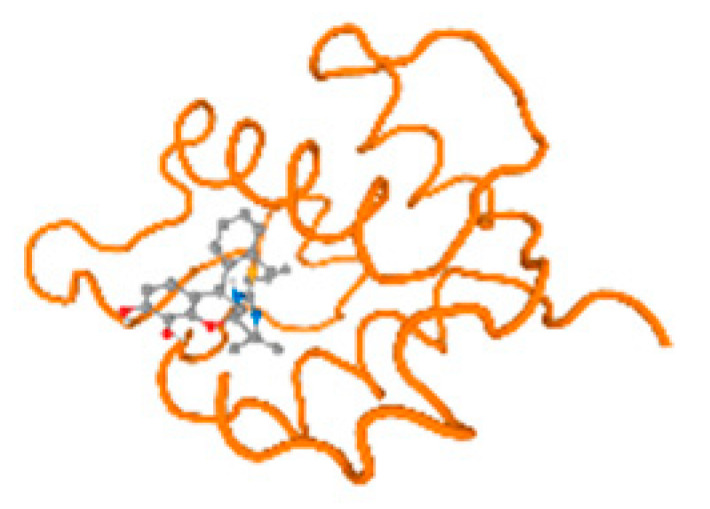
		7BTS	−8.68 (Lig. Pos. 3/100) 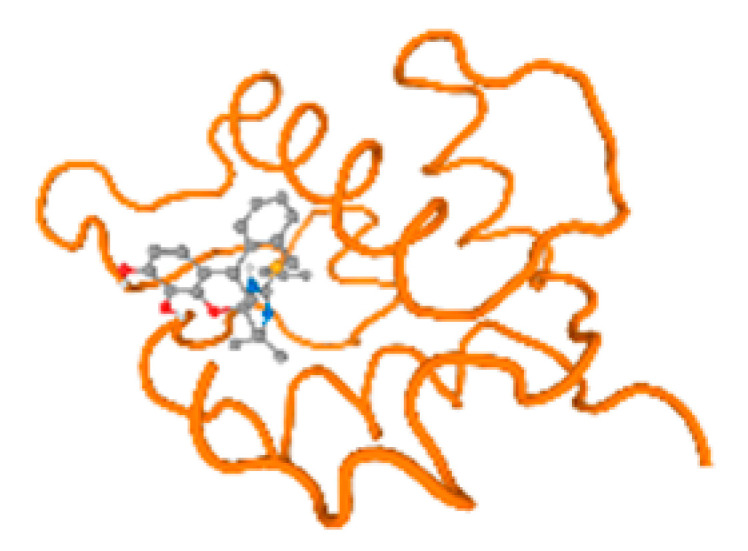	−8.56 (Lig. Pos. 3/100) 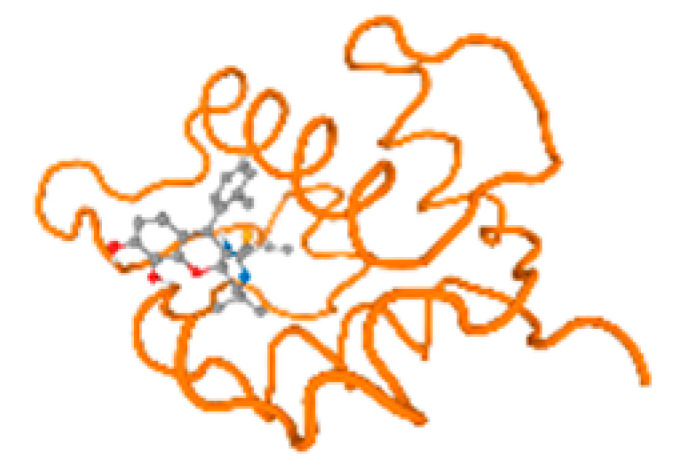	−8.61 (Lig. Pos. 3/100) 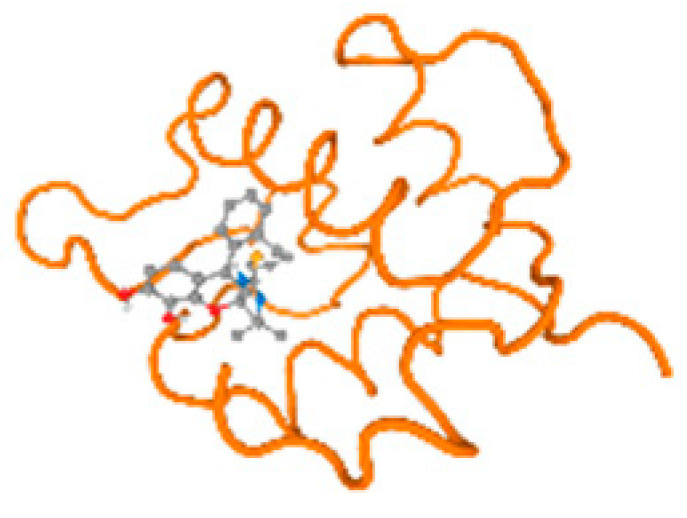	−8.66 (Lig. Pos. 3/100) 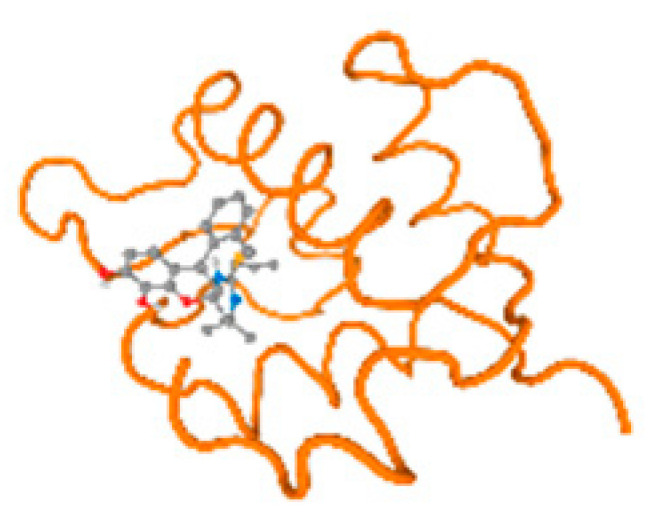
	Target Receptor	2RH1	−8.68 (Lig. Pos. 3/100) 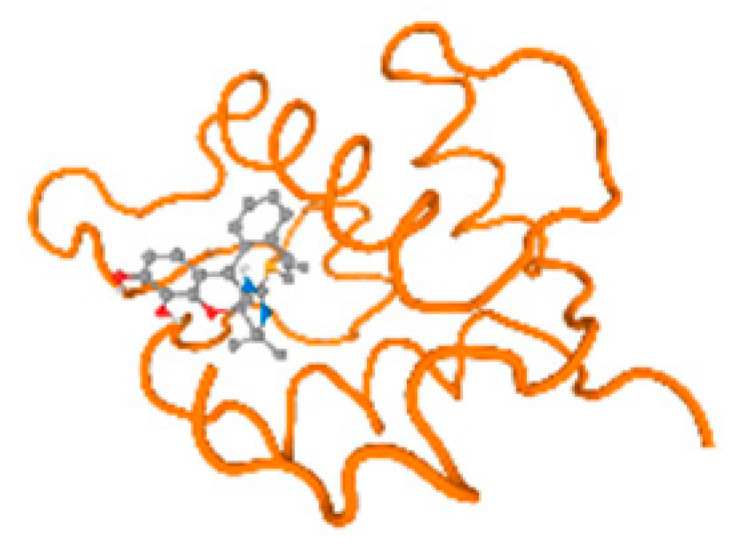	−8.67 (Lig. Pos. 3/100) 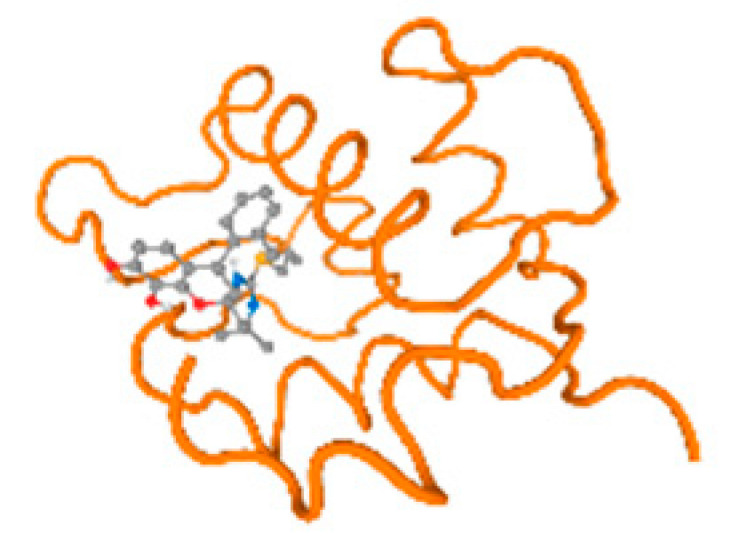	−8.69 (Lig. Pos. 3/100) 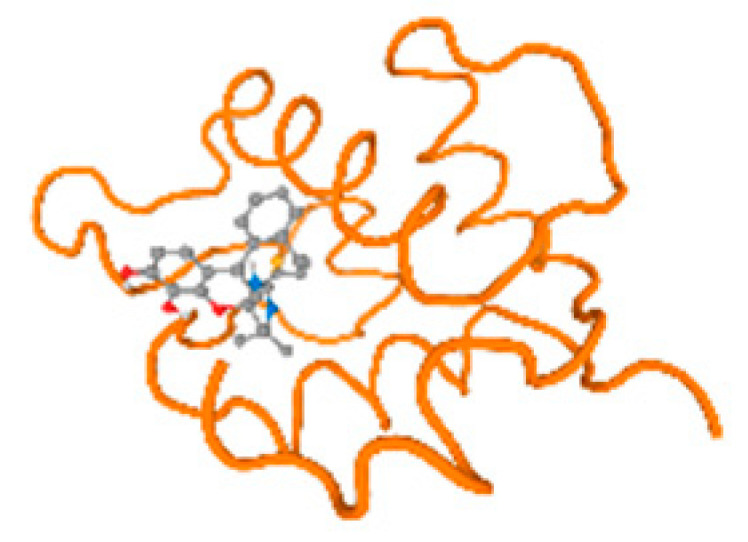	−8.7 (Lig. Pos. 3/100) 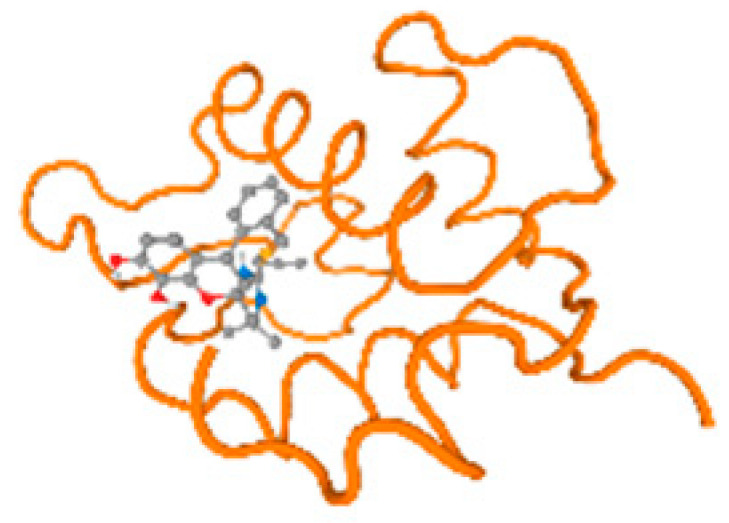
Cortisol	Target Protein	2VDX	−8.66 (Lig. Pos. 3/100) 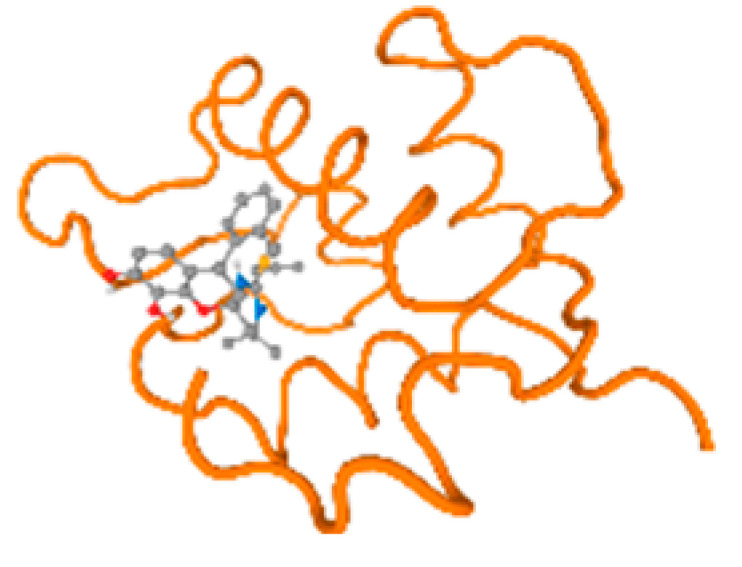	−8.61 (Lig. Pos. 3/70) 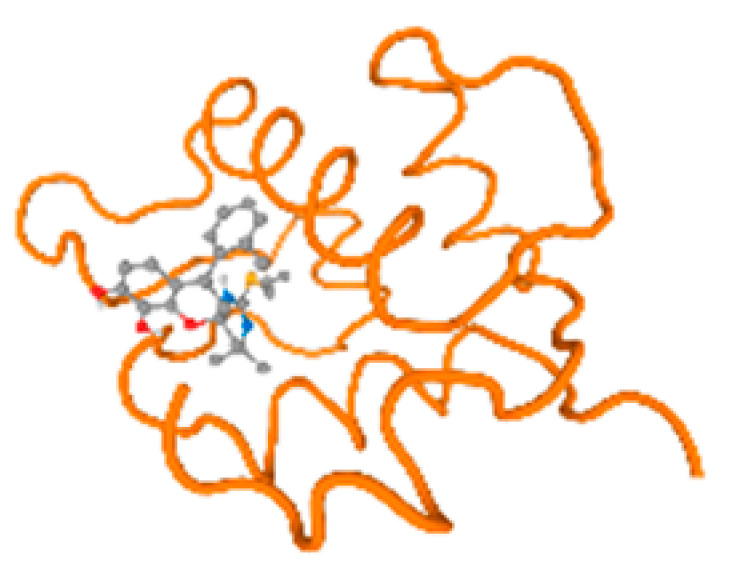	−8.64 (Lig. Pos. 3/100) 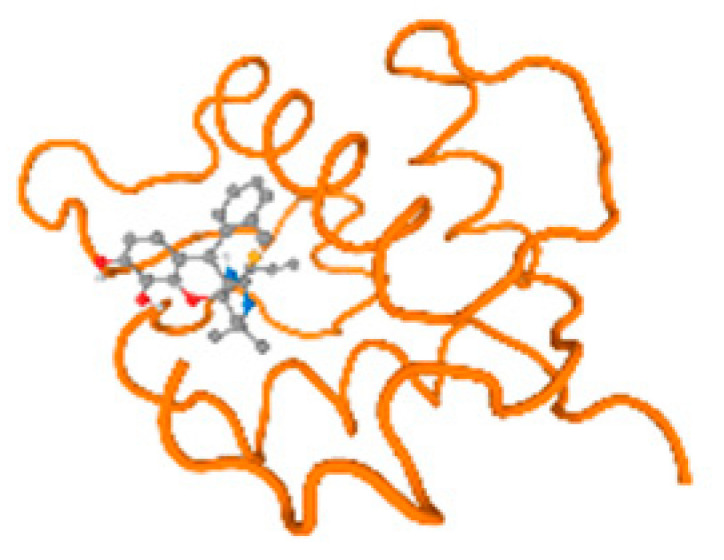	−8.46 (Lig. Pos. 3/100) 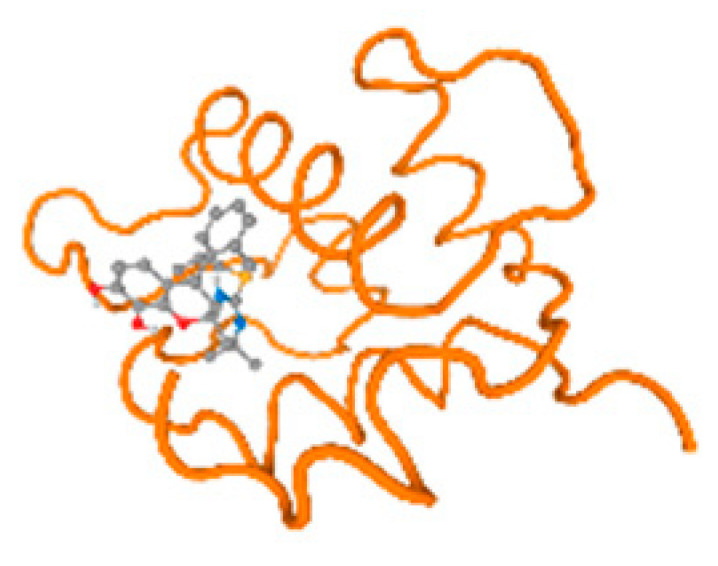
	Target Receptor	2VDY	−8.7 (Lig. Pos. 3/100) 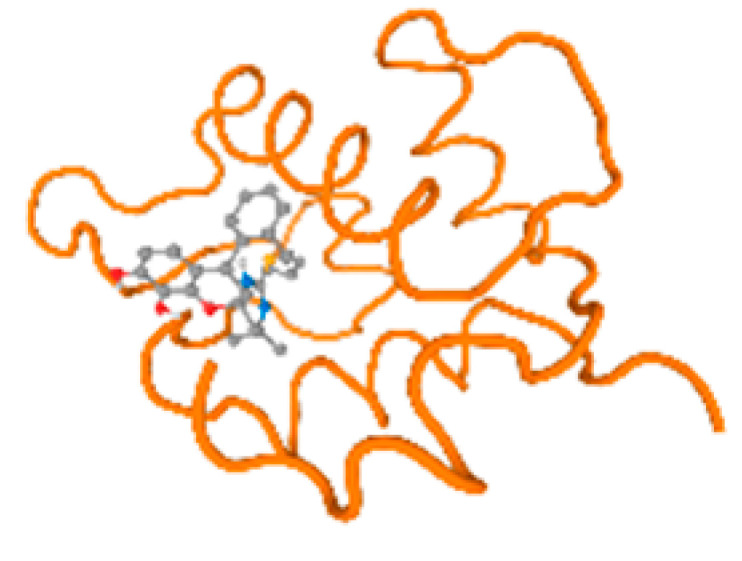	−8.73 (Lig. Pos. 3/100) 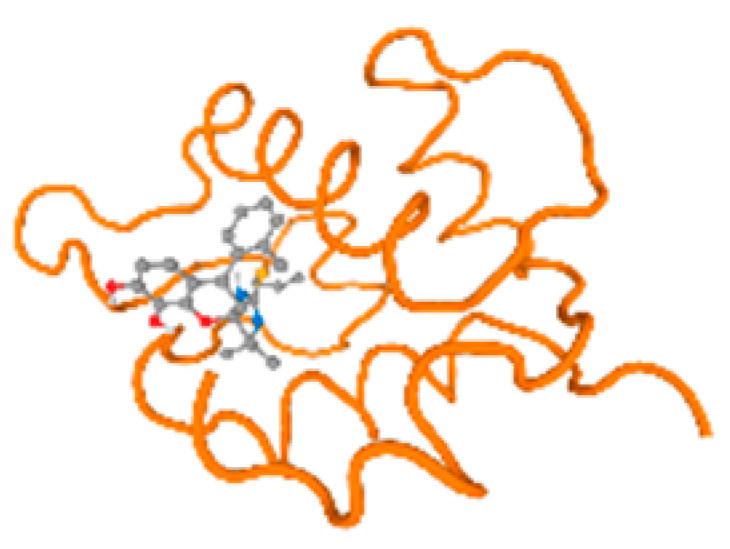	−8.74 (Lig. Pos. 3/100) 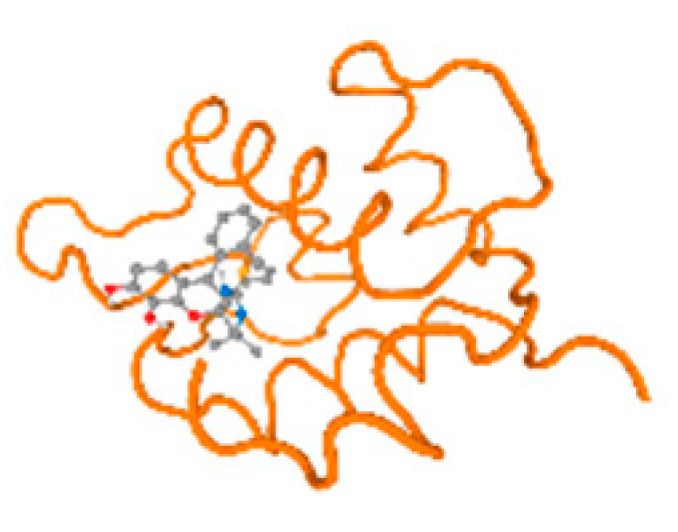	−8.65 (Lig. Pos. 3/100) 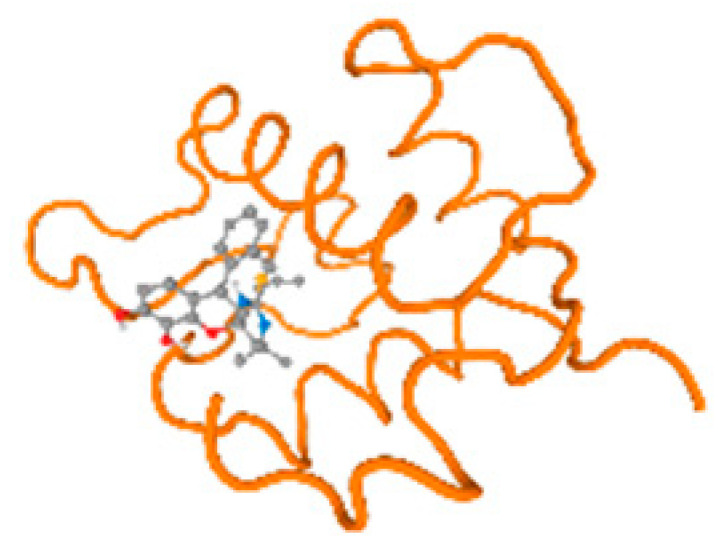
		4P6X	−8.71 (Lig. Pos. 3/90) 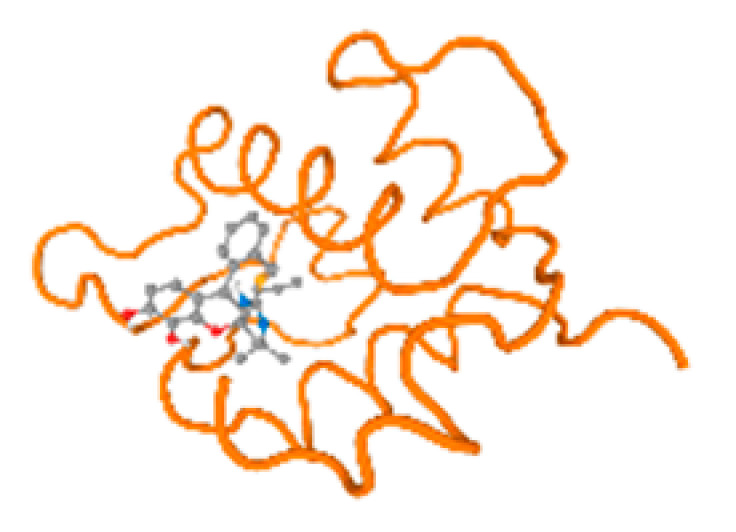	−8.72 (Lig. Pos. 3/100) 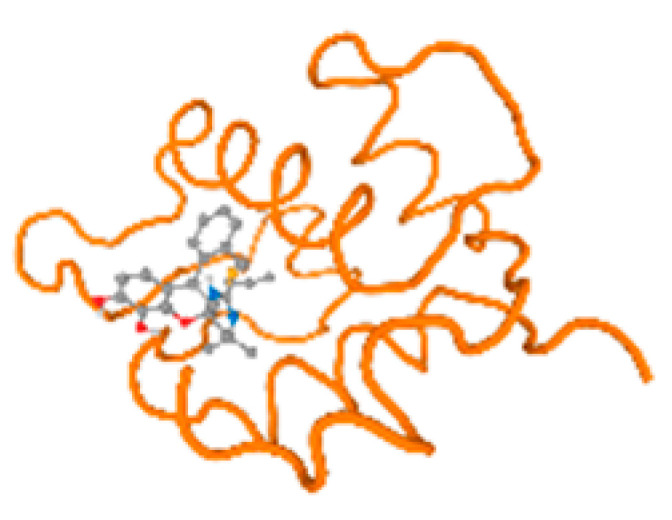	−8.67 (Lig. Pos. 3/100) 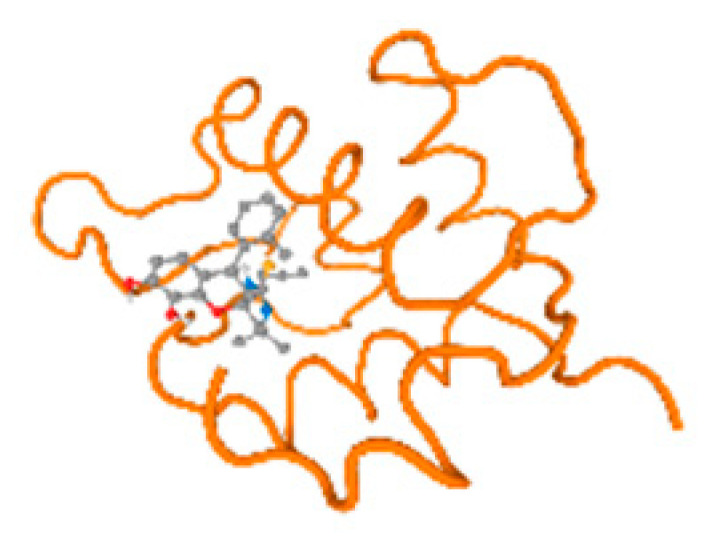	−8.72 (Lig. Pos. 3/100) 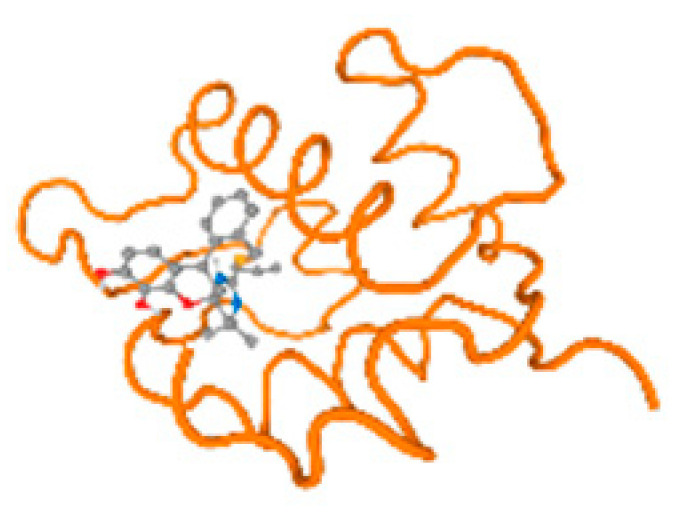
TNF-α	Target Cytokine	6RMJ	−8.47 (Lig. Pos. 3/100) 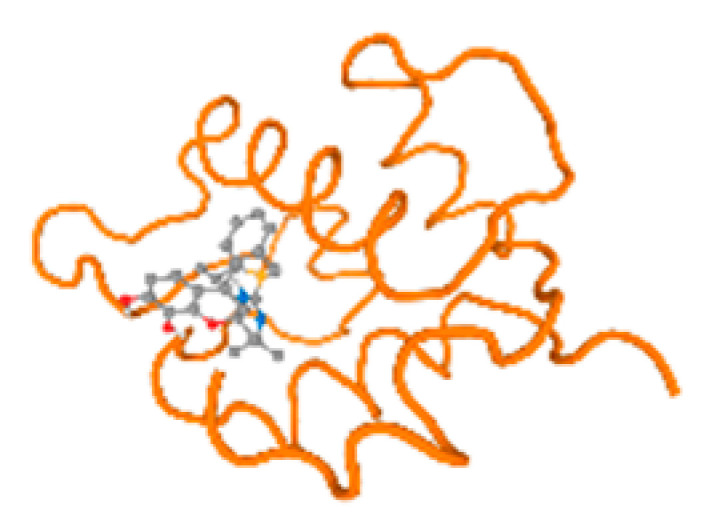	−8.64 (Lig. Pos. 3/100) 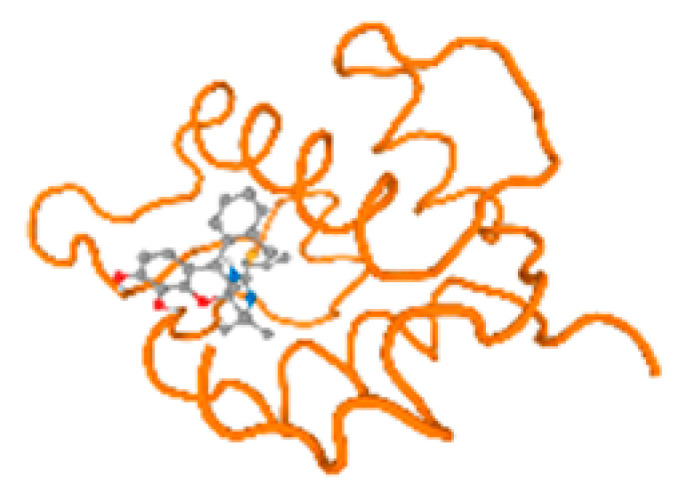	−8.65 (Lig. Pos. 3/100) 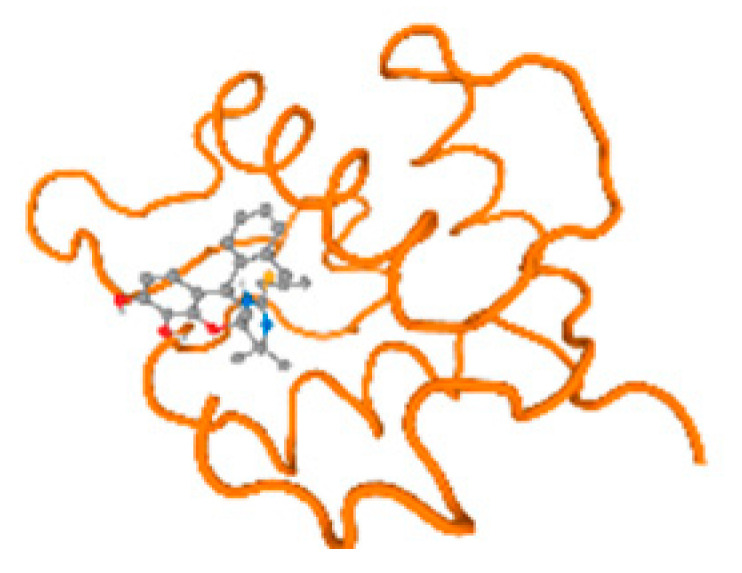	−8.7 (Lig. Pos. 3/100) 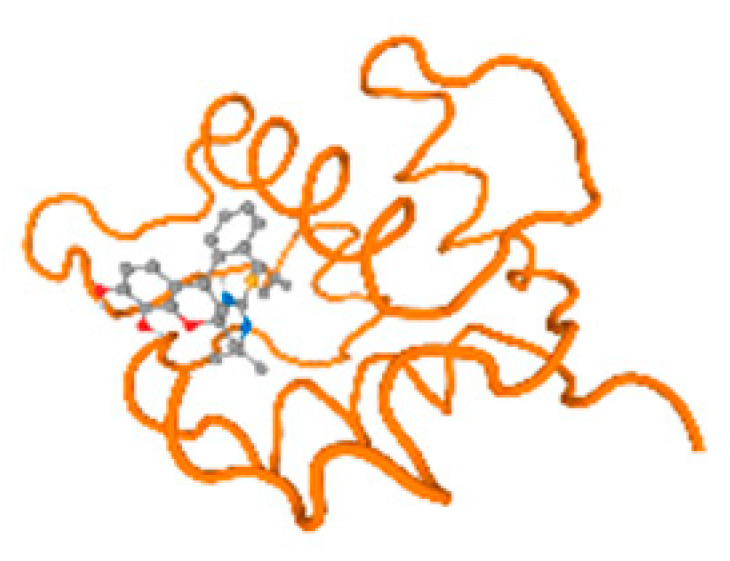
	Target Receptor	5TLJ	−8.73 (Lig. Pos. 3/100) 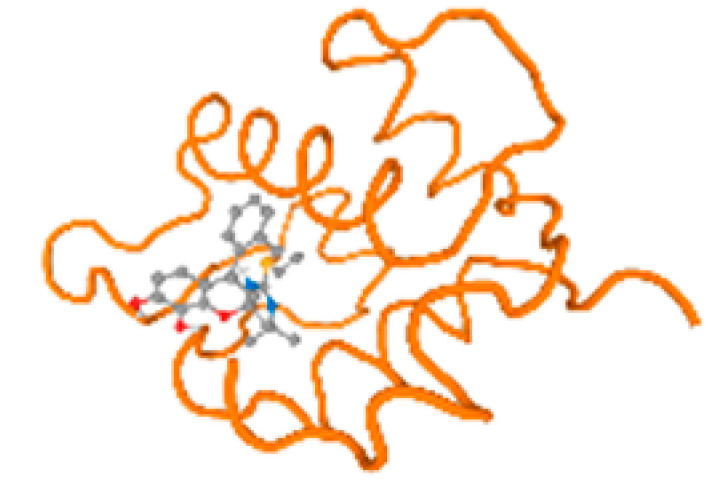	−8.67 (Lig. Pos. 3/100) 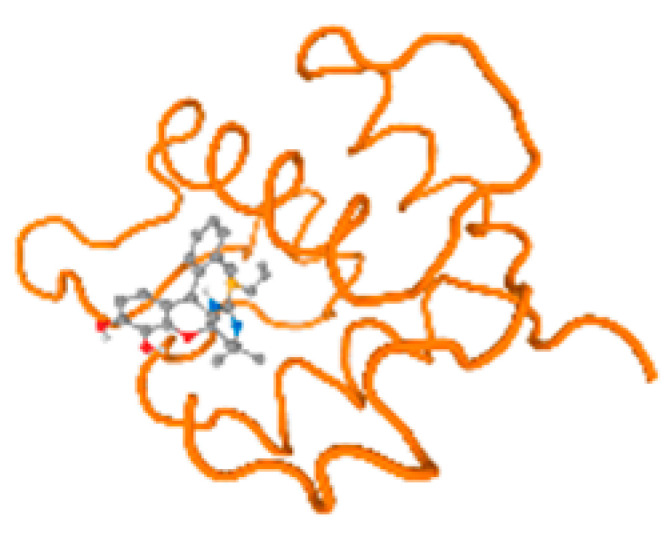	−8.72 (Lig. Pos. 3/100) 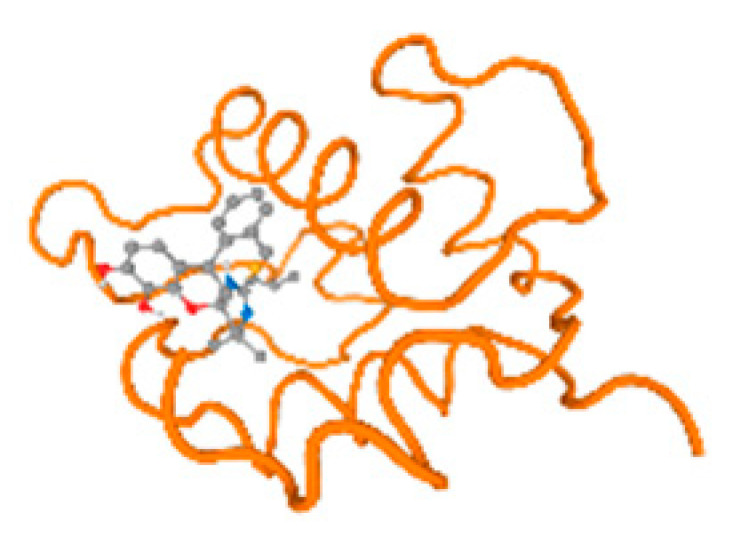	−8.7 (Lig. Pos. 3/80) 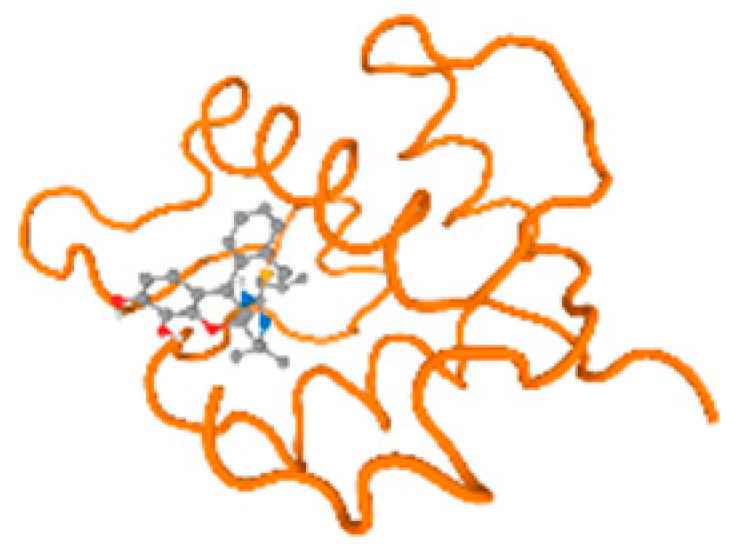
		6PE7	−8.71 (Lig. Pos. 3/100) 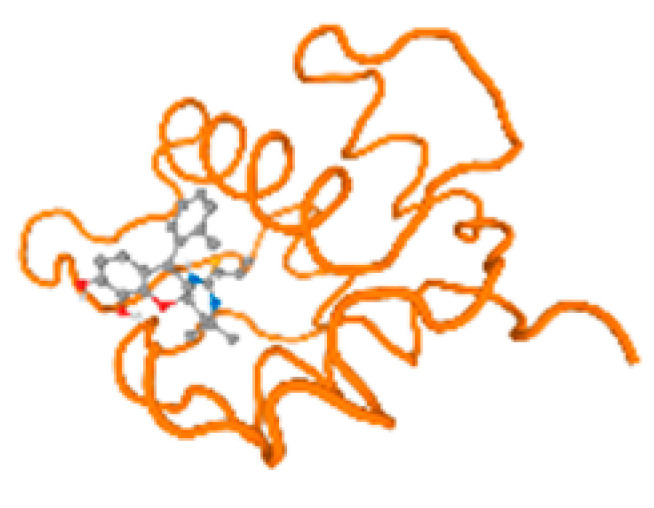	−8.66 (Lig. Pos. 3/100) 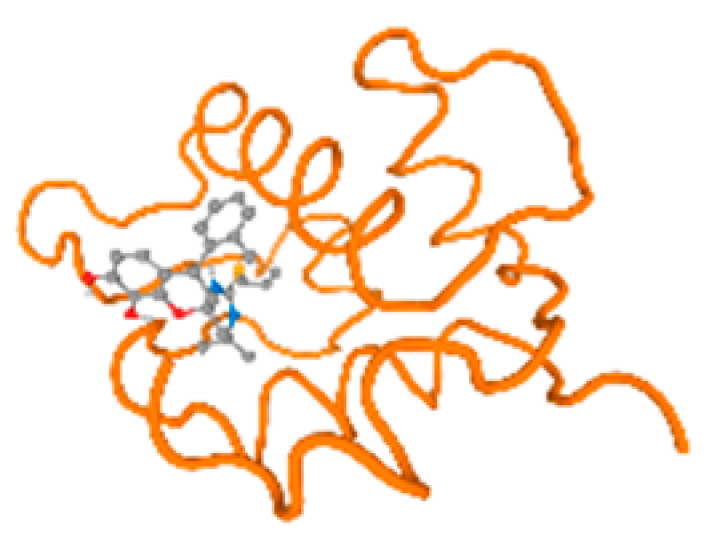	−8.71 (Lig. Pos. 3/100) 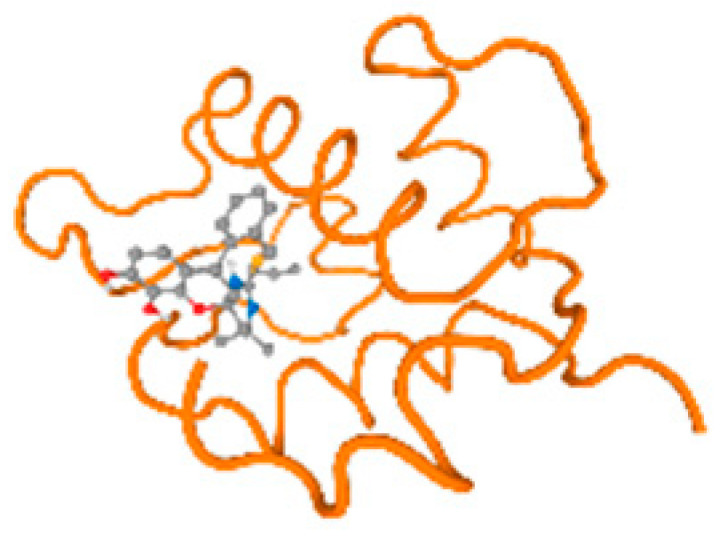	−8.71 (Lig. Pos. 3/60) 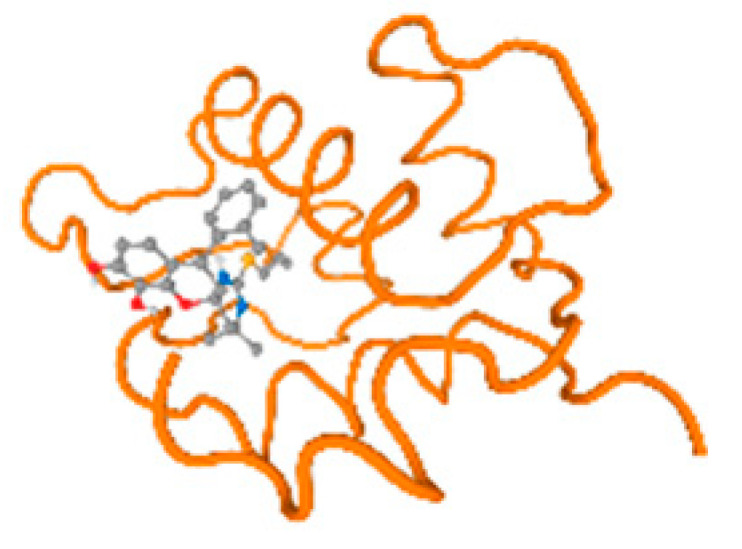
IL-1𝛽	Target Receptor	4GAF	−8.69 (Lig. Pos. 3/80) 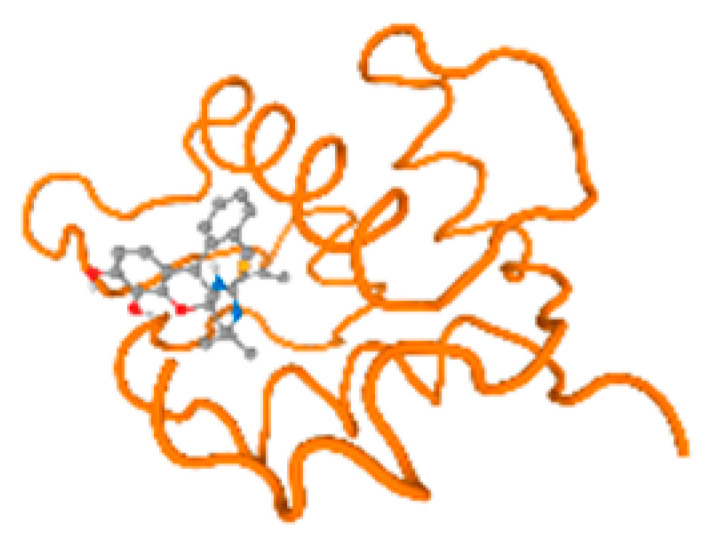	−8.7 (Lig. Pos. 3/100) 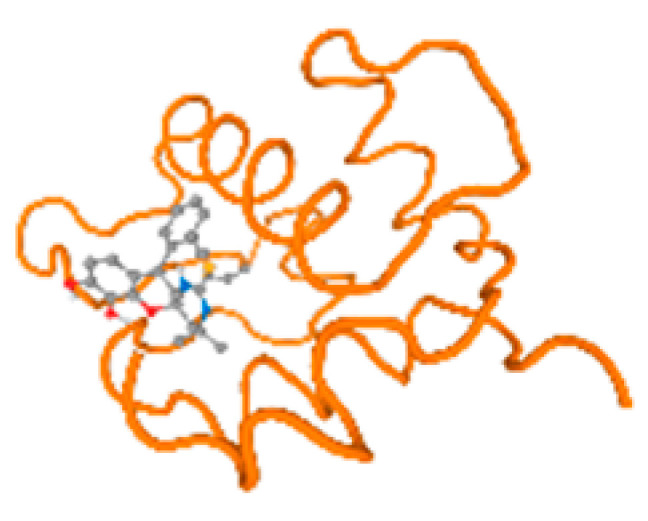	−8.72 (Lig. Pos. 3/90) 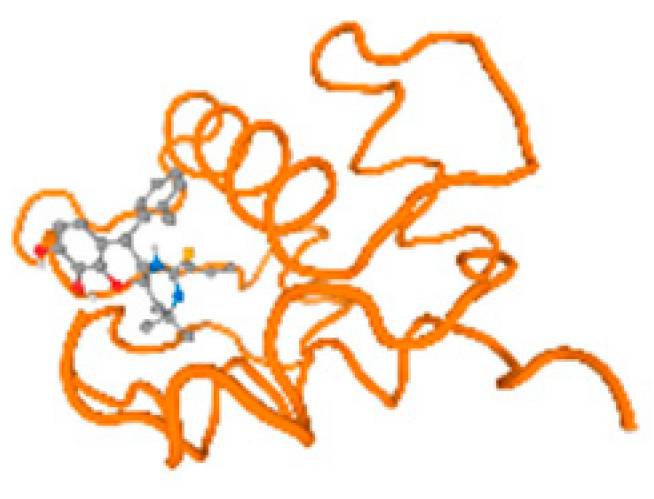	−8.67 (Lig. Pos. 3/90) 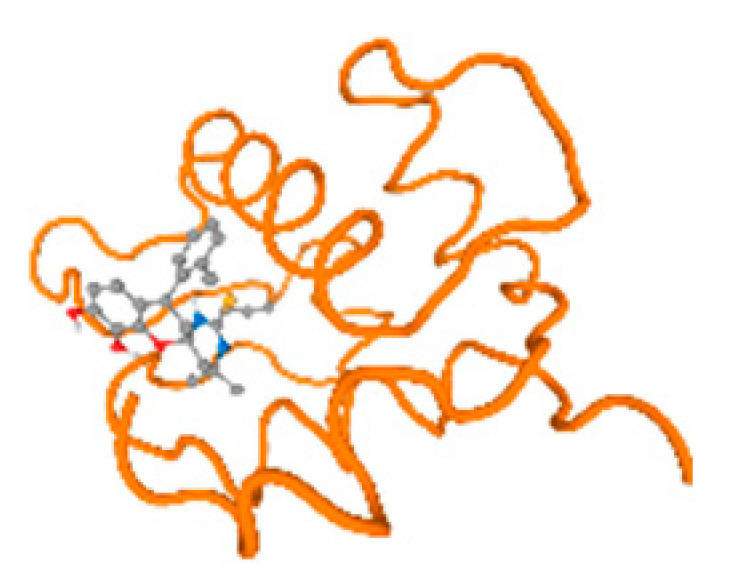
		4GAI	−8.67 (Lig. Pos. 3/100) 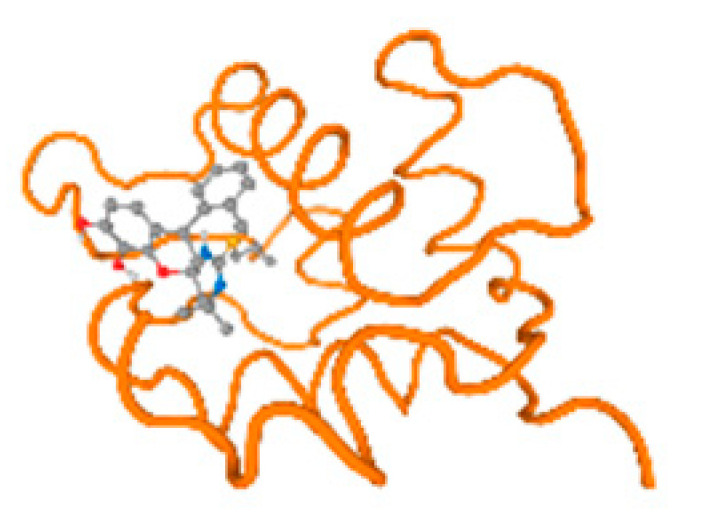	−8.51 (Lig. Pos. 3/100) 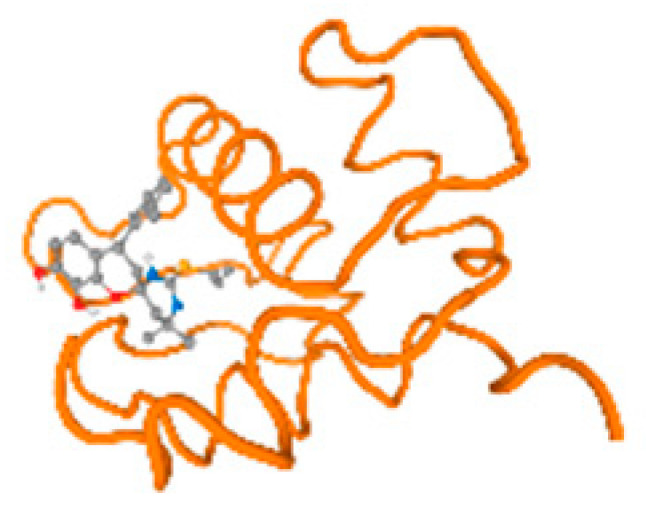	−8.63 (Lig. Pos. 3/100) 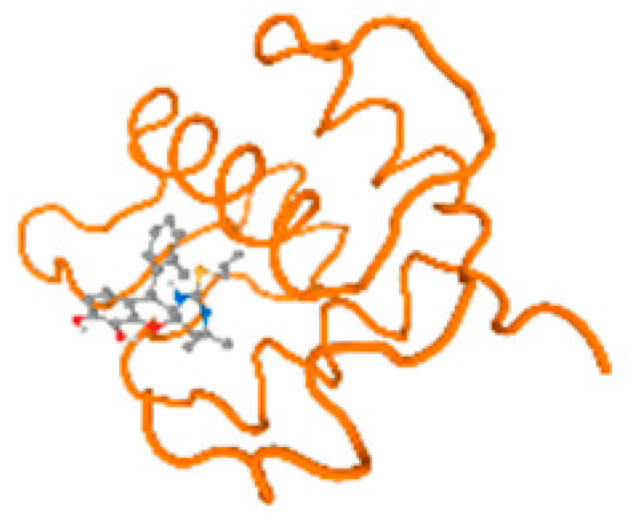	−8.74 (Lig. Pos. 3/70) 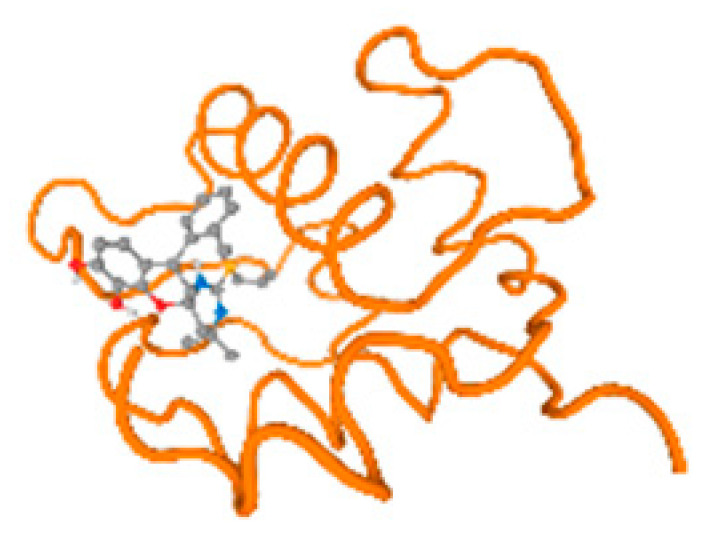
COX-2	Target Protein	5JW1	−8.66 (Lig. Pos. 3/100) 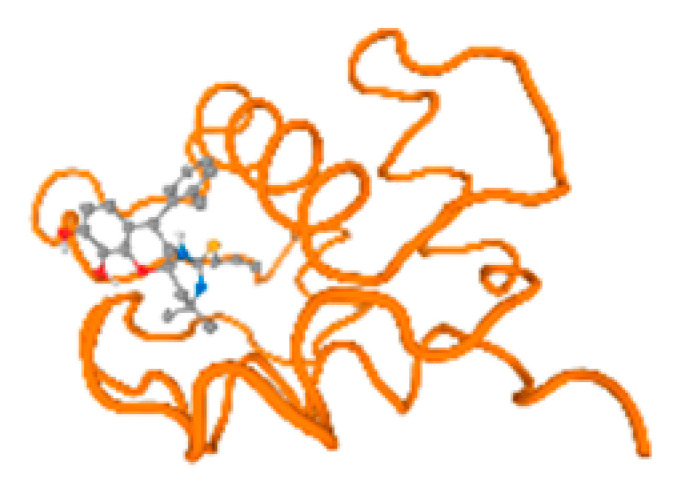	−8.66 (Lig. Pos. 3/80) 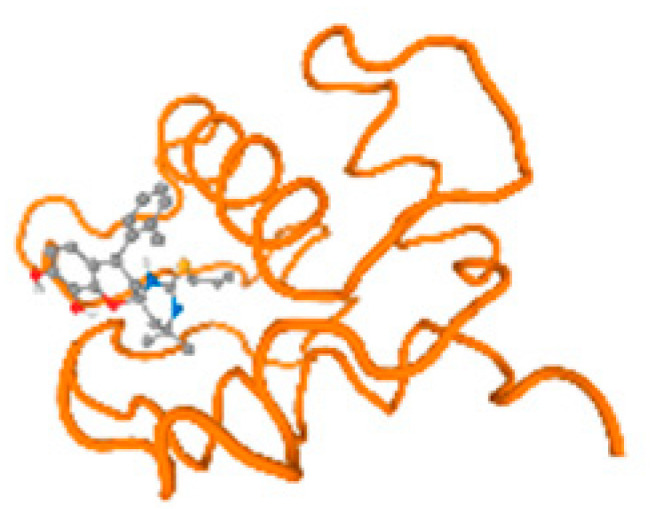	−8.69 (Lig. Pos. 3/100) 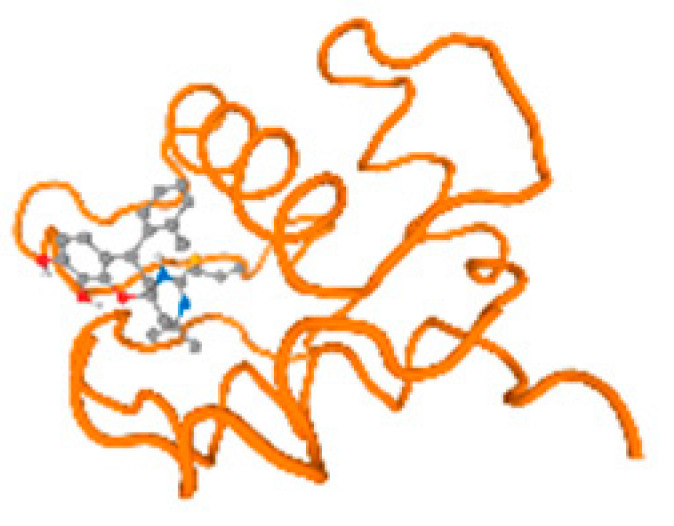	−8.6 (Lig. Pos. 3/100) 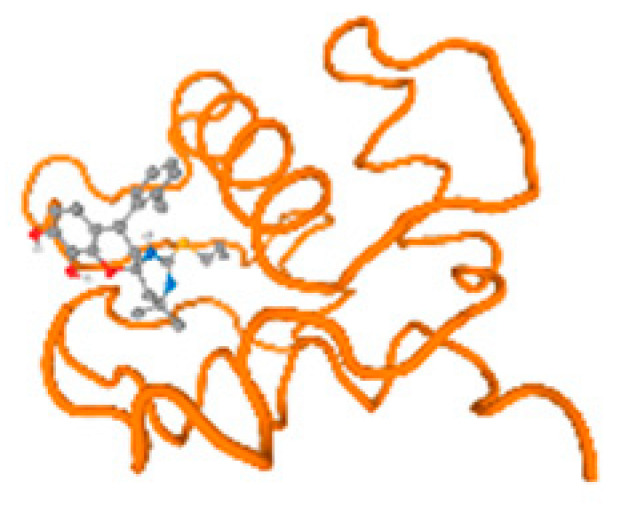
	Target Receptor	4E1G	−8.68 (Lig. Pos. 3/90) 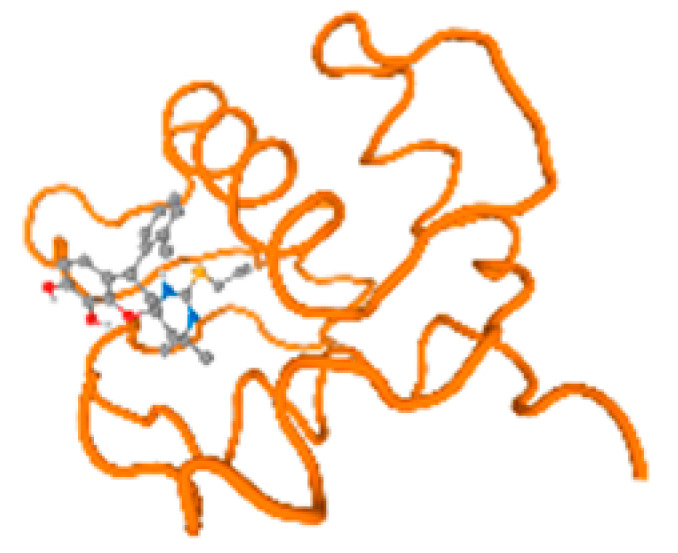	−8.71 (Lig. Pos. 3/100) 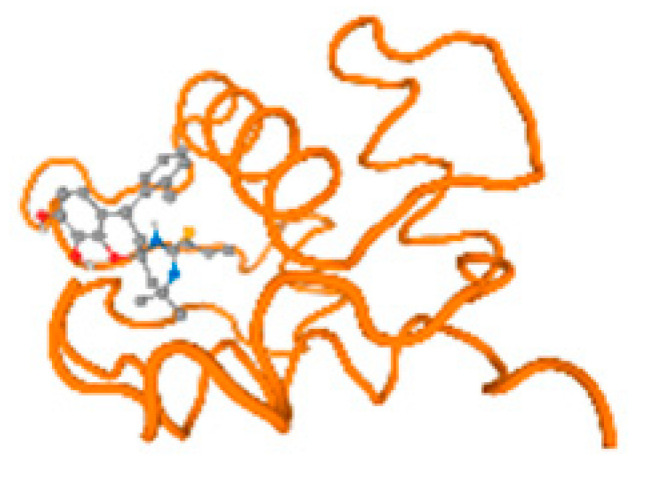	−8.62 (Lig. Pos. 3/100) 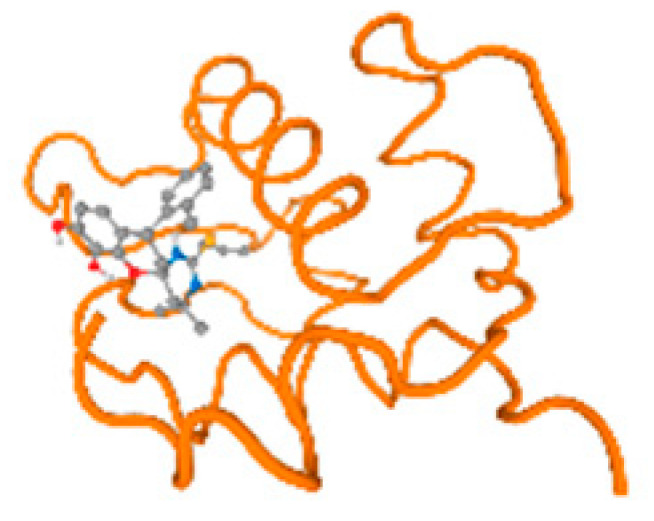	−8.72 (Lig. Pos. 3/100) 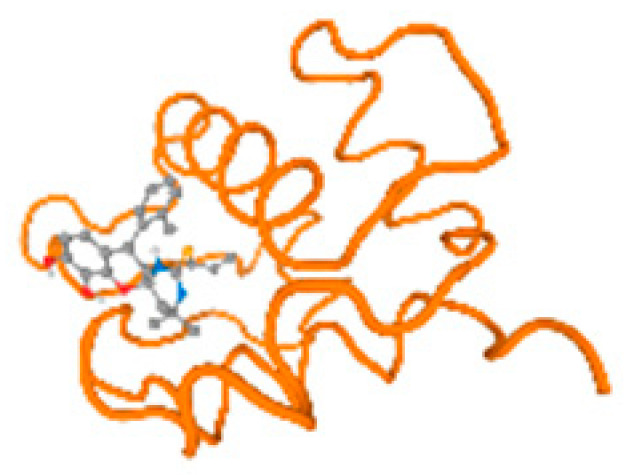
		3TZI	−8.68 (Lig. Pos. 3/100) 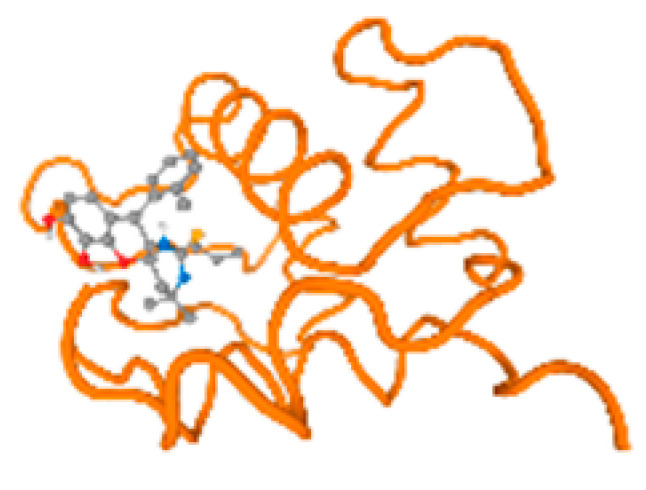	−8.68 (Lig. Pos. 3/100) 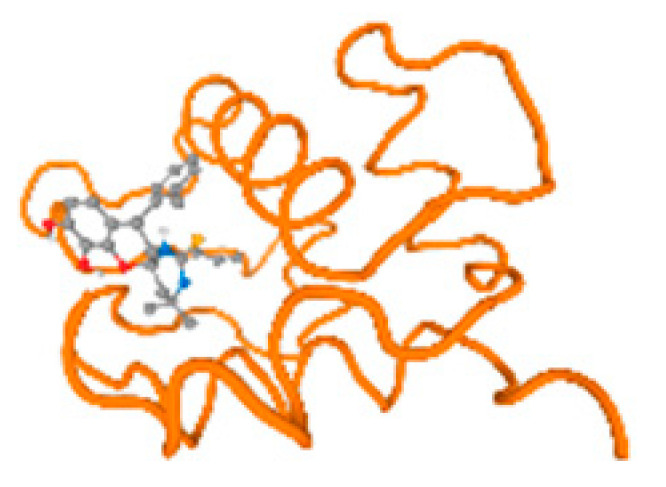	−8.66 (Lig. Pos. 3/100) 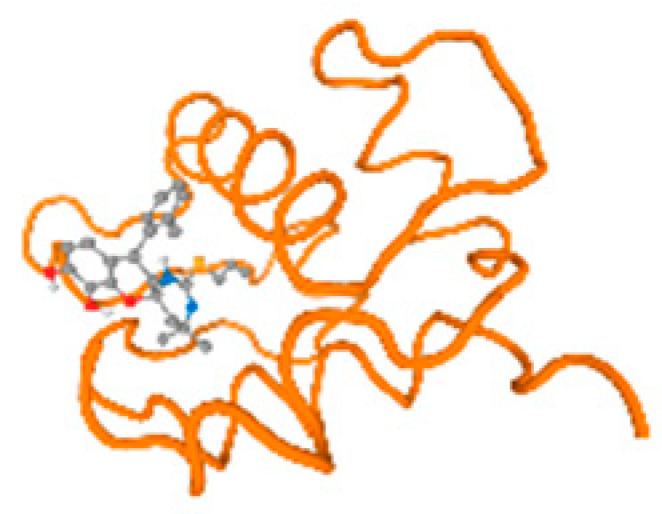	−8.62 (Lig. Pos. 3/100) 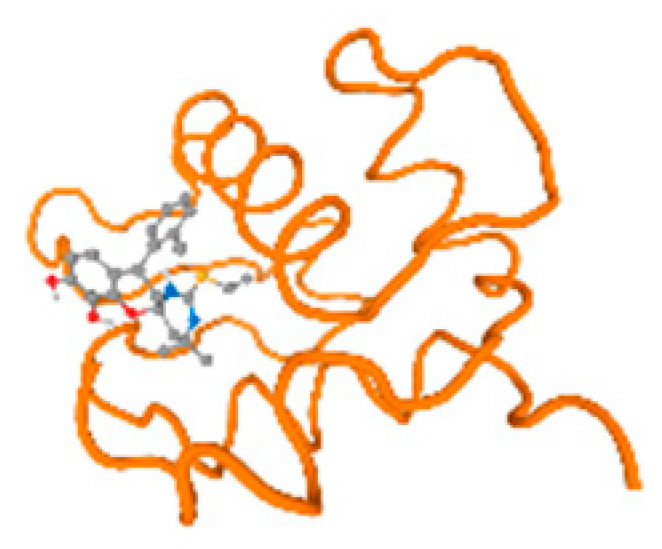
LOX-1	Target Receptor	1YPO	−8.73 (Lig. Pos. 3/100) 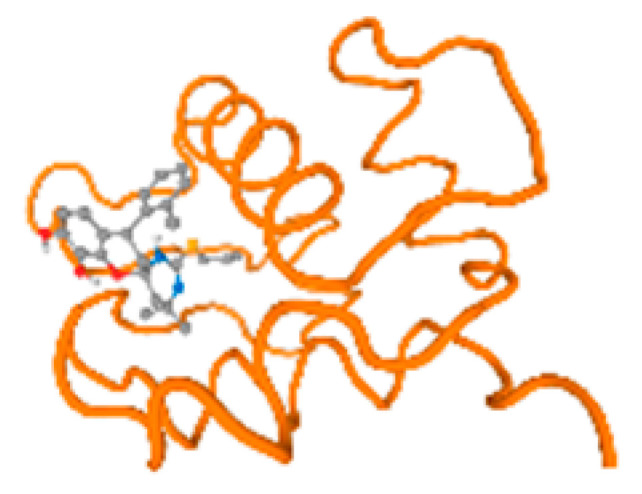	−8.71 (Lig. Pos. 3/100) 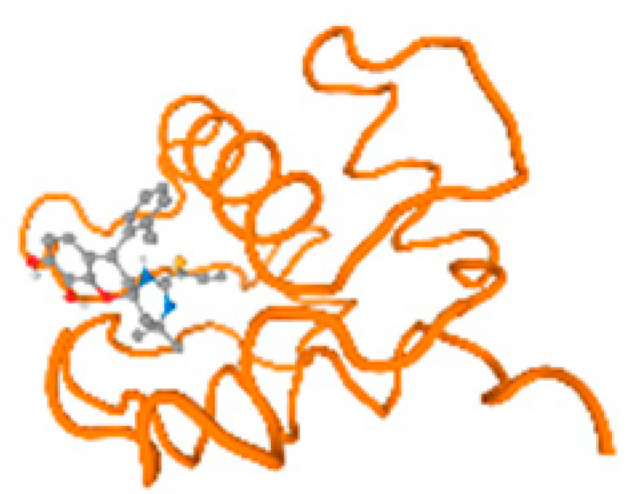	−8.72 (Lig. Pos. 3/100) 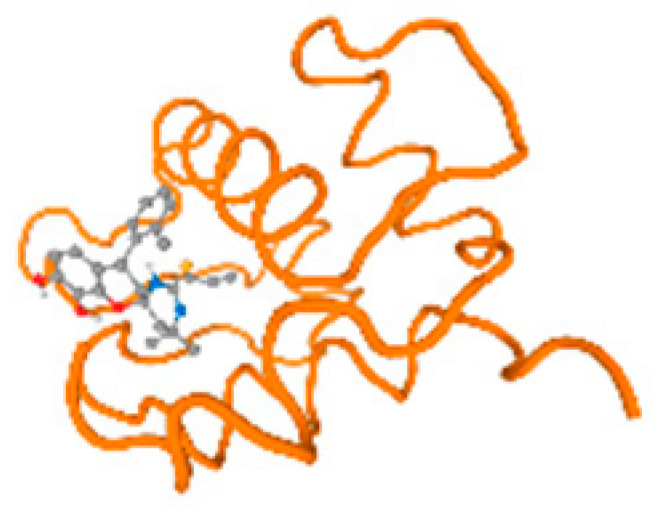	−8.71 (Lig. Pos. 3/100) 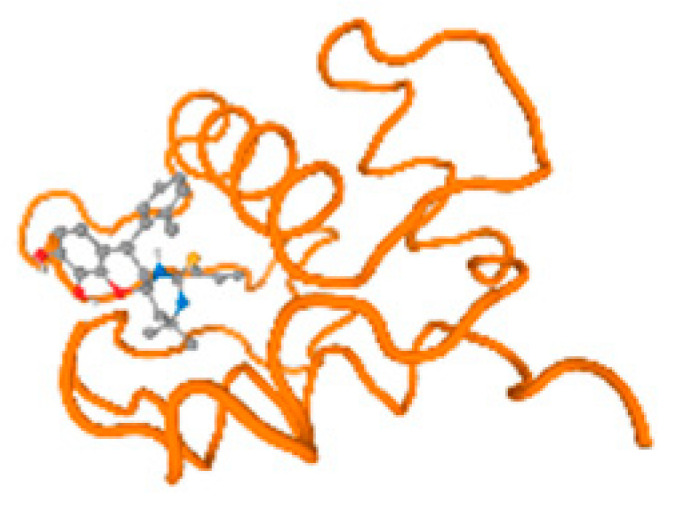
		1YPU	−8.74 (Lig. Pos. 3/100) 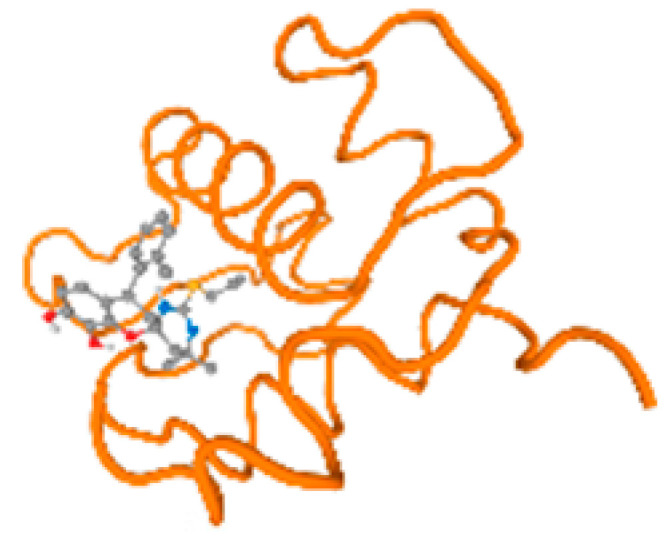	−8.7 (Lig. Pos. 3/100) 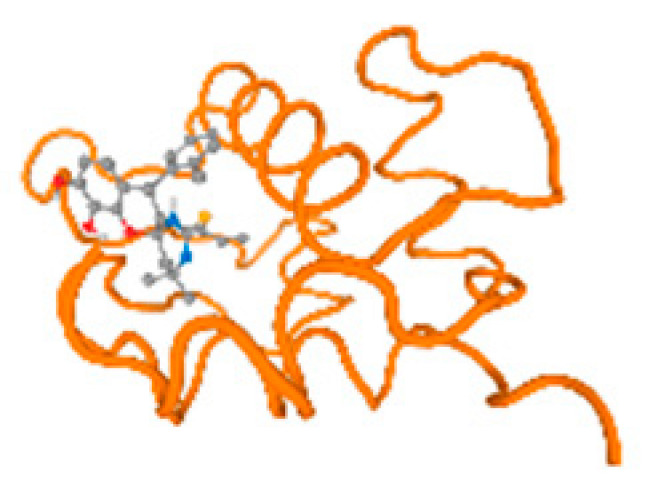	−8.64 (Lig. Pos. 3/60) 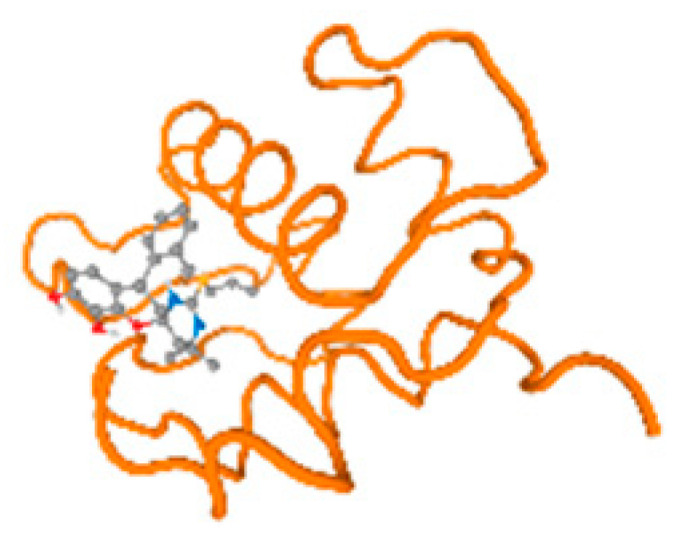	−8.73 (Lig. Pos. 3/100) 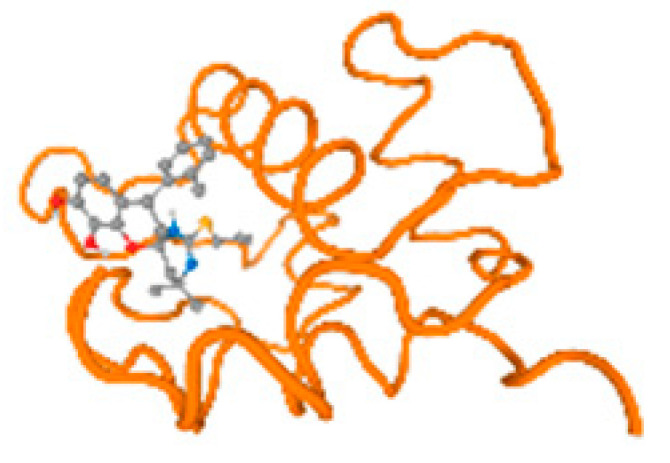

## Data Availability

No new data were created or analyzed in this study. Data sharing is not applicable to this article.

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
