# Peer review of "A Bio-Indicator Pilot Study Screening Selected Heavy Metals in Female Hair, Nails, and Serum from Lifestyle Cosmetic, Canned Food, and Manufactured Drink Choices"

_molecules, 2023, doi:10.3390/molecules28145582_

Round 1

Reviewer 1 Report

The authors have presented research regarding the key factors contributing to health issues related to heavy metals, such as lifestyles, genetic predispositions, environmental factors, and geographical regions. It states that heavy metals can enter the body through various sources, including the environment, daily lifestyle choices, food, beverages, cosmetics, and other products. The accumulation of heavy metals in the human body can lead to neurological problems, carcinogenic effects, organ failure, and reduced treatment sensitivity.

The study mentioned in the text focuses on screening chromium (Cr), aluminum (Al), lead (Pb), and cadmium (Cd) in selected foods, beverages, and cosmetics products. Female volunteers participated in the study, and their questionnaire responses were used to determine their exposure to these heavy metals. The researchers also analyzed the levels of Cr, Al, Pb, and Cd in the volunteers' hair, nails, and serum using inductively coupled plasma–mass spectrometry (ICP–MS). Additionally, serum cholinesterase and complete blood count (CBC) were examined. An AutoDock study was conducted to investigate the potential interaction of Cr, Al, Pb, and Cd as ligands.

The results of the questionnaire indicated that 71 percent of the female volunteers used cosmetics, food, and beverages contaminated with Cr, Al, Pb, and Cd, leading to a high concentration of these heavy metals in the serum, nails, and hair of these individuals. In contrast, only 29 percent of the female volunteers who did not use the studied samples exhibited lower levels of these heavy metals. The study also found elevated cholinesterase levels in the serum of the first group (high heavy metal exposure) compared to the second group (low heavy metal exposure). This finding was confirmed by the Autodock results. Furthermore, the CBC results showed negative variations compared to the reference ranges.

The text concludes by suggesting that future studies should focus on understanding the effects of heavy metal contamination on various human organs individually.

Overall, the text provides an overview of the key factors related to heavy metal exposure and highlights the results of a specific study involving female volunteers. It touches upon the methodology used, the prevalence of heavy metal contamination in the studied samples, and the potential health consequences observed in the participants. However, without additional details or context, it is challenging to assess the full scope and implications of the study.

Streamline the information: Some parts of the introduction contain repetitive information or details that could be condensed. Focus on providing concise and relevant information to maintain the reader's interest and ensure clarity.

Expand on the importance of monitoring: While the introduction mentions the importance of continuous screening and monitoring of heavy metal toxicity, it would be beneficial to elaborate on why this monitoring is crucial. Discuss the potential health risks, the need for regulatory measures, and the impact on public health.

Not thing to be declared

Author Response

Comments and Suggestions for Authors

Dear reviewer thank you for your comments that make our work more clear: with due all respect,

Reviewer 1 comment Point 1: Overall, the text provides an overview of the key factors related to heavy metal exposure and highlights the results of a specific study involving female volunteers. It touches upon the methodology used, the prevalence of heavy metal contamination in the studied samples, and the potential health consequences observed in the participants. However, without additional details or context, it is challenging to assess the full scope and implications of the study.

Author reply:

We agree with the reviewer's opinion,

yes, it is challenging to assess the full scope and implications of the study.

this is a pilot study (self-funding by the author), this study is the pilot introduction of our next project that will be included by chemical assay of all heavy metals and minerals as well as will assay (blood film, liver functions, kidney functions, endocrinology profile, cardiac functions, tumor markers, other biochemical investigation), in progress.

Reviewer 1 comment Point 2: Streamline the information: Some parts of the introduction contain repetitive information or details that could be condensed. Focus on providing concise and relevant information to maintain the reader's interest and ensure clarity.  

Author reply: Done.

Reviewer 1 comment Point 3: Expand on the importance of monitoring: While the introduction mentions the importance of continuous screening and monitoring of heavy metal toxicity, it would be beneficial to elaborate on why this monitoring is crucial. Discuss the potential health risks, the need for regulatory measures, and the impact on public health.  

Author reply: Done inside the manuscript

We will wait and appreciate other comments if any, with due all respect.

Reviewer 2 Report

This study briefly explains heavy metal contamination in food and other used items. However, some questions were raised. Please see the PDF attached.

There is too much data present in the paper. Did you think all the data is suitable in a single paper?

The introduction section need  to add a table mention heavy metals there impact on human and environment, source, acceptable ranges with references. This will increase the reader interest

Author Response

Comments and Suggestions for Authors

Dear reviewer thank you for your comments that make our work more clear: with due all respect,

Reviewer 2 comment Point 1: There is too much data present in the paper. Did you think all the data is suitable in a single paper?

Author reply: We agree with the reviewer's comment yes, it is so much data because:

1- this is an introduction to our next study and we have to support our next fund with strong data since we need support for our next project, where this study (self-funding by the author), this study is the pilot introduction of our next project which will be included by chemical assay of all heavy metals and minerals as well as will assay (blood film, liver functions, kidney functions, endocrinology profile, cardiac functions, tumor markers, other biochemical investigation), in progress.

2- our intention is to support our readers (which may be undergraduate students, master, or Ph.D. students) So, so they may need accumulated and expanded data and we hope to help them in their work.

Reviewer 2 comment Point 2: The introduction section need to add a table mention heavy metals there impact on human and environment, source, acceptable ranges with references. This will increase the reader interest.

Author reply: Done

We will wait and appreciate other comments if any, with due all respect.

Reviewer 3 Report

In the work entitled “A Bio-Indicator Pilot Study of Screening Selected Heavy Metals in the Female’s Hair, Nails, and Serum Due to the Lifestyle Cosmetic and Canned Food and Manufactured Drink Choices”, the authors screened Cr, Al, Pb, and Cd in selected foods, beverages, and cosmetics products and also screened their content in the hairs, nails, and serum by inductively coupled plasma–mass spectrometry (ICP–MS) in female volunteers. They also analyzed serum cholinesterase and complete blood count and performed an AutoDock study of Cr, Al, Pb, and Cd as potential Ligands. The manuscript is well organized with necessary data and within the scope of this Journal. The introduction and background are reasonable given the promise of the paper. Figures and tables are comprehensive and helpful. In general, the manuscript needs corrections to be published and need to be addressed before acceptance. There are some comments in attached file, which should be addressed. My comments regarding this article are as follows,

1.       The manuscript has grammatical errors and needs improvement.

2.       The abstract should be revised to address the development and novelty of this work, especially the superiority or enhancement when compared with other advances.

3.       Ensure that all abbreviations are defined for the first time they appear in the abstract, main text, figures, tables and check throughout the main text (LINE 18).

4.       Italicize “in vivo”, “in vitro” and “in silico” throughout the manuscript including References.

5.       Line 826 superscript 3 in HNO3

6.       The authors should use the third person pronouns instead of personal pronouns throughout the paper including the lines 835-844.

7.       All the figures need to be revised in consistent layout/marks to improve the readability. High resolution figures and revisions are also required to clearly show the details, especially Figure 6.

8.       In materials and methods, line 796, BMI stands for what?

9.       Put significant letters on data presented in the Figures 2-6.

10.   In the line 115 authors described the concentration of Cr, Al, Cd, and Pb in each cosmetics product. However two samples (5 and 6) both are body lotion, what is deference between these two products? Are they different brands of body lotion? # (sample 1= Canned meat, sample 2 = Canned meat)?

11.   Conclusion is long, present the main outcome of the study in the conclusion, but do not repeat the materials and methods and results presented in the text.

12.   The references cited are in more instances old (highlighted in red in the references list), it is of utmost importance to cite newest possible references, and not the ones that are 20-25 or 40 years old. Majority of cited publications must be published in the last 10 years meaning from 2013 to 2023. Exchange the old references with new ones.

Moderate editing of English language required

Author Response

Comments and Suggestions for Authors

Dear reviewer thank you for your comments that make our work more clear: with due all respect, We did all the comments inside the manuscript.

Reviewer 3 comment Point 1: The manuscript has grammatical errors and needs improvement.

Author reply: Done.

Reviewer 3 comment Point 2: The abstract should be revised to address the development and novelty of this work, especially the superiority or enhancement when compared with other advances.  

Author reply: Done.

Reviewer 3 comment Point 3:  Ensure that all abbreviations are defined for the first time they appear in the abstract, main text, figures, tables and check throughout the main text (LINE 18).

Author reply: Done.

Reviewer 3 comment Point 4: Italicize “in vivo”, “in vitro” and “in silico” throughout the manuscript including References.

Author reply: Done (according to MDPI policy).

Reviewer 3 comment Point 5:  Line 826 superscript 3 in HNO3

Author reply: Done.

Reviewer 3 comment Point 6: The authors should use the third person pronouns instead of personal pronouns throughout the paper including the lines 835-844.

Author reply: Done (we send the manuscript to a grammar editing service.).

Reviewer 3 comment Point 7: All the figures need to be revised in consistent layout/marks to improve the readability. High resolution figures and revisions are also required to clearly show the details, especially Figure 6.

Author reply: Done (we did all the figures as 300dpi).

Reviewer 3 comment Point 8:  In materials and methods, line 796, BMI stands for what?

Author reply: Done (Body Mass Index).

Reviewer 3 comment Point 9: Put significant letters on data presented in the Figures 2-6.

Author reply: Done, by stars, only in Figure 5, because in Figures 2, 3, and 4 we study non-biological samples (single sample), in (non-biological sample), we just assay the presence of heavy metals between single samples. but in Figure 5 (biological sample) we study the presence of heavy metal between two studies population with mean and standard divisions, so we adjust studies significantly in detail in Table 2.

In Figure 6 we also study an individual docking position and individual binding energy to individual receptor, we didn’t do any significant study.

Reviewer 3 comment Point 10:  In the line 115 authors described the concentration of Cr, Al, Cd, and Pb in each cosmetics product. However, two samples (5 and 6) both are body lotion, what is deference between these two products? Are they different brands of body lotion? # (sample 1= Canned meat, sample 2 = Canned meat)?

Author reply: Done (yes, they are different brands).

Reviewer 3 comment Point 11: Conclusion is long, present the main outcome of the study in the conclusion, but do not repeat the materials and methods and results presented in the text.

Author reply: Done.

Reviewer 3 comment Point 12: The references cited are in more instances old (highlighted in red in the references list), it is of utmost importance to cite newest possible references, and not the ones that are 20-25 or 40 years old. Majority of cited publications must be published in the last 10 years meaning from 2013 to 2023. Exchange the old references with new ones.

Author reply: Done (as possible most of them, after your permission there is some essential references can’t remove).

We will wait and appreciate other comments if any, with due all respect.

Round 2

Reviewer 1 Report

Thank you for the revision..

I don't feel like judging. 

Author Response

Dear reviewer, 

with due all respect,

Thank you for your revision and help that make our work better. 

Reviewer 2 Report

The authors have addressed all the issues, but still, one question is not addressed. 

Why do they select females only for this study? Read the previous comments in the PDF.

Author Response

Dear reviewer, 
with due all respect,
Thank you for your revision and help that make our work better. 

Forgive us for miss oversight regarding answer comments, we choose female volunteers where we work in a university for females only, and this is a pilot study it will be our basis for the next project we will ask for cooperation from the male section in the next work, in progress. 

Reviewer 3 Report

Accept

Minor editing of English language required

Author Response

(The authors gave the same response as above.)
